# PCLAF-DREAM drives alveolar cell plasticity for lung regeneration

Bongjun Kim [1] ✉, Yuanjian Huang[1], Kyung-Pil Ko[1], Shengzhe Zhang[1], Gengyi Zou[1], Jie Zhang[1], Moon Jong Kim [1], Danielle Little[2], Lisandra Vila Ellis[2], Margherita Paschini[3], Sohee Jun[1], Kwon-Sik Park [4], Jichao Chen [2], Carla Kim [3] & Jae-Il Park [1,5,6] ✉

Cell plasticity, changes in cell fate, is crucial for tissue regeneration. In the lung, failure of regeneration leads to diseases, including fibrosis. However, the mechanisms governing alveolar cell plasticity during lung repair remain elusive. We previously showed that PCLAF remodels the DREAM complex, shifting the balance from cell quiescence towards cell proliferation. Here, we find that PCLAF expression is specific to proliferating lung progenitor cells, along with the DREAM target genes transactivated by lung injury. Genetic ablation of *Pclaf* impairs AT1 cell repopulation from AT2 cells, leading to lung fibrosis. Mechanistically, the PCLAF-DREAM complex transactivates *CLIC4*, triggering TGF-β signaling activation, which promotes AT1 cell generation from AT2 cells. Furthermore, phenelzine that mimics the PCLAF-DREAM transcriptional signature increases AT2 cell plasticity, preventing lung fibrosis in organoids and mice. Our study reveals the unexpected role of the PCLAF-DREAM axis in promoting alveolar cell plasticity, beyond cell proliferation control, proposing a potential therapeutic avenue for lung fibrosis prevention.

Lung disease is one of the leading causes of death, with one in six deaths worldwide[1]. Various environmental pathogens and stresses cause lung injuries[2–4]. Stem and progenitor cell–driven regenerative potency repairs damaged lungs[5,6]. The failure of lung stem and progenitor cell activation or differentiation results in lung disease, including pulmonary fibrosis[5–8]. AT2 cells have been suggested as a pivotal cell type in lung regeneration. Dysregulation of the facultative progenitor capacity of AT2 cells induces lung fibrosis rather than regeneration[7,9–13]. Downregulation of Toll-like receptor 4-hyaluronan interaction[7] and sustained mechanical tension[13] inhibits AT2 cell stemness and plasticity. Telomere dysfunction[10] and mutant surfactant protein C (SPC)[11] promote AT2 cell hyperplasia, causing the failure of lung regeneration. Therefore, the precise control of AT2 cells is vital

for lung regeneration. However, the detailed mechanism of AT2 cell activation and plasticity remains ambiguous.

The PCLAF (PCNA Clamp Associated Factor; also known as KIAA0101 or PAF) was identified as a proliferating cell nuclear antigen (PCNA)-interacting protein modulating DNA repair and replication[14,15]. PCLAF also hyperactivates WNT/β-catenin signaling independently of PCNA in colorectal cancer[16], promotes pancreatic tumorigenesis[17], induces cell plasticity of breast cancer[18], facilitates stem cell activation and proliferation for intestinal regeneration[19], and accelerates lung tumorigenesis via the DREAM complex[20].

The dimerization partner, retinoblastoma (RB)-like, E2F, and multi-vulval class B (DREAM) complex is an evolutionarily conserved multiprotein complex that orchestrates cell quiescence and the cell

[1]Department of Experimental Radiation Oncology, Division of Radiation Oncology, The University of Texas MD Anderson Cancer Center, Houston, TX 77030, USA. [2]Department of Pulmonary Medicine, The University of Texas MD Anderson Cancer Center, Houston, TX 77030, USA. [3]Stem Cell Program and Divisions of Hematology/Oncology and Pulmonary Medicine, Boston Children's Hospital, Boston, MA 02115, USA. [4]Department of Microbiology, Immunology, and Cancer Biology, University of Virginia, Charlottesville, VA 22908, USA. [5]Graduate School of Biomedical Sciences, The University of Texas MD Anderson Cancer Center, Houston, TX 77030, USA. [6]Program in Genetics and Epigenetics, The University of Texas MD Anderson Cancer Center, Houston, TX 77030, USA. ✉e-mail: bkim6@mdanderson.org; jaeil@mdanderson.org

cycle[21–24]. Dissociation of the multi-vulval class B (MuvB) core complex with RBL2 (retinoblastoma-like protein 2/p130), E2F4, and DP1 (E2F dimerization partner 1) drives the MuvB complex to bind to BYMB and FOXM1, transactivating cell cycle-related DREAM target genes and leading to cell quiescence exit and cell proliferation[25]. PCLAF directly binds to and remodels the DREAM complex to bypass cell quiescence and promote cell proliferation[20]. Given the roles of the PCLAF in modulating cell proliferation, cell plasticity, or stemness in various pathophysiological conditions[16–20], we interrogated the roles of the PCLAF-DREAM axis in lung regeneration.

Employing comprehensive approaches, including single-cell transcriptomics, organoids, and mouse models, we herein showed that the PCLAF-DREAM-mediated alveolar cell plasticity is indispensable for lung regeneration. PCLAF depletion impaired lung regeneration and led to lung fibrosis with decreased repopulation of alveolar type I (AT1) cells. In addition, we found that the PCLAF-DREAM complex transactivates *CLIC4* to activate TGF-β signaling for AT1 cell generation from alveolar type II (AT2) cells. Furthermore, we identified a potential drug candidate mimicking the PCLAF-activated transcriptome for suppressing lung fibrosis.

## Results

### PCLAF-positive cells are elevated during lung regeneration
We first characterized PCLAF-positive cells in the lungs by analyzing the dataset of single-cell RNA-sequencing (scRNA-seq) of murine (GSE141259)[26] and human (GSE135893)[27] lung tissues. The mouse lungs showed that Mki67⁺ proliferating alveolar progenitor cells (PAPCs) specifically expressed *Pclaf* (Fig. 1A–C). These PAPCs barely expressed mature lung epithelial cell markers or lung stem/progenitor cell markers (Supplementary Fig. 1A, C, D, and Supplementary Data 1). The human lungs also showed specific expression of *PCLAF* in PAPCs (Fig. 1D–F, Supplementary Fig. 1B, E, F, and Supplementary Data 2). The bleomycin-induced lung injury model is widely used to study lung injury, regeneration, and fibrosis[28]. Upon the instillation of bleomycin, the number of PAPCs was elevated in the mouse lungs (Fig. 1G). The expressions of inflammatory genes, *Tnfα*, *Il6*, and *Cox2*, reached their peak at 3 days post-injury (dpi) (Fig. 1H) and the ratio of *Bax/Bcl2*, which reflects cell apoptosis, also reached its maximum level at 3 dpi (Fig. 1I). Bleomycin injury also led to increased *Pclaf* expression, which peaked at 7 dpi and decreased at 14 dpi in the mouse lung tissues (Fig. 1H), consistent with the immunostaining results for PCLAF (Fig. 1J). In addition, *Pclaf-lacZ* knock-in mice validated by LacZ expression in the intestinal crypt base columnar cells as previously identified[19] (Supplementary Fig. 2A, B) displayed a similar increase in PCLAF⁺ cells in the regenerating lungs (Fig. 1K–M). Of note, PCLAF⁺ cells were also found in the immune and mesenchymal cells (Supplementary Fig. 2C–F).

### Pclaf is indispensable for lung regeneration
Next, to determine the role of PCLAF in lung regeneration, we tested several concentrations of bleomycin (1.4, 2.8, and 7 U/kg) in the lung and monitored blood oxygen levels (peripheral oxygen saturation [SpO₂]). Mice with a lower concentration of bleomycin (1.4 U/kg) displayed a restoration of SpO₂ levels at 14 dpi, whereas those with 2.8 U/kg and 7 U/kg exhibited a decrease in SpO₂ levels until 14 dpi (Supplementary Fig. 3A). Thus, we administered a low dose of bleomycin (1.4 U/kg) to *Pclaf* wild-type (WT) and knock-out (KO) mice to examine lung regeneration (Fig. 2A). Bleomycin initially damages the alveolar epithelium, including AT1 and AT2 cells, followed by interstitial fibrosis[8,28,29]. At the early stage of bleomycin administration, both *Pclaf* WT and KO mice exhibited a gradual decrease in blood oxygen levels and breath rate per minute until 5 dpi (Fig. 2B). Interestingly, *Pclaf* KO mice showed the attenuated and delayed restoration of SpO₂ saturation and breath rate per minute at 9 dpi and decreased O₂ saturation at 21 dpi (Fig. 2B). We initially hypothesized that PCLAF is involved in

activating lung progenitor cells based on our previous finding that PCLAF drives cell quiescence exit[20]. Unexpectedly, *Pclaf* KO lung showed increased proliferating epithelial cells at the regeneration stage (7 dpi, Supplementary Fig. 3B, C). Cell apoptosis was not altered by *Pclaf* KO at the acutely damaged stage (3 dpi, Supplementary Fig. 3D, E). At 3 dpi, both *Pclaf* WT and KO mice showed a severely reduced ratio of RAGE / CDH1 area (AT1) and number of SPC⁺ (AT2) cells (Fig. 2C–F, and Supplementary Fig. 3F, G). However, consistent with the results of pulmonary functional analysis, *Pclaf* KO showed decreased AT1 cells at the regeneration stage (7 dpi) and chronic fibrosis stage (21 dpi) (Fig. 2C–F, and Supplementary Fig. 3F, G). Intriguingly, the number of AT2 cells was elevated in *Pclaf* KO mice upon bleomycin (Fig. 2C–F and Supplementary Fig. 3F, G). Unlike WT lung tissues, *Pclaf* KO lungs exhibited more inflamed and condensed tissue (Fig. 2G), with severe fibrotic features confirmed by picrosirius red (a dye staining collagen), αSMA/ACTA2 (a marker for myofibroblasts) staining, and hydroxyproline assay (a quantification of collagen from lung lysates) (Fig. 2G–L). These results suggest that *Pclaf* is indispensable for bleomycin-induced lung regeneration.

AT2 cells replenish AT1 cells during lung regeneration[30]. Having observed that *Pclaf* KO impaired lung regeneration with the aberrantly increased AT2 cells, we next determined the impact of *Pclaf* KO on cell plasticity between AT2 and AT1 cells by using lung organoids (LOs). We evaluated the lung organoid-forming efficiency (OFE) and cellular heterogeneity generation using well-established LO culture methods, co-culture systems of lung epithelial cells with lung endothelial cells (LuECs) in the 3D organoid air-liquid interface (Supplementary Fig. 4A)[31]. LOs with LuECs developed into both alveolar and bronchiolar types of organoids. (Supplementary Fig. 4B, C). We isolated and cultured lung epithelial cells (EPCAM⁺/TER119⁻/CD31⁻/CD45⁻) from *Pclaf* WT or KO mice at 7 dpi (bleomycin) with LuECs (Supplementary Fig. 4D). Notably, OFE was reduced in *Pclaf* KO cells compared to in *Pclaf* WT cells (Supplementary Fig. 5A, B). In addition, *Pclaf* KO LOs showed fewer alveolar (-15%) and bronchioalveolar types (-40%), whereas *Pclaf* WT LOs displayed alveolar (-33%) and bronchioalveolar types (-51%) (Supplementary Fig. 5C–E). The bronchiolar type of LOs was increased by *Pclaf* KO (-39%) compared to WT (-15%) (Supplementary Fig. 5C–E). Moreover, *Pclaf* KO alveolar-type LOs displayed fewer AT1 cells (Fig. 2M, N) and increased proliferating cells (Supplementary Fig. 5F, G), consistent with in vivo results. These data suggest that *Pclaf* is required for the plasticity of AT1 and AT2 cell lineages rather than cell proliferation during lung regeneration.

### Pclaf KO suppresses AT2 cell lineage plasticity
To further explore the cellular mechanism of PCLAF-controlled lung regeneration, we leveraged single-cell transcriptomics. We isolated pulmonary epithelial cells from *Pclaf* WT or KO lung tissues at 7 dpi (bleomycin) and performed scRNA-seq (Supplementary Fig. 6A). Cells were clustered and annotated based on the established cell markers (Fig. 3A, B, Supplementary Figs. 6B, 7, and Supplementary Data 3). The AT2 cells were refined into two subsets, 'AT2 cells' expressing the high level of *Slc34a2* and 'activated AT2 cells' specifically expressing *Lcn2*[26,32] (Fig. 3B and Supplementary Fig. 7D, E). In line with previous studies[26,30,32], a cell lineage trajectory inference using Slingshot[33] and RNA velocity[34] predicted the cellular trajectory from AT2 cells into AT1 cells in *Pclaf* WT (Fig. 3C–E and Supplementary Fig. 8). Activated AT2 cells serve as root cells that differentiate into AT1 cells through two different paths, via the intermediated cells, AT1med/AT2med cells or PAPCs and Krt8⁺ cells. Additional cell lineage trajectories from activated AT2 cells to AT2 cells through PAPCs were observed (Fig. 3C–E), implying self-regeneration of AT2 cells during bleomycin-induced lung regeneration. Intriguingly, unlike *Pclaf* WT, *Pclaf* KO lung showed a favored trajectory of PAPCs into AT2 cells (Fig. 3C–E). In addition, in *Pclaf* KO lungs, Krt8⁺ cells and AT1med/AT2med cells showed lineage trajectories into activated AT2 and AT2 cells, respectively, rather than

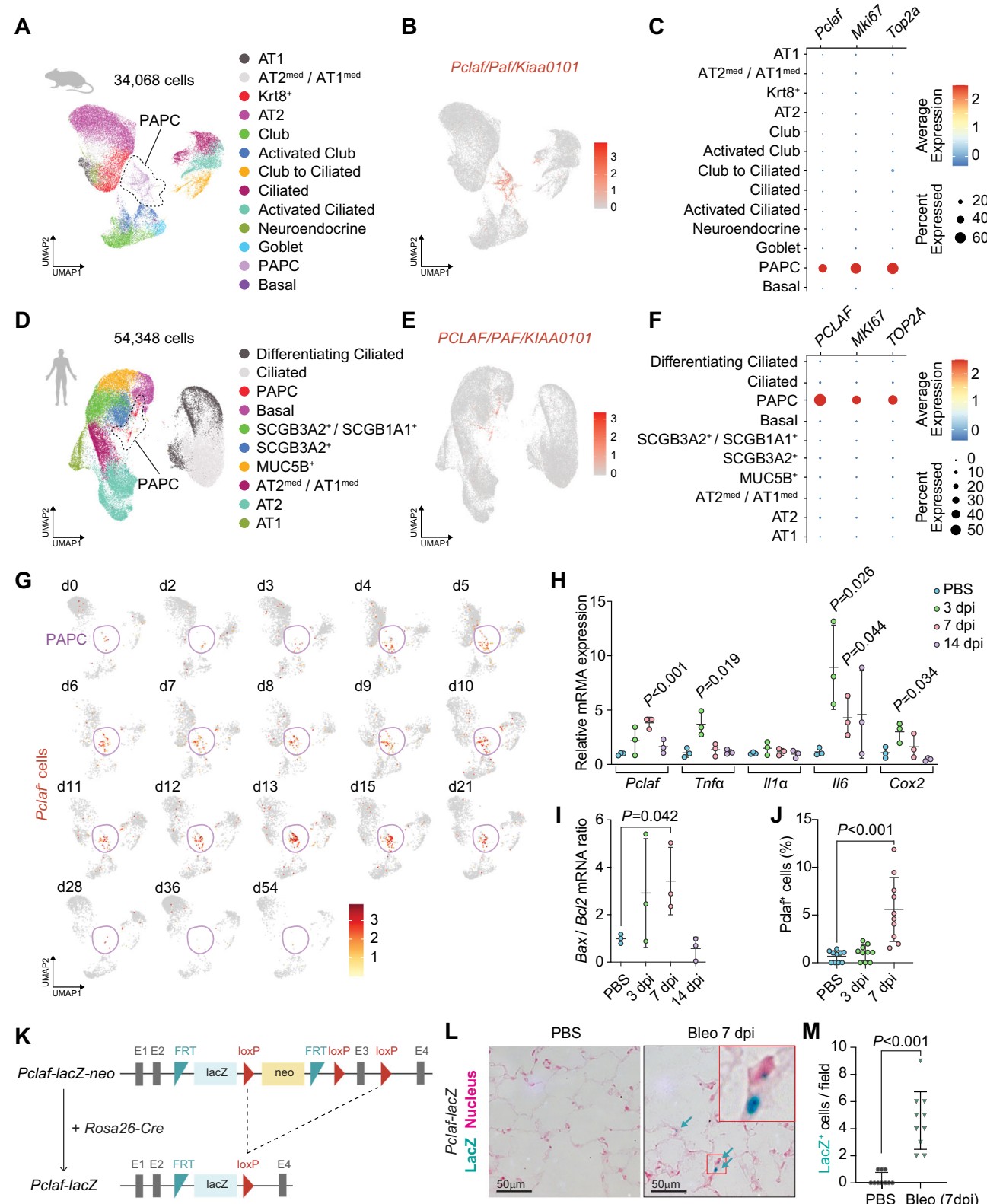

toward AT1 cells (Fig. 3C–E). Consistently, immunostaining of *Pclaf* KO lung tissues showed decreased AT1 cells and increased AT2 cells (Fig. 3F), consistent with in silico results. In *Pclaf* KO lung, KRT8+ cells (intermediate cells) were increased, while activated AT2 cells (LCN2+) were not changed, compared to WT (Fig. 3F and Supplementary Fig. 7F). Of note is that PAPCs were highly enriched with SPC, a representative marker for AT2 cells (Supplementary Fig. 9), implying that most PAPCs are likely derived from AT2 cells.

For experimental validation of the predicted cell lineage trajectories, we also performed a cell lineage tracing analysis using genetically engineered mice. We first established *Pclaf-fl (floxed)/fl* mice for conditional KO (cKO) of *Pclaf* alleles (Supplementary Fig. 10A, B). AT2 cell-specific Cre driver (*Sftpc^CreERT2* [*Sftpc-CreERT2*]) and Cre-loxP recombination reporter (*Rosa26^Sun1GFP* [*Sun1-GFP*]) were combined with either *Pclaf-/-* (germline KO) or *Pclaf-fl/fl* (for conditional KO [cKO]) strains. Then, bleomycin was instilled into the lung, followed by

**Fig. 1 | Pclaf⁺ cells are elevated during lung regeneration. A–C** A public scRNA-seq dataset (GSE141259) was generated by subjecting sorted cells from the mouse lung epithelial compartment at 18 different time points after bleomycin instillation. Uniform manifold approximation and projection (UMAP)-embedding displays cells colored by cell type identity (**A**). Feature plot of *Pclaf* expression (**B**). Dot plots showing *Pclaf*, *Mki67*, and *Top2a* gene expression in each cell type (**C**). **D–F** A public scRNA-seq dataset (GSE135893) was generated from the human lung cells of 20 pulmonary fibrotic diseases and 10 control lungs. UMAP embedding displays cells colored by cell type identity (**D**). Feature plot of expression of *PCLAF* (**E**). Dot plots for *PCLAF*, *MKI67*, and *TOP2A* gene expression in each cell type (**F**). **G** Feature plots of *Pclaf* expression in mouse lung at the indicated time points using data shown in (**A**). **H–J** Mouse lungs (*n* = 3) were collected after bleomycin (1.4 U/kg) had been added to the trachea at the indicated time points. RT-qPCR analysis of the *Pclaf*, *Tnfα*, *Il1α*, *Il6*, and *Cox2* mRNA level (**H**) and the ratio of *Bax*/*Bcl2* mRNA level (**I**). Quantification of Pclaf⁺ cells by immunostaining (**J**). **K** Scheme of establishing *Pclaf-lacZ* knock-in mice. The neo cassette was deleted by breeding *Pclaf-lacZ-neo* with Rosa26-Cre driver. **L, M** Representative images of X-gal staining. *Pclaf-lacZ* mice were instilled with bleomycin (1.4 U/kg; *n* = 10; Bleo) or PBS (0 dpi; *n* = 10). At 7 dpi, lungs were collected and stained for X-gal (5-bromo-4-chloro-3-indolyl-β-D-galactopyranoside; blue) and nuclear fast red (pink) (**L**). Quantification of lacZ⁺ cells (**M**). Two-sided Student's *t*-test; error bars: mean +/− standard deviation (SD). Represented data are shown (*n* ≥ 3). Source data are provided as a Source Data file. Graphic icons were created with BioRender.com.

tamoxifen administration for genetic labeling and lineage tracing of cells derived from AT2 cells (Sftpc-Cre-driven GFP+ cells). Consistent with cell lineage trajectory inference and immunostaining results (Fig. 3C–F, Supplementary Fig. 8), AT1 cells (HOPX and GFP double-positive) derived from AT2 cells were significantly reduced and AT2 cells (SPC and GFP double-positive) were elevated in both *Pclaf* KO and cKO lung tissues compared to *Pclaf* WT (Fig. 3G, H). In addition, KRT8 and GFP double-positive cells were reduced in both *Pclaf* KO and cKO lung (Fig. 3G, H). In line with these results, the CytoTRACE analysis[35] showed a relatively less differentiated cell status in activated AT2 cells of *Pclaf* KO lungs than of WT lungs, indicating the impaired maturation of these cells into AT1 cells in the condition of *Pclaf* ablation (Fig. 3I). These results suggest that PCLAF plays a pivotal role in alveolar cell lineage plasticity that induces AT1 cell regeneration from AT2 cells during lung regeneration.

### PCLAF-DREAM axis is required for AT1 cell differentiation

Next, we elucidated the underlying mechanism of PCLAF-controlled AT2 cell plasticity. PCLAF remodels the repressive DREAM complex to activate the target gene transcription[20]. Consistent with the specific expression of *PCLAF*/*Pclaf* in PAPCs, the expression of DREAM target genes was exclusively enriched in the PAPCs of human and mouse lung tissues (Fig. 4A, B, and Supplementary Fig. 11A). The gene set enrichment analysis (GSEA) showed enrichment of DREAM target genes in *Pclaf* WT lung tissue (vs. KO lung, Fig. 4C and Supplementary Fig. 11B) confirmed by the module score of DREAM target genes (Fig. 4D), implying that the DREAM complex might be functionally associated with PCLAF-controlled AT2 cell plasticity. To test this, we pharmacologically and genetically manipulated the DREAM complex in the LOs isolated from the mice treated with bleomycin. First, we used harmine, an inhibitor of DYRK1A that suppresses the DREAM complex[36,37]. DYRK1A-phosphorylated serine residue (Ser28) of LIN52, a core subunit of the DREAM/MuvB complex, is essential for recruiting RBL2/p130, E2F4, and DP1 to the MuvB core complex to form a repressive DREAM complex, which subsequently suppresses gene transactivation. Thus, harmine-inhibited DYRK1A constitutively activates DREAM target gene expression. Harmine treatment rescued OFE inhibited by *Pclaf* KO (Fig. 4E–G). Harmine also restored the lung regeneration impaired by *Pclaf* KO, with increased AT1 and decreased AT2 cells (Fig. 4H–J). Moreover, the ectopic expression of the Lin52-S28A mutant, which is no longer phosphorylated by DYRK1A and consequently activates the DREAM target gene expression, rescued the *Pclaf* KO phenotype by AT1 cell repopulation, restored the number of AT2 cells, and increased OFE (Fig. 4K–N, Supplementary Fig. 12). WNT signaling maintains AT2 cell stemness[38]. Although PCLAF hyperactivates WNT/β-catenin in the intestine[16,19], β-catenin target genes (*Cd44*, *Ccnd1*, and *Bmp4*) were marginally downregulated in *Pclaf* KO PAPCs. In addition, the Sox9-based progenitor gene signature, which positively regulates lung stem/progenitor cell plasticity[39], and *Myc* transactivated by Pclaf in the intestine[19] were not downregulated in *Pclaf* KO PAPCs (Supplementary Fig. 13). These results suggest that the PCLAF-DREAM axis−controlled transcriptome mediates alveolar cell lineage plasticity for lung regeneration.

### CLIC4-TGFβ axis is required for AT1 cell differentiation

Among the differentially expressed genes (DEGs; *Pclaf* WT vs. KO) of the DREAM target genes, *Clic4* showed reduced expression in transitioning cells (PAPC, Krt8⁺, and AT2^med/AT1^med cells (Supplementary Fig. 14A), validated by immunostaining (Fig. 5A, B). *CLIC4* has been predicted as a direct target gene of the DREAM complex[40] and PCLAF[20], and various DREAM components were recognized to bind the proximal promoter of *CLIC4* (Supplementary Fig. 14B). Thus, we tested whether PCLAF directly transactivates *CLIC4* by chromatin immunoprecipitation (ChIP) assay. Indeed, PCLAF bound to the proximal promoter of *CLIC4* (Fig. 5C and Supplementary Fig. 14C). CLIC4 is an integral component of TGF-β signaling, required to transdifferentiate AT2 cells into AT1 cells during lung regeneration[41,42]. CLIC4 positively modulates TGF-β signaling by preventing the dephosphorylation of phospho-SMAD2/3 in the nucleus[43,44]. Given the crucial role of TGF-β/SMAD signaling in lung regeneration[41,42], we hypothesized that the PCLAF-DREAM-activated CLIC4-TGF-β signaling axis is required for AT2 cell plasticity into AT1 cells. Indeed, SMAD3 target genes from three datasets[45,46] were downregulated in *Pclaf* KO PAPCs but not in other cell types compared to WT (Fig. 5D). In addition, a GSEA showed the enrichment of Smad2 or Smad3 target gene expression in *Pclaf* WT PAPCs compared to in *Pclaf* KO PAPCs (Supplementary Figs. 11B, 14D). Moreover, a SCENIC gene network analysis showed reduced Smad3 regulon activity in *Pclaf* KO PAPCs and Krt8⁺ cells (Supplementary Fig. 14E), consistent with the reduced p-SMAD3 in the lung of *Pclaf* KO mice treated with bleomycin, compared to the *Pclaf* WT regenerating lung (Fig. 5E, F). Conversely, harmine treatment restored Clic4 and p-SMAD3 expression downregulated by *Pclaf* KO (Supplementary Fig. 15A–D). Next, to determine the role of the CLIC4-TGF-β signaling axis in AT2 cell plasticity, we ectopically expressed Clic4 in *Pclaf* KO LOs. Clic4 expression increased phosphorylation of SMAD3 (Supplementary Fig. 15E, F) and rescued the AT1 differentiation blocked by *Pclaf* KO (Fig. 5G–I and Supplementary Fig. 15G), as did temporarily controlled TGF-β treatment (Fig. 5J–M and Supplementary Fig. 16). Notably, human PAPCs from idiopathic pulmonary fibrosis (IPF) showed reduced CLIC4 and SMAD3-target genes expression compared to PAPCs from normal lungs (Fig. 5N, O). TGF-β signaling also contributes to lung fibrosis[47]. While regenerating alveoli lesions showed decreased p-SMAD3 by *Pclaf* KO, the inflamed lesions exhibited elevated p-SMAD3 (Supplementary Fig. 17). In addition, human IPF fibroblasts showed increased SMAD3-target gene expression compared to normal lung fibroblasts (Supplementary Fig. 18). These results suggest that the PCLAF-DREAM-activated CLIC4-TGF-β axis is required for AT2 cell plasticity in lung regeneration.

### PCLAF signature mimicry reduces lung fibrosis

For the pharmacological application of PCLAF-DREAM-controlled lung regeneration, we exploited the perturbational datasets (L1000) using the CLUE platform[48], identifying drug candidates that mimic

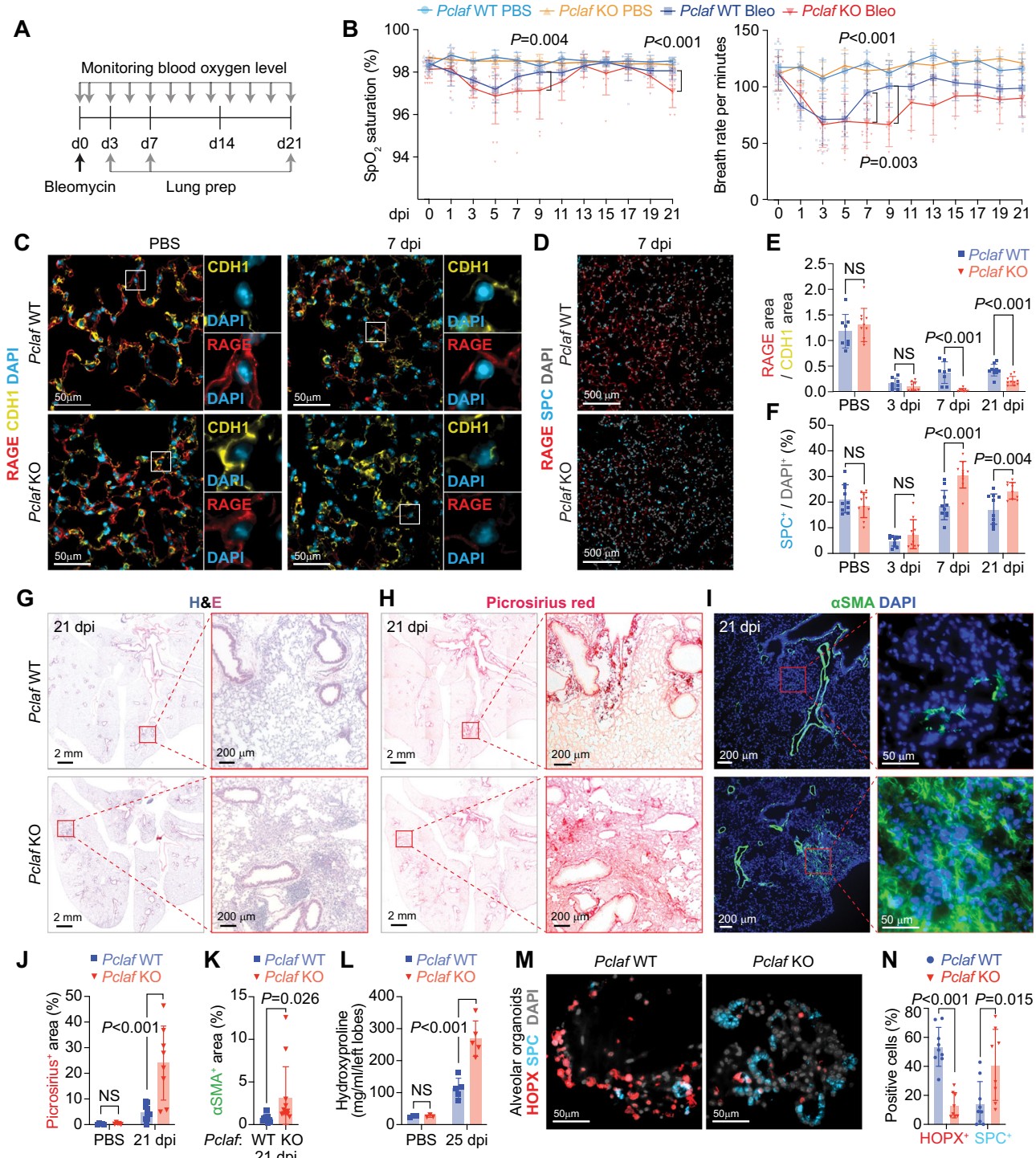

PCLAF-activated transcriptional signatures. Since PCLAF also promotes lung tumorigenesis[20], to minimize the risk of tumorigenesis by drug candidates, we excluded any drugs that were identified from the *Pclaf/PCLAF* depleted gene sets of the mouse lung tumors (*Kras^{G12D}/Trp53^{-/-}*; KP) and human (H1792) lung cancer cell lines (Fig. 6A, B, and Supplementary Data 4 and 5)[20], which identified four drugs (phenelzine, trimethobenzamide, ibuprofen, and RS 23597-190) (Fig. 6B). Among those, phenelzine, an FDA-approved anti-depressor, increased OFE (Supplementary Fig. 19A, B) and AT1 cells in LOs (Fig. 6C, D). In addition, phenelzine altered the cell plasticity of bronchiolar cells (Fig. 6E, F), which partly contributes to alveolar regeneration[39], resulting in the reduced numbers of ciliated and elevated club cells in

LOs (Fig. 6E, F). Next, we tested the impact of phenelzine on lung regeneration in mice given a high dosage of bleomycin (2.7 U/kg) (Fig. 6G), a concentration of developing lung fibrosis at 21 dpi (Supplementary Fig. 19C, D). Phenelzine ameliorated bleomycin-decreased SpO₂ levels (Fig. 6H) and markedly reduced lung fibrosis with a restored AT1 cell repopulation (Fig. 6I–N). These data indicate the therapeutic potential of phenelzine, a PCLAF activation–mimicking drug, in lung regeneration and fibrosis.

## Discussion
Accumulating evidence suggests that AT2 cells act as progenitor cells that differentiate toward AT1 cells through intermediate cell states.

**Fig. 2 | *Pclaf* KO impairs lung regeneration. A** Experimental scheme for the bleomycin-induced lung injury model. *Pclaf* WT and KO mice were treated with PBS (*n* = 5 for *Pclaf* WT, *n* = 5 for *Pclaf* KO) or bleomycin (1.4 U/kg; *n* = 25 for *Pclaf* WT, *n* = 24 for *Pclaf* KO; Bleo) by intratracheal instillation. The blood oxygen level (SpO$_2$) and breath rate were measured by pulse-oximetry at the indicated time points. At 3 days after injury (dpi, *n* = 7 for each group with bleomycin), 7 dpi (*n* = 8 for each group with bleomycin), and 21 dpi (*n* = 10 for *Pclaf* WT, *n* = 8 for *Pclaf* KO with bleomycin, *n* = 5 for each group with PBS (21 dpi)), the lungs were collected for further analysis. **B** The dynamics of SpO$_2$ levels (*left* panel) and breath rate per minute (*right* panel) were measured at the indicated time points. *Pclaf* WT and KO with PBS (*n* = 5); *Pclaf* WT with bleomycin until 3 dpi (*n* = 25); 5 dpi and 7 dpi (*n* = 18), after 9 dpi (*n* = 10); *Pclaf* KO with bleomycin until 3 dpi (*n* = 24); 5 dpi and 7 dpi (*n* = 17), 9–15 dpi (*n* = 9), after 17 dpi (*n* = 8). **C** Representative immunostaining images for RAGE (AT1) and CDH1 (Epithelial cell) in PBS-instilled lung and 7 dpi of bleomycin-instilled lung. **D** Representative images of immunostaining for RAGE and SPC (AT2 cells) at 7 dpi. **E** Quantification of RAGE$^+$ area/CDH1$^+$ area. *Pclaf* WT with PBS (*n* = 8), 3 dpi (*n* = 8), 7 dpi (*n* = 8), and 21 dpi (*n* = 10); *Pclaf* KO with PBS

(*n* = 10), 3 dpi (*n* = 8), 7 dpi (*n* = 9), and 21 dpi (*n* = 10). **F** Quantification of SPC$^+$ cells. *n* = 10 for each strain at indicated time point. **G** Representative images of hematoxylin and eosin (H&E) staining at 21 dpi. **H** Representative images of picrosirius staining (collagen fiber) at 21 dpi. **I** Representative images of immune-staining for alpha-smooth muscle actin (αSMA/Acta2; smooth muscle cell) at 21 dpi. **J** Quantification graphs of the picrosirius$^+$ area. *Pclaf* WT or *Pclaf* KO with PBS (*n* = 5) or bleomycin at 21 dpi (*n* = 10). **K** Quantification graphs of the αSMA$^+$ area. *n* = 12 for each strain. **L** Quantification of hydroxyproline contents in the left lobe of bleomycin (*n* = 5) or PBS-instilled (*n* = 3) lung at 25 dpi. **M, N** The lung epithelial cells were isolated from bleomycin-treated lungs of *Pclaf* WT or *Pclaf* KO mice at 7 dpi by MACS. The lung epithelial cells (TER119$^-$/CD31$^-$/CD45$^-$/EPCAM$^+$) were cultured with lung endothelial cells (CD31$^+$) at a liquid-air interface to generate LOs. Representative Images of alveolar type organoids fluorescently immunostained for HOPX (AT1) and SPC (**M**). Quantification graph of HOPX$^+$ and SPC$^+$ cells. *Pclaf* WT (*n* = 9) or *Pclaf* KO (*n* = 8) (**N**). Two-sided Student's *t*-test; error bars: mean +/− SD. Represented images are shown (*n* ≥ 3). Source data are provided as a Source data file.

Such transient cells express low levels of AT1 and AT2 markers and their specific genes, such as *Krt8*[26,30,32]. Several developmental signaling pathways (BMP[49], NOTCH[50], and YAP/TAZ[51]) contribute to cell transition from AT2 to AT1 during regeneration. However, the detailed mechanism of AT2 cell lineage plasticity during regeneration remains elusive. Our study identified a crucial role of the PCLAF-DREAM axis in cell plasticity (from AT2 to AT1). The primary function of the DREAM complex in orchestrating the cell cycle and cell quiescence by transactivating or repressing gene expression has mainly been investigated using in vitro systems[24,25,40,52–56]. Recently, we found that the dysregulated DREAM complex promotes lung tumorigenesis in vivo[20]. Genome-wide analyses showed that most DREAM targets are cell cycle-related genes[40,52] and genes involved in other biological processes (Supplementary Data 6). Given that the DREAM complex coordinates the cell cycle[24,25,40,52–55], we initially hypothesized that the PCLAF-DREAM axis induces mitotic activation of lung progenitor cells for regeneration. Indeed, the expression of cell cycle-related genes (*Top2a* and *Mki67*) was decreased in *Pclaf* KO PAPCs (Supplementary Fig. 14A) with reduced OFE (Supplementary Fig. 5A, B). However, *Pclaf* KO lung tissues and organoids showed a higher number of PAPCs compared to *Pclaf* WT (Supplementary Fig. 3B, C and Supplementary Fig. 5F, G), which might be due to stalled transitioning from PAPCs to Krt8$^+$ cells. *Pclaf* WT PAPCs exhibited the cellular trajectories into AT2 and AT1 cells during normal regeneration. However, *Pclaf* KO PAPCs failed to differentiate into AT1 cells (Fig. 3C–E and Supplementary Fig. 8), which might increase the number of PAPCs in *Pclaf* KO mice. Indeed, cell lineage tracing experiments showed impaired regeneration of AT1 cells from AT2 cells by *Pclaf* genetic ablation (germline as well as conditional) (Fig. 3G, H and Supplementary Fig. 10). It is worth noting that the number of AT2 cells was elevated by *Pclaf* KO (Fig. 3), potentially due to their self-regeneration through PAPCs. Intriguingly, our data also showed that DREAM complex–mediated gene transactivation is indispensable for the differentiation into AT1 cell lineage (Fig. 4). CLIC4 transactivated by the PCLAF-DREAM complex positively modulates TGF-β signaling[43], driving AT1 differentiation (Fig. 5). Our results unveil an unexpected role of the DREAM complex in positively regulating cell plasticity and promoting tissue regeneration beyond its canonical role in controlling cell quiescence and proliferation.

Despite the recent single-cell transcriptomics of the lungs[26,27,32,39], the proliferating cell clusters have been less studied. Our results suggest that PAPCs are the transitioning cells during alveolar regeneration. Recently, Murthy et al. proposed that AT0 cells expressing *SFTPC* and *SCGB3A2* transiently differentiate into AT1 cells in human lungs[57]. These AT0 cells are proliferating cells expressing MKI67[57], consistent with our finding that PAPCs are intermediate transitioning cells generating AT1 cells.

Unexpectedly, *Pclaf* KO lungs exhibited increased MKI67+ PAPCs during regeneration compared to *Pclaf* WT (Supplementary Fig. 3B, C). Consistently, *Pclaf* KO LOs showed elevated PAPCs (Supplementary Fig. 5F, G). These results might be the consequences of inhibition of AT1 differentiation. Recent studies have shown the accumulation of intermediate cells along with the disrupted AT1 differentiation during lung regeneration[32,58]. On the other hand, during development or tissue regeneration, cell proliferation and cell differentiation are inversely correlated and often counteract each other[59]. Thus, it is also possible that increased cell proliferation of PAPCs by *Pclaf* KO might inhibit AT1 cell differentiation. More detailed mechanisms of how PAPCs are engaged in lung regeneration remain to be determined in future studies.

TGF-β signaling has been proposed as one of the driving factors for lung fibrosis[60] by promoting the epithelial-mesenchymal transition of AT2 cells[61]. Other studies suggest that TGF-β signaling is required for the alveolar cell plasticity[41] and alveolar regeneration[42]. Our results showed that TGF-β signaling activation is sufficient to rescue the *Pclaf* KO-suppressed AT1 cell generation in the context of PAPCs. In *Pclaf* KO lung, p-SMAD3 was significantly decreased in regenerative alveoli lesions (Fig. 5E, F) but elevated in inflamed fibrotic lesions (Supplementary Fig. 17). In line with this, SMAD3-target genes were downregulated in IPF PAPCs while upregulated in IPF fibroblasts compared to normal lungs (Fig. 5N, O, and Supplementary Fig. 18). Thus, it is highly likely that TGF-β signaling spatiotemporally contributes to lung fibrosis or regeneration, depending on cell types or lesions. For the mechanism of TGF-β signaling activation, we identified CLIC4, a positive regulator for TGF-β signaling[43], as one of the PCLAF-DREAM target genes[20,40]. Indeed, PCLAF directly occupies the *CLIC4* proximal promoter, and ectopic expression of CLIC4 markedly rescued p-SMAD3 levels in *Pclaf* KO LOs (Fig. 5C and Supplementary Figs. 14C, 15E, F).

The current therapeutic strategy for lung fibrosis has focused on inhibiting fibroblasts using pirfenidone and nintedanib[62]. Recent studies suggested that failure of lung regeneration is one of the fundamental mechanisms of lung fibrosis[7,9–13]. Intriguingly, disrupted AT2 cell lineage plasticity by *Pclaf* KO drove lung fibrosis rather than lung regeneration (Fig. 2). Based on our findings, we identified PCLAF-activation–mimicking drugs and tested their therapeutic potential on lung regeneration in vivo and in vitro (Fig. 6). This strategy, which facilitates the repopulation of functional lung epithelial cells, may be an alternative regimen or preventive or therapeutic measures for patients with potential lung fibrosis, such as those caused by thoracic radiotherapy.

Using *Pclaf-LacZ* mice, we observed that PCLAF was also expressed in non-epithelial cells such as immune cells and fibroblasts (Supplementary Fig. 2). *Pclaf* KO mice showed slight alteration of immune

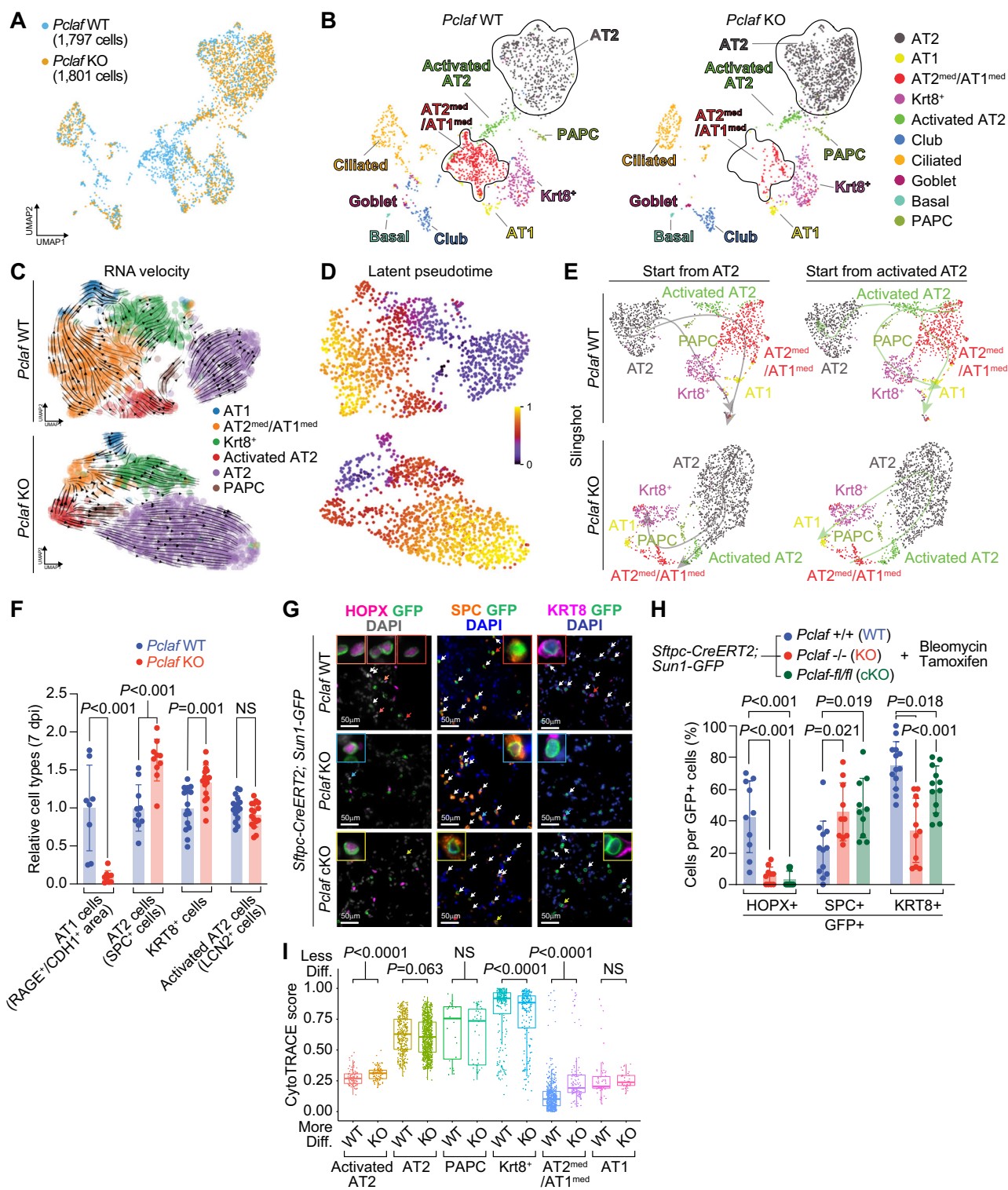

cell status[63]. Since immune cells[64] and fibroblasts[64,65] play an essential role in lung regeneration, the impact of *Pclaf* KO on these cells might also contribute to impaired lung regeneration. However, consistent with the *Pclaf* KO mouse and LO phenotypes, *Pclaf* cKO in AT2 cells is sufficient to inhibit the regeneration of AT1 cells from AT2 cells (Fig. 3G, H, Supplementary Fig. 10). Therefore, it is highly likely that the PCLAF-DREAM axis-driven lung epithelial cell plasticity is independent of immune cells and fibroblasts. Nonetheless, our finding in murine systems remains to be tested in a human-relevant system, such as human lung organoids.

Together, our comprehensive approaches reveal an unexpected role of the PCLAF-DREAM axis in driving alveolar cell plasticity via CLIC4-TGF-β for lung regeneration. Furthermore, this study suggests that phenelzine, a drug mimicking the PCLAF-DREAM transcriptional signature, may be a viable therapeutic option for lung disease.

## Methods

The research reported here research complies with all relevant ethical regulations. All mouse experiments were approved by the MD Anderson Institutional Animal Care and Use Committee and

**Fig. 3 | *Pclaf* KO suppresses AT2 cell lineage plasticity. A** Integrated UMAP displaying each cell cluster of pulmonary epithelial cells isolated from *Pclaf* WT and KO mice. **B** UMAPs (split by *Pclaf* WT and KO) displaying each cell cluster, colored by cell types. **C**–**E** Alveolar cell clusters (AT1, AT2$^{med}$/AT1$^{med}$, Krt8$^+$, Activated AT2, AT2, and PAPCs) were introduced for RNA velocity analysis. RNA velocity analyses (**C**), latent pseudotime analysis (**D**), and Slingshot analysis (**E**) of *Pclaf* WT or KO based on scRNA-seq are depicted. **F** Quantification of cells derived from AT2 cells immunostaining for RAGE$^+$ area/CDH1$^+$ area (AT1; $n = 8$ for *Pclaf* WT and $n = 9$ for *Pclaf* KO), SPC$^+$ cells (AT2; $n = 10$), KRT8$^+$ cells (Krt8$^+$; $n = 16$), and LCN2$^+$ cells (Activated AT2; $n = 16$) from the lung tissues (7 dpi). Two-sided Student's *t*-test; error bars: mean +/− SD. **G, H** Quantification of cells derived from AT2 cells using AT2 cell lineage-tracing animal model. The lung tissues of *Sftpc-CreERT2*; *Sun1-GFP* lineage-tracing mice combined with *Pclaf* WT ($n = 5$), *Pclaf* KO ($n = 3$), or *Pclaf* cKO (*Pclaf-fl/fl*; $n = 3$) (14 dpi with tamoxifen for five days) were analyzed by immunostaining (**G**) and quantification (**H**). Each cell type was detected by co-immunostaining with an anti-GFP antibody and calculated by their cell numbers per GFP+ cells. HOPX+ ($n = 10$); SPC+ for *Pclaf* WT ($n = 12$), *Pclaf* KO ($n = 10$), or *Pclaf* cKO ($n = 10$); KRT8+ for *Pclaf* WT ($n = 13$), *Pclaf* KO ($n = 11$), or *Pclaf* cKO ($n = 12$). Sun1-GFP is localized in the inner nuclear membrane. Two-sided Student's *t*-test; error bars: mean +/− SD. **I** Boxplots of predicted cluster-differentiation based on the CytoTRACE analysis using scRNA-seq data shown in (**B**). The edges of the box plot represent the first and third quartiles, the center line represents the median, and the whiskers extend to the smallest and largest data points within 1.5 interquartile ranges from the edges. Statistical significance is determined using an unpaired Wilcox test. Source data are provided as a Source data file.

performed under MD Anderson guidelines and the Association for Assessment and Accreditation of Laboratory Animal Care international standards.

## Mice
*Pclaf* KO mice from a previously established model[19] were used. *Pclaf-lacZ* knock-in strain was generated by breeding *Pclaf-lacZ-neo* mice with *CMV-Cre* (B6.C-Tg[CMV-Cre] 1Cgn/J; Jackson Laboratory; RRID:IMSR_JAX:006054) driver mice. The *Pclaf-lacZ-neo* strain was established by in vitro fertilization of the sperm carrying the *Pclaf-lacZ-neo* targeted allele (C57BL/6N-A$^{tm1Brd}$ Pclaf$^{tm1a(EUCOMM)Wtsi}$/WtsiPh; EMMA ID: EM:09820). Genomic DNA PCR and Sanger sequencing validated the targeted or recombinant alleles. *Pclaf-fl/fl* mice strain was established by breeding *Pclaf-lacZ-neo* mice with *Rosa26-Flp1* (B6.129S4-*Gt(ROSA) 26Sor$^{tm1(FLP1)Dym}$*/RainJ; Jackson Laboratory; RRID: IMSR_JAX:009086) driver mice. *Sftpc$^{CreERT2}$*; *Rosa26$^{Sun1GFP}$*; *Pclaf$^{/-}$* or *Sftpc$^{CreERT2}$*; *Rosa26$^{Sun1GFP}$*; *Pclaf-fl/fl* mice were generated by breeding *Pclaf* KO or *Pclaf-fl/fl* mice with *Sftpc$^{CreERT2}$*; *Rosa26$^{Sun1GFP}$* mice, respectively. *Sftpc$^{CreERT2}$*; *Rosa26$^{Sun1GFP}$* mice were kindly provided by Dr. Jichao Chen[66]. All animal experiments were performed on 8- to 10-week-old male mice.

## Bleomycin-induced lung injury
Eight- to ten-week-old mice were anesthetized via inhalation of isoflurane using a calibrated vaporizer for ~3 min. The mice were positioned on the intratracheal intubation stand and intubated IV catheters (22GA × 1.00 IN, 0.9 × 25 mm; BD, NJ) in the trachea using BioLite (Biotex, TX) intubation system with optic fiber. The intubated mice were given 1.25 U/kg of bleomycin in 50 µl of phosphate-buffered saline (PBS). One day before the injury, the blood arterial oxygen saturation and breath rate per minute were recorded using a small rodent oximeter sensor mounted on the thigh of each tested mouse (MouseOX; STARR Life Sciences, Oakmont, PA) every other day. Data were collected for a minimum of 10 seconds with no error code. The mice were euthanized at the acute injury phase (3 dpi, regeneration phase [7 dpi], and chronic fibrotic phase [21 dpi]) for post-mortem lung tissue collection, followed by histology.

## Histology and immunohistochemistry
Lung tissues were perfused with cold PBS (pH 7.4) into the right ventricle, fixed with 10% formalin, embedded in paraffin, and sectioned at 5-µm thickness. The sections were stained with hematoxylin and eosin for histological analysis. For picrosirius red staining, deparaffinized and hydrated sections were stained with picrosirius red solution (0.1% Direct Red 80 [Sigma] in saturated [1.3%] picric acid) for 1 h. Sections were washed with 0.5% acetic acid and dehydrated before mounting. For the immunohistochemistry analysis, sections were immunostained according to standard protocols[20]. For antigen retrieval, sections were subjected to heat-induced epitope retrieval pre-treatment at 120 °C using citrate-based antigen unmasking solution (Vector Laboratories, Burlingame, CA, USA), tris-based antigen unmasking solution (Vector Laboratories), universal antigen retrieval reagent (R&D System), or basic antigen retrieval reagent (R&D System). Specific information on antibodies is described in Supplementary Data 7.

## Cell lineage tracing assay
*Sftpc$^{CreERT2}$*; *Rosa26$^{Sun1GFP}$*, *Sftpc$^{CreERT2}$*; *Rosa26$^{Sun1GFP}$*, *Pclaf$^{/-}$*, and *Sftpc$^{CreERT2}$*; *Rosa26$^{Sun1GFP}$*; *Pclaf-fl/fl* mice at age of 10 weeks were subjected to anesthesia and intratracheal instillation of bleomycin at a dose of 1.25 U/kg. Subsequently, one day after bleomycin administration, tamoxifen (75 mg/kg, Sigma, dissolved in corn oil) was intraperitoneally injected into the mice for 5 consecutive days. On day 14 post-bleomycin instillation, the mice were euthanized, and lung tissues were collected for post-mortem histological analysis.

## Hydroxyproline assay
Eight- to ten-week-old *Pclaf* WT or Pclaf KO mice were instilled 1.25 U/kg of bleomycin in 50 µl of PBS or PBS by intratracheal ($n = 5$ each group). The mice were euthanized at 25 dpi for post-mortem lung tissue collection, followed by hydroxyproline assay. The hydroxyproline assay kit (Sigma) was used according to the manufacturer's instructions.

## X-gal staining
Lung tissues were perfused with cold PBS (pH 7.4) into the right ventricle, incubated with a mixture of 30% sucrose and 10% optimal cutting temperature (OCT) compound in PBS for 30 min at 4 °C, embedded in OCT compound, and sectioned at 10-µm thickness. Frozen lung sections were fixed with 0.2% glutaraldehyde for 10 min and washed in wash buffer (0.02% IGEPAL CA-630, 0.01% sodium deoxycholate, 2 mM MgCl$_2$, 0.1 M phosphate buffer pH 7.5) for 10 min twice. Then, slides were incubated with X-gal staining solution (1 mg/ml X-gal, 5 mM potassium ferricyanide, 5 mM potassium ferrocyanide in wash buffer) for 16 h in 37 °C. X-gal-stained slides were post-fixed with 4% PFA for 10 min and washed with PBS for 5 min 3 times. For counterstaining, slides were incubated with nuclear fast red solution (Vector Laboratories, Burlingame, CA, USA) for 2 min.

## Lung cell isolation
Lungs were harvested from euthanized mice after perfusing 10 ml of cold PBS into the right ventricle. Lungs were minced after the removal of extra-pulmonary tissues and digested in Leibovitz media (Gibco, USA, no. 21083-027) with 2 mg/ml collagenase type I (Worthington, CLS-1, LS004197), 2 mg/ml elastase (Worthington, ESL, LS002294), and 0.4 mg/ml DNase I (Sigma, DN-25) for 45 min at 37 °C. To stop the digestion, fetal bovine serum (FBS, HyClone; Cytiva) was added to a final concentration of 20%. The digested tissues were sequentially filtered through a 70-µm and a 40-µm cell strainer (Falcon, 352350 and 352340, respectively). The samples were incubated with 1 ml of red blood cell lysis buffer (15 mM NH$_4$Cl, 12 mM NaHCO$_3$, 0.1 mM EDTA, pH 8.0) for 2 min on ice. Leibovitz with 10% FBS and 1 mM EDTA was used for resuspension and washing for magnetic-activated cell sorting (MACS) or fluorescence-activated cell sorting.

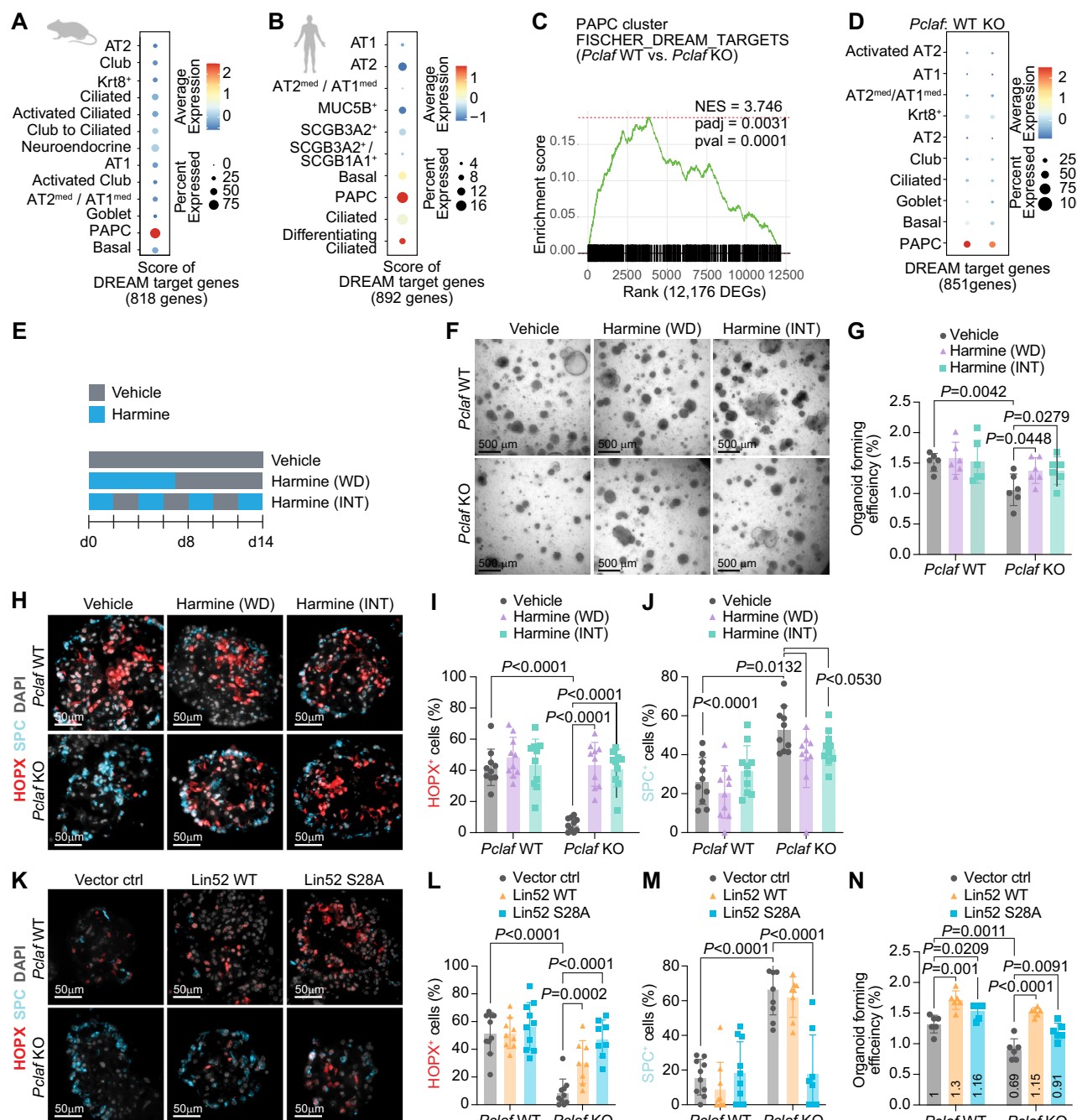

**Fig. 4 | PCLAF-DREAM axis mediates AT2 cell lineage plasticity for lung regeneration. A** Dot plots depicting transcriptional module scores of the gene sets of DREAM-target genes with the mouse scRNA-seq dataset shown in Fig. 1A. **B** Dot plots depicting transcriptional module scores of the gene sets of DREAM-target genes with the human scRNA-seq dataset shown in Fig. 1D. **C** Gene set enrichment analysis (GSEA) of *Pclaf* WT vs. *Pclaf* KO in the PAPC cluster using the dataset shown in Fig. 3. The enrichment plot presents the gene sets of DREAM-target genes. **D** Dot plots depicting transcriptional module scores of the DREAM-target gene set using the dataset shown in Fig. 3. **E** Experimental scheme for LO culture under stimuli of harmine (200 nM). Harmine was used to treat LOs for the first 7 days and withdrawn (WD). Alternatively, LOs were cultured with harmine intermittently (INT) at the indicated time points. **F** The representative bright-field z-stack images of LOs on day 12. **G** Quantification graph of lung OFE (*n* = 6). **H** Representative images of IF staining for HOPX and SPC on day 14. **I, J** Quantification graph of HOPX+ (*n* = 10) (**I**) and SPC+ cells (*n* = 10) (**J**). **K–N** Isolated lung epithelial cells were transduced with RFP, Lin52 WT, or Lin52 S28A by lentivirus and then cultured with LO. Representative images of IF staining for HOPX and SPC on day 14 (**K**). Quantification graph of HOPX+ (*n* = 10) (**L**) and SPC+ cells (*n* = 10) (**M**). Quantification graph of lung OFE (*n* = 6) (**N**). Two-sided Student's *t*-test; error bars: mean +/− SD. Represented images and data are shown (*n* ≥ 3). Source data are provided as a Source data file. Graphic icons were created with BioRender.com.

For lung epithelial cell isolation, cells were resuspended in 400 μl of buffer with 30 μl of CD31 MicroBeads (130-097-418; Miltenyi Biotec, Bergisch Gladbach, Germany), 30 μl of CD45 MicroBeads (130-052-301; Miltenyi Biotec), and 30 μl of anti-Ter-119 MicroBeads (130-049-901; Miltenyi Biotec) and incubated for 30 min at 4 °C, followed by negative selection according to the manufacturer's instructions. Cells were then resuspended with 400 μl of buffer with 30 μl of CD326 (EPCAM) MicroBeads (130-105-958; Miltenyi Biotec) and incubated for 30 min at 4 °C, followed by positive selection according to the manufacturer's instructions. Isolated lung

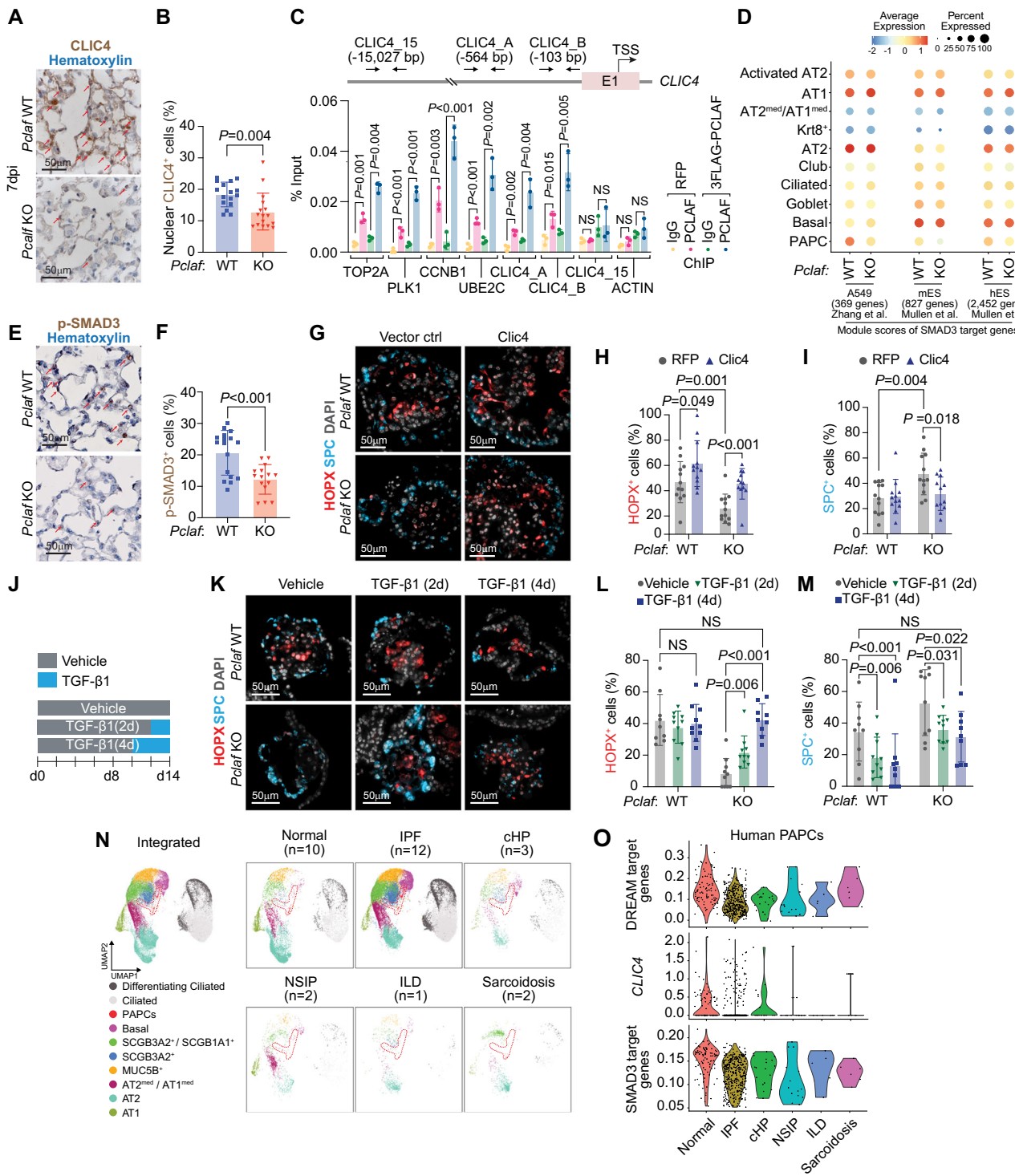

**Fig. 5 | PCLAF-DREAM-mediated CLIC4-TGF-β signaling axis is required for AT1 cell differentiation. A** Representative images of *Pclaf* WT and *Pclaf* KO lung at 7 dpi, immunostained for CLIC4. **B** Quantification graph of nucleus CLIC4⁺ cells (*n* = 16). **C** qPCR analysis using indicated primer sets targeting proximal promoter of DREAM target genes, *CLIC4*, or *ACTB*. ChIP was performed using anti-PCLAF antibody (*n* = 3). H358 cells ectopically expressing 3FLAG-PCLAF or RFP were used for ChIP. **D** Dot plots depicting transcriptional module scores of SMAD3 gene sets from A549, mouse embryonic stem cells (mES), and human embryonic stem cells (hES) using the scRNA-seq dataset shown in Fig. 3. **E** Representative images of *Pclaf* WT and *Pclaf* KO lung at 7 dpi, immunostained for p-SMAD3. **F** Quantification graph of p-SMAD3⁺ cells. *Pclaf* WT (*n* = 17) or *Pclaf* KO (*n* = 15). **G–I** Isolated lung epithelial cells were transduced with RFP- or CLIC4-expressing lentiviruses and cultured with LO. Representative images of IF staining for HOPX and SPC on day 14 (**G**).

Quantification of HOPX⁺ (*n* = 12) (**H**) and SPC⁺ cells (*n* = 12) (**I**). **J–M** *Pclaf* WT, or KO lung epithelial cells-derived LOs were cultured with TGF-β1 (2 ng/ml) for 2 days (TGFβ1 [2D]) or 4 days (TGFβ1 [4D]). Experimental scheme for TGF-β1 treatment (**J**). Representative images of IF staining for HOPX and SPC on day 14 (**K**). Quantification graph of HOPX⁺ (*n* = 10) (**L**) and SPC⁺ cells (*n* = 10) (**M**). **N** UMAP plots displaying each cell cluster from the scRNA-seq datasets of normal human lung and fibrotic diseases shown in Fig. 1D. (IPF idiopathic pulmonary fibrosis, cHP chronic hypersensitivity pneumonitis, NSIP nonspecific interstitial pneumonia, ILD interstitial lung disease). **O** Violin plots showing the expression of module scores of DREAM-target genes, *CLIC4* expression, and hES-SMAD3-target genes in the PAPC clusters from the human scRNA-seq dataset. Two-sided Student's *t*-test; error bars: mean +/− SD. Represented images are shown (*n* ≥ 3). Source data are provided as a Source data file.

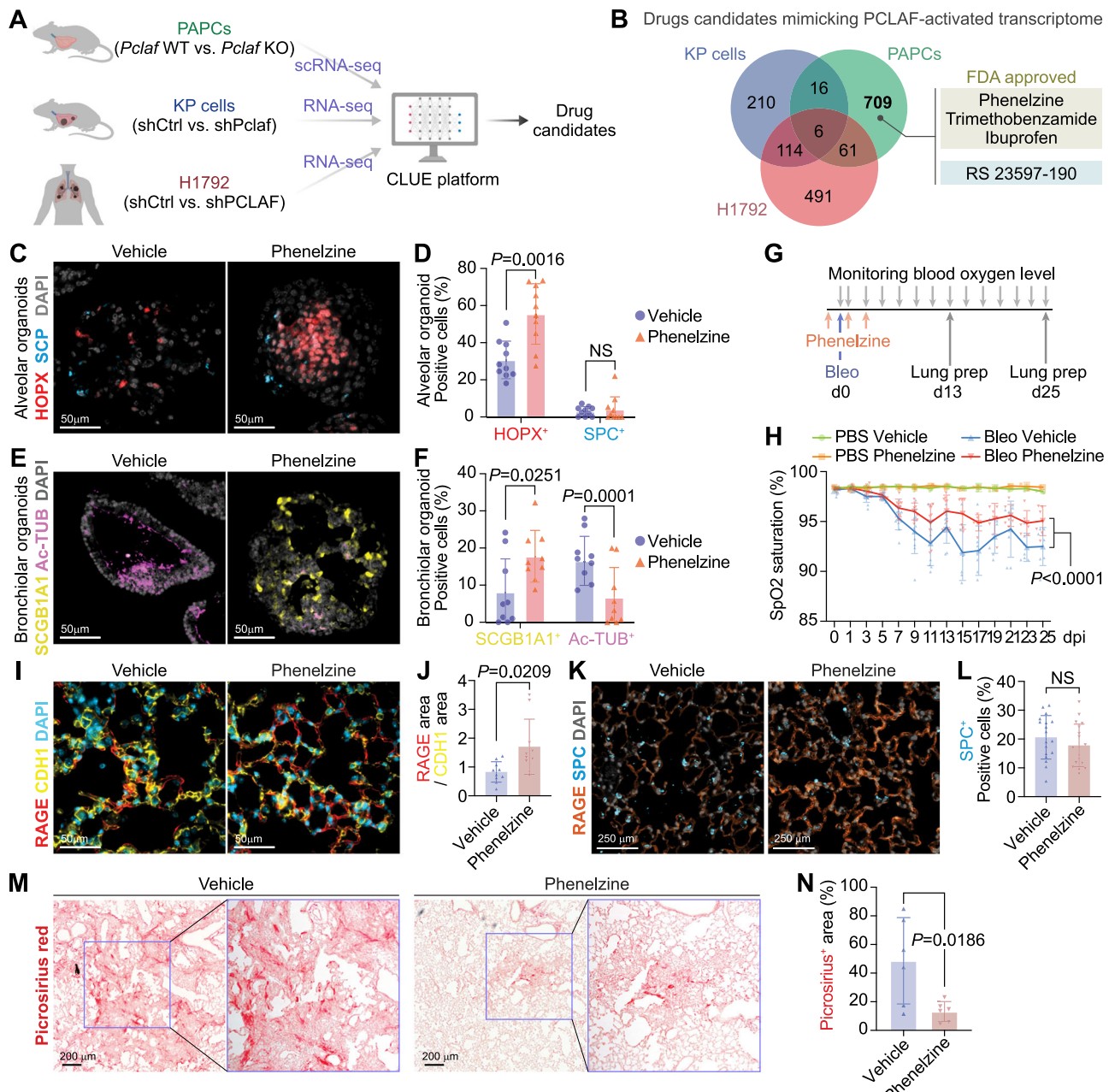

**Fig. 6 | Pharmacological mimicking of PCLAF-DREAM-activated transcriptional signature restores lung regeneration. A** Experimental scheme to identify drug candidates by the CLUE platform. **B** Venn diagram of Connectivity Map results, identifying drug candidates specific to normal lung PAPCs. **C** Representative images of LOs with phenelzine (10 μM) that fluorescently immunostained for HOPX and SPC. **D** Quantification graph of HOPX⁺ and SPC⁺ cells ($n = 10$). **E** Representative images of LOs with phenelzine (10 μM) that fluorescently immunostained for SCGB1A1 (Club) and Ac-TUB (Ciliated). **F** Quantification graph of SCGB1A1⁺ and Ac-TUB⁺ cells ($n = 9$). **G** Experimental scheme for bleomycin-induced lung regeneration by phenelzine. Mice were treated with bleomycin (2.8 U/kg) by intratracheal instillation. The vehicle control (DMSO, $n = 10$) or phenelzine ($n = 10$; 750 μg/head) were administered via intraperitoneal injection at −1, 1, and 3 dpi. The blood oxygen level (SpO₂) and breath rate were measured by pulse-oximetry every other day. At 13 dpi ($n = 4$ for each group) and 25 dpi ($n = 6$ for each group), lungs were collected

for further analysis. **H** The dynamics of spO₂ levels were measured at the indicated time points. Two-way ANOVA with post-hoc Tukey test. PBS with vehicle or phenelzine until 13 dpi ($n = 4$) and after 15 dpi ($n = 2$); bleomycin with vehicle until 13 dpi ($n = 10$) and after 15 dpi ($n = 7$); bleomycin with phenelzine until 7 dpi ($n = 10$), 9–13 dpi ($n = 9$), and after 15 dpi ($n = 6$). **I** Representative images of the lungs at 13 dpi, fluorescently immunostained for RAGE (AT1) and CDH1 (Epithelial cell). **J** Quantification graph of RAGE⁺/CDH1⁺ area ($n = 10$). **K** Representative images of the lungs at 13 dpi, fluorescently immunostained for RAGE (AT1) and SPC (AT2). **L** Quantification graph of SPC⁺ cells ($n = 18$). **M** Representative images of picrosirius staining at 25 dpi. **N** Quantification graphs of the picrosirius⁺ area ($n = 6$). Two-sided Student's $t$-test except for (**H**); error bars: mean +/− SD. Representative images are shown ($n \geq 3$). Source data are provided as a Source data file. Graphic icons in Figure A were created with BioRender.com.

epithelial cells were used for the lung organoid culture. In single-cell RNA sequencing (scRNA-seq), digested lung cells were resuspended in 400 μl of buffer with 5 μl of anti-CD31-FITC (BD Biosciences, CA, USA), 5 μl of anti-CD45-APC (BD Biosciences), and 5 μl of anti-CD326 (EPCAM)-PE-Cy7 (BD Biosciences) and incubated for 30 min at 4 °C.

Cells were then washed twice, followed by sorting of the epithelial cells by fluorescence-activated cell sorting.

For lung endothelial cell (LuEC) isolation, cells were resuspended in 400 μl of buffer with 30 μl of CD31 MicroBeads and incubated for 30 min at 4 °C, followed by positive selection according to the

manufacturer's instructions. Isolated LuECs were cultured with EC growth media (DMEM; Corning; 10-013-CV, 20% FBS, 1 × Penicillin-Streptomycin-Glutamine; Gibco, USA; 10378016, 100 µg/ml endothelial cell growth factor (ECGS); Sigma; E2759, 100 µg/ml heparin; Sigma; H3149, 25 mM HEPES) on 0.1% gelatin (Sigma, G1393)-coated plates. Cultured LuECs were then isolated with CD31 MicroBeads and expanded until passage 3. Expanded LuECs were cryopreserved for lung organoid culture.

## Lung organoid culture

Lung epithelial cells (Ter119[-]/Cd31[-]/Cd45[-]/Epcam[+]) isolated from 7- to 10-week-old *Pclaf* WT or *Pclaf* KO mice at 7 dpi of bleomycin were cultured with lung stromal cells in a 3D organoid air-liquid interface, as described previously[31]. In brief, freshly sorted lung epithelial cells were resuspended in 3D organoid media Dulbecco's modified Eagle's medium (DMEM)/F12 (Gibco, USA), 10% fetal bovine serum (FBS; HyClone, USA), 1 × penicillin-streptomycin-glutamine (Gibco, USA), and 1X insulin-transferrin-selenium (Sigma)) and mixed with LuECs at a ratio of 1:1. Cells containing 3D media were mixed with growth factor-reduced Matrigel (BD Biosciences) at a ratio of 1:1. The 100 µl of mixtures containing lung epithelial cells ($5 \times 10^3$) and LuECs ($5 \times 10^4$) were placed in the transwell insert (0.4-µm pore, Corning, Lowell, MA). After incubation for 30 min at 37 °C in an incubator, 500 µl of 3D media was placed in the bottom chamber to generate the liquid-air interface. Media were exchanged every other day, with or without harmine (200 nM; Abcam), phenelzine (10 µM; Cayman, MI, USA), or TGF-β1 (2 ng/ml; R&D System). For viral transduction, freshly isolated lung epithelial cells were resuspended in 500 µl of 3D media containing lentivirus ($2.5 \times 10^7$ TU/ml) and polybrene (7 µg/ml) and centrifuged for 1 h at $600 \times g$ at 32 °C. Cells were incubated at 37 °C in an incubator for 7 h, followed by organoid culture.

## Mouse embryonic fibroblast culture

Embryos (at embryonic day 13.5) were isolated from pregnant *Pclaf-fl/fl* mouse. After dissecting, minced embryos were incubated in trypsin/EDTA (Invitrogen) with 0.4 mg/ml DNase I (Sigma, DN-25) for 15 min at 37 °C. Then, digested embryos were cultured on 0.2% gelatin-coated culture plates. After 2 passages, $2 \times 10^5$ mouse embryonic fibroblasts (MEFs) were cultured in 6-well plates and introduced $1 \times 10^7$ or $1 \times 10^8$ PFU of adenoviral Cre (Ad-Cre; kindly provided by Dr. Kwon-Sik Park, The University of Virginia). After 2 days of Ad-Cre infection, cells were harvested and genotyped by PCR.

## Gene construct

pLenti-CMV-MSC-RFP-SV-Puro vector was purchased from Addgene (#109377) and used as a control vector or backbone for further cloning. For constructing the pLenti-3HA-Clic4-T2A-GFP plasmid, we prepared a pLenti vector backbone using restriction enzymes, XbaI (NEB, USA) and BamHI (NEB, USA). Three DNA fragments of Kozak-3HA, Clic4, and T2A-GFP were amplified with Phusion High-Fidelity DNA Polymerase (NEB, USA) using pLJC5-Tmem192-3HA (Addgene, #102930) for Kozak-3HA, pAltermax-3HA-Clic4[44] for Clic4, and pLenti-CMV-mCherry-T2A-GFP for T2A-GFP. We assembled the pLenti-3HA-Clic4-T2A-GFP vector with pLenti backbone and fragments using NEBuilder HiFi DNA Assembly Master Mix (NEB, USA) according to the manufacturer's instructions. Primer information is described in Supplementary Data 7. We used pLenti-3FLAG-Lin52 (WT or S28A) and pLenti-3FLAG-PCLAF, as previously generated[20].

## Lentivirus preparation

Lentiviruses were produced using 2nd-generation packaging vectors in 293T cells. 293T cells were cultured until 70–80% confluent, and the media were replaced with antibiotics-free DMEM (10% FBS). After 1 h of media exchange, cells were transfected with vector mixtures in Opti-MEM (Gibco, USA). To generate a vector mixture, pMD2.G (1.3 pmol),

psPAX2 (0.72 pmol), DNA (1.64 pmol), and polyethyleneimine (PEI, 39 µg) were added to 800 µl of Opti-MEM and incubated for 15 min. After 12 h of transfection, the media were exchanged with complete media (DMEM, 10% FBS, and 1X penicillin-streptomycin). The virus supernatant was collected after 24 and 48 h and filtered with a 0.45-µm syringe filter (Thermo Fisher, CA, USA); polyethylene glycol (PEG) solution (40% PEG-8000 [Thermo Fisher, CA, USA], and 1.2 M NaCl in pH 7.4 PBS) were added in a 3:1 ratio. After overnight incubation of the virus-PEG mixture at 4 °C, the virus pellet was generated by centrifugation at $1600 \times g$ and 4 °C for 1 h. The pellet was reconstituted in 1 ml of 3D media and incubated for 15 min at 4 °C. Reconstituted virus was centrifuged at $20,000 \times g$ and 4 °C for 3 min to remove protein debris. The collected virus was stored in a −80 °C freezer. pLenti-RFP, pLenti-3FLAG-Lin52WT, pLenti-3FLAG-Lin52S28A, pLenti-3HA-Clic4-T2A-GFP, and pLenti-3FLAG-PCLAF vectors were used for lentivirus generation. For virus titration, $5 \times 10^5$ 293T cells were seeded into 48-well plates (Thermo Fisher, CA, USA). After 24 h, cells were infected by the serially diluted virus with 10 µg/ml polybrene. After 48 h of transfection, virus transduction units per ml (TU/ml) were measured by fluorescence microscopy.

## Chromatin immunoprecipitation quantitative real-time PCR (ChIP-qPCR)

pLenti-3FLAG-PCLAF was introduced to H358, a human lung adenocarcinoma cell line, by lentiviral infection, followed by selection using puromycin and used for ChIP-qPCR assay. ChIP-qPCR assays were performed as previously described with minor modifications[20]. Briefly, cells were cross-linked with 1% formaldehyde for 15 min at room temperature. Formaldehyde was quenched by adding glycine (final concentration, 0.125 M). After the cells were washed with cold PBS, the cells were harvested with lysis buffer (20 mM Tris, pH 8.0, 85 mM KCl, 0.5% NP-40, freshly supplemented with a protease inhibitor mixture) and rocked at 4 °C for 15 min. Pellets containing chromatin were collected by centrifugation, resuspended in nuclear lysis buffer (50 mM Tris, pH 8.0, 10 mM EDTA, 1% SDS), and subjected to sonication (30 s on and 30 s off, 80 times, Bioruptor 300 [Diagenode]). The supernatant was collected by centrifugation, diluted 20 times in IP buffer (50 mM Tris, pH 8.0, 150 mM NaCl, 0.5% NP-40, and a protease inhibitor mixture), and subjected to IP with each antibody. The enrichment of 300–600 bp of sheared DNA was confirmed by gel electrophoresis. Approximately 20–30 µg of DNA was used for IP. Diluted protein-DNA complexes were precleared and incubated overnight with preblocked protein G Dynabeads and antibodies at 4 °C. Immunoprecipitants were washed serially with LiCl wash buffer (Tris 50 mM, pH 8.0, EDTA 1 mM, LiCl 250 mM, 1% NP-40, and 0.5% deoxycholate), IP buffer, and Tris-EDTA buffer. For DNA extraction, immunoprecipitants were reversely cross-linked by incubation at 65 °C overnight, and further incubated with RNase A and proteinase K. Then, ChIP DNAs were isolated using a PCR purification kit (QIAGEN). The following antibodies were used for ChIP: FLAG M1 (Sigma, 1:200), PCLAF (Abcam, 1:200). Previously validated DREAM binding sequences of CCNB1, TOP2A, PLK1, and UBE2C promoters, and predicted two proximal promoter regions of CLIC4 were analyzed by ChIP-qPCR. ACTIN promoter amplicons served as negative controls. Detailed information on antibodies and primers is in Supplementary Data 7.

## Library preparation and sequencing of single-cell transcriptome

Freshly isolated lung epithelial cells from *Pclaf* WT and *Pclaf* KO mice at 7 dpi of bleomycin instillation were used to prepare the single-cell RNA library. To generate single-cell gene expression libraries, $1.6 \times 10^4$ cells per group of single cells were subjected to the Chromium Single Cell Gene Expression 3v3.1 kit (10x Genomics) at the Single Cell Genomics Core at Baylor College of Medicine. In brief, reverse transcription (RT) reagents, gel beads containing barcoded oligonucleotides, and oil were loaded on a chromium controller (10x Genomics) to generate

single-cell gel beads-in-emulsions (GEMs) in which full-length cDNA was synthesized and barcoded for each single-cell. Subsequently, the GEMs were broken, and cDNA from every cell was pooled. Following clean-up using Dynabeads MyOne Silane Beads, cDNA was amplified by PCR. The amplified product is fragmented to an optimal size before end-repair, A-tailing, and adapter ligation. The final library was generated by amplification. Libraries were run on Novaseq for Illumina PE150 sequencing with 20,000 reads per cell. Post-processing and quality control were performed by Novogene using the 10× Cell Ranger package (v. 3.1.0, 10x Genomics). Reads were aligned to the GRCm38 reference assembly using STAR aligner[67] (v. 2.7.10a).

### Single-cell RNA preprocessing and quality control
The Cell Ranger output was used as the input to Seurat v. 4.1.0 for further analysis of the scRNA-seq samples. For each sample, poor-quality cells were filtered based on the number of features (min.-features = 200, min.cells = 3), number of RNAs detected (>200 and <5000), and percentage of reads arising from the mitochondrial genome (<20%). We combined *Pclaf* WT and *Pclaf* KO samples using the merge function of Seurat. Clustering using default parameters and uniform manifold approximation and projection (UMAP) non-linear dimensionality reduction were performed using Seurat v. 4.1.0, and we referred to the result as our integrated clusters. The marker genes of each cluster (differentially expressed genes [DEGs] in each cluster compared with all other clusters) and differentially expressed genes between samples were identified with Seurat using the FindAllMarkers function with the parameters logfc.threshold = 0.25 and min.pct = 0.25. For the lung epithelial cell analysis, we downsized the *Pclaf* KO sample by one-sixth (from 10,807 cells to 1797 cells).

### Gene set expression score analysis
The scores of gene sets were analyzed using the AddModuleScore function of Seurat. The SMAD3 target genes[45,46], DREAM-target genes[40], and SOX9-based progenitor-related genes[39] were used for module score analysis. A detailed list of genes for each module score is listed in Supplementary Data 8.

### Gene set enrichment analysis
The DEGs between *Pclaf* WT vs. *Pclaf* KO in the PAPC clusters were identified by the Wilcoxon sum test and AUROC statistics using the Presto package v. 1.0.0. They were then subjected to a gene set enrichment analysis (GSEA) using the fgsea package v. 1.16.0. The curated gene sets (C2) in the Molecular Signature Database (MsigDB) v. 7.5.1 were used for the GSEA using the msigdbr package.

### Pseudotime trajectory analysis
Velocyto.py was used to estimate the RNA velocities of single cells by distinguishing between unspliced and spliced mRNAs in the standard single-cell RNA-sequencing data[68]. The BAM files, output from Cell Ranger, were subjected to generate loom files containing three categories: spliced, unspliced, and ambiguous. The loom files were applied to calculate the dynamic velocity graph using scVelo v. 0.2.4[34].

### Developmental state analysis
CytoTRACE v. 0.3.3 was used to predict the relative differentiation state of a single cell[35]. The cells were given a CytoTRACE score according to their differentiation potential, with a higher score indicating higher stemness/fewer differential characteristics.

### Gene regulatory network analysis
For regulon identification, a gene regulatory network analysis was performed using the Python version of the SCENIC (pySCENIC) method (v. 0.11.2)[69]. pySCENIC integrates a random forest classifier (GENIE3) (v. 1.0.0) to identify potential TF targets based on their co-expression with RcisTarget (v. 0.99.0) for *cis*-regulatory motif enrichment analysis in the promoter of target genes (±500 bp of the transcription start site) and identify the regulon, which consists of a TF and its co-expressed target genes. The *Mus musculus* 10 (mm10) motif database provided by the pySCENIC authors was used. Finally, for each regulon, pySCENIC uses the AUCell (v. 0.99.5) algorithm to score the regulon activity in each cell (Z score, a standardized AUCell score, was used). All parameters used for running were specified in the original pySCENIC pipeline. The ranking databases used were mm10-refseq-r80-10 kb-up-and-down-tss.mc9nr.feather mm10-refseq-r80-500bp-up-and-100bp-down-tss.mc9nr.feather, which were downloaded from https://resources.aertslab.org/cistarget/. The motif database was downloaded from https://resources.aertslab.org/cistarget/motif2tf/.

### Public scRNA-seq data analysis
A public scRNA-seq dataset (GSE141259)[26], generated by subjecting sorted cells from the mouse lung epithelial compartment at 18 different time points after bleomycin instillation, was analyzed according to the code provided in the original study. A public scRNA-seq dataset (GSE135893)[27] was generated from the cells of 20 pulmonary fibrosis and 10 control human lungs and analyzed according to the code provided in the original study.

### Drug identification for mimicking PCLAF-mediated transcriptome
CLUE platform (https://clue.io/), containing L1000-based Connectivity Map[48], was used for drug identification. We identified drug candidates using the three gene lists: (1) elevated in *Pclaf* WT PAPCs (compared to *Pclaf* KO PAPCs), (2) downregulated genes in human LUAD cells, H1792 cells, by *shPCLAF* (compared to control shRNA), (3) decreased genes in mouse LUAD cells, KP-1 cells, by *shPclaf* (compared to control shRNA). We selected each drug candidate above a connectivity score 1.5 and isolated PAPC-specific drugs.

### Gene expression analysis by qRT-PCR
RNAs were extracted by TRIzol (Invitrogen) and used to synthesize cDNAs using the iScript cDNA synthesis kit (Biorad). qRT-PCR was performed using an Applied Biosystems 7500 Real-Time PCR machine with the primers listed in Supplementary Data 7. Target gene expression was normalized to that of mouse *Hprt1*. Comparative $2^{-\Delta\Delta Ct}$ methods were used to quantify qRT-PCR results.

### Statistics
All statistical analyses were performed using GraphPad Prism (version 10.3.0, GraphPad Software, San Diego, CA). Data are presented as mean ± standard derivation of the mean (SD) unless otherwise specified in the figure legend. For comparisons between two groups, two-tailed Student's t tests were used. Two-way ANOVA with Tukey's multiple comparisons test was used to analyze data with two independent variables in some experiments. Concentration–response curves were fitted using a four-parameter logistic equation. For all analyses, $p < 0.05$ was considered statistically significant. The specific statistical test used and the number of experiments performed for each analysis are indicated in the corresponding figure legend. Normal distribution was assumed for all analyses, and no data points were excluded. To identify marker genes using scRNA-seq data, the Wilcox Rank Sum test was used, using Seurat package (Supplementary Data 1–3). GO analysis with a gene set of FISCHER_DREAM_TARGETS was analyzed using a Binomial statistic by PANTHER db (Supplementary Data 6). All experiments were performed three or more times independently under identical or similar conditions.

### Reporting summary
Further information on research design is available in the Nature Portfolio Reporting Summary linked to this article.

## Data availability

The scRNA-seq data generated in this study have been deposited in the Gene Expression Omnibus database under accession code GSE205815. Public scRNA-seq data of mice GSE141259 or humans GSE135893 are available via the Gene Expression Omnibus. Source data are provided with this paper.

## Code availability

The code used to reproduce the analyses described in this manuscript can be accessed via Zenodo (https://zenodo.org/records/13871131)[70] and will be available upon request.

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

## Acknowledgements

We are grateful to Pierre D. McCrea and Malgorzata Kloc for insightful comments; Joel M. Sederstrom for technical assistance; Stuart H. Yuspa (Center for Cancer Research, NCI) for reagents (pAltermax-3HA-CLIC4); and Ann Sutton (Research Medical Library, MD Anderson) for editing the manuscript. This work was supported by the Cancer Prevention and Research Institute of Texas (RP200315 to J.-I.P.), the National Cancer Institute (K99 CA286761 to K.-P.K; R01 CA193297, R01 CA278971, R03 CA279867, and R03 CA256207 to J.-I.P.; R01 CA278967 to K.-S.P. and J.-I.P.), an Institutional Research Grant (MD Anderson to J.-I.P.). The core facilities at MD Anderson (DNA sequencing and Genetically Engineered Mouse Facility) were supported by National Cancer Institute Cancer Center Support Grant (P30 CA016672 to MD Anderson). The core facilities at Baylor College of Medicine (Cytometry & Cell Sorting Core and Single Cell Genomics Core) were supported by CPRIT (RP180672, RP200504) and the National Institutes of Health (CA125123, RR024574).

## Author contributions

Conceptualization: B.K. and J.-I.P.; methodology: B.K., L.V.E., K.-S.P., J.C., M.P., C.K., and J.-I.P.; validation: B.K. and J.-I.P.; formal analysis: B.K. and J.-I.P.; investigation: B.K., Y.H., K.-P.K., S.Z., G.Z., J.Z., M.J.K., S.J., and J.-I.P.; data curation: B.K., D.L., and J.C.; visualization: B.K. and J.-I.P.; supervision: J.-I.P.; project administration: B.K. and J.-I.P.; funding acquisition: J.-I.P.; writing—original draft: B.K. and J.-I.P.; writing—review and editing: B.K and J.-I.P.

## Competing interests

The authors declare no competing interests.
