## [Peer Review File · Nature Communications]

REVIEWER COMMENTS

Reviewer #1 (Remarks to the Author):

In their manuscript, Kim et al. examine the role of the PCLAF-DREAM axis in lung regeneration in the context of pulmonary fibrosis. They show that activation of DREAM targets induces the exit of alveolar stem progenitor proliferation and that its abrogation reduces AT1 cell numbers. This is mediated through *Clic4* which promotes TGF- β signaling and AT2 differentiation. I believe that this work is in line with the journal's mission and that it would be a valuable resource toward pharmacological interventions to boost lung regeneration in IPF patients. On the other hand, the exact mechanistic aspect of PCLAF-DREAM interactions remained elusive. I provide below my comments to improve the manuscript.

Major comments

1. The introduction section is quite short and the PCLAF-DREAM axis needs to be described in more detail. For example, what is known about the role of PCLAF and DREAM targets in the homeostatic and perturbed lung? Which cells express the components of this signaling pathway and are there human *ex vivo* or animal models where inhibition of this pathway has promoted disease? In addition, it would be nice if the authors added one more paragraph where they summarize their key findings.
2. What cells does the PPC cluster consist of? From Fig. S1A-B, I conclude that they also express alveolar epithelial cell markers. I would therefore recommend being more specific about the nomenclature to indicate that these cells are alveolar progenitors.
3. How does the induction of *Pclaf* expression in the Bleomycin model coincide with inflammatory and lung injury and regeneration markers across the chosen time course? This is crucial to understand the gene expression dynamics with respect to lung damage/healing.
4. The lung organoid system raises a few concerns; the fact that bronchioalveolar and bronchiolar organoids are formed suggests that the epithelial cell source is quite diverse. What was the rationale of using *Epcam*⁺ cells? Did the authors perform flow cytometry staining of the magnetically purified epithelial cells? Have the authors tried AT2 cell sorting instead?
5. Fig. 3: the message of altered differentiation process in *Pclaf* KO mice is not very straightforward. It is not intuitive how AT2 cells undergo differentiation from one phenotype (*Lcn2*) to another (*Slc34a2*) via a proliferative state. The authors should investigate further the identified alveolar epithelial populations in the two groups and provide a working model of how they think *Pclaf* deficiency affects alveolar remodeling upon injury. In addition, other trajectory inference tools (e.g. Slingshot, Diffusion maps) and experimental validation *ld* be used to validate the model and strengthen the figure.

6. A major limitation of the manuscript is that it evades elucidating the mechanism via which Pclaf-DREAM interactions regulate Clic4-TGF- β signaling. Is it a direct interaction between the DREAM complex and Clic4 or does it occur via other mediators? Unfortunately, the authors did not comment on this. This should be a point of further exploration in this study.

7. I would strongly recommend the authors to repeat the Phenelzine effects on a human ex vivo model, such as precision-cut lung slices to provide the human perspective, as well. Does Phenelzine treatment enhance regeneration and AT2 differentiation?

8. The transcriptomics section in the Methods severely lacks detail. The authors should revise accordingly to include missing aspects (e.g. data integration, combined signature enrichment, CLUE platform, etc.) of their analysis. Furthermore, following the code availability statement, I was not able to locate the data on github.

9. Again, in the Discussion, the authors should comment more on the mechanism via which PCLAF-DREAM regulates CLIC4-TGF- β and alveolar regeneration. Currently, this aspect is underrepresented. Additionally, the authors did not discuss the outcome of the Phenelzine treatment experiments as a potential treatment for pulmonary fibrosis.

Minor comments

1. Fig. 1G: please stretch the panel, it is difficult to evaluate the gene expression differences.

2. The contrast in Fig. 1 K is not optimal and the data cannot be properly evaluated. Since the mouse strain is not published anywhere else, the authors must provide the validation together with the strategy in a supplementary figure for the interested reader.

3. The PBS WT/KO control is missing from Fig. 2D. Please add.

4. What was the rationale for the two experimental designs in lung organoids? Why did the authors prompt for the intermediate and withdrawal models? This was not very clear in the manuscript, especially given the fact that they may yield contradictory results (e.g. Fig. S11).

5. Overall, the authors should consider better color codes in their figures. The default hue color code is not optimal for labeling several cell types on UMAP plots.

6. Fig. 6: as stated elsewhere, since the focus of the manuscript is the alveolar regeneration, I do not follow panels C-D bottom.

Reviewer #2 (Remarks to the Author):

General Comments:

This manuscript addresses an important problem, the mechanism of lung regeneration and fibrosis. However, the authors use a global knockout mouse to make conclusions about PCLAF function in epithelial cells, though it is expressed in other cell types. The methods used to measure proliferation and AT1 transition are problematic. PCLAF induces the cell cycle. The most convincing data is that PCLAF is necessary for AT2 cell proliferation in the organoids, resulting in enhanced CFE. The quantitation of AT2 and AT1 cell numbers is not convincing and the conclusions about the role of PCLAF in AT1 transition is based on this quantitation as well as pseudotime data. The organoid data is helpful to exclude the role of PCLAF in other cell types; however, the dramatic effect of PCLAF on the CFE confounds interpretation of AT2 to AT1 transition data. The therapeutic potential of PCLAF mimicking drugs is a notable strength.

Major Comments:

1. Fig. 1I-L: Other cell types besides epithelial cells proliferate in this model. Costaining with epithelial cell markers should be performed. Representative images and quantitation of PCLAF+ epithelial cells should be shown. Sections should be costained for immune and mesenchymal markers, and PCLAF expression in these cells should be shown.
2. A major point of this manuscript is the effect of PCLAF on AT2 cell proliferation and the transition to AT1 cells. However, the data in Fig. 2C,D are problematic. First, the RAGE staining at 0 dpi does not appear to be specific for AT1 cells, as there are large, round RAGE+ entities. An enlarged image should be shown to convince that the RAGE staining is working. More importantly, there are concerns about the quantitative data. Methods for the quantitation in Fig. 2D should be explained. Specifically, the denominator for the percent calculation should be stated. If the denominator is total cells (DAPI+), this is problematic because the injured lungs have many more cells due to immune cell recruitment to the lung, which will artificially decrease the percent of epithelial cells. In addition, it is not possible to count the number of RAGE+ cells since individual cells cannot be identified. Many other papers have demonstrated that there are less AT1 cells at 21 dpi than at 0 dpi in WT lungs in the bleomycin model, so the quantitation performed here is highly inconsistent with established literature. A marker of S phase may be helpful. Lineage tracing should be performed because the number of AT1 cells, even if appropriately quantified, may reflect a role of PCLAF in cell death instead of regeneration.
3. In Fig. 2E-G, the authors claim to show that KO mice have more fibrosis. The data are fairly convincing, but 0 dpi data should be added to the quantitation in 2H. The time point for the data in 2H should be stated; if it is 21 dpi, the amount of fibrosis in WT mice appears to be very low compared to the literature. Hydroxyproline assay is a less biased method to measure the degree of fibrosis than staining and image analysis. Finally, since a global KO is used, the authors must demonstrate that PCLAF is not expressed in other cell types, particularly fibroblasts and immune cells. Review of the dataset that the authors used shows that Pclaf (2810417H13Rik) is highly expressed in immune cells. To definitively support their claim, a conditional knockout in epithelial progenitors and lineage tracing should be performed.

4. The immunostaining in Fig. S4 is unclear. High resolution images and isotype control staining should be shown. It is concerning that there are no SPC+ cells in the WT organoids. The impaired colony forming efficiency shown in Fig. S4 is dramatic but not surprising since the function of proliferating cell nuclear antigen-associated factor (PCLAF) is to promote cell proliferation. Moreover, the dramatic inability to form organoids confounds the AT1 transition data. It is difficult to draw conclusions about AT1 cell transition from the limited remaining organoids.

5. scRNAseq cannot be used to quantify cell numbers (Fig. 3) since enzymatic dissociation results in cell loss, particularly of AT1 cells. Pseudotime analysis is predictive but lineage tracing is needed to validate the predictions.

Minor Comments:

1. The Abstract should be edited for language. For example, there are several confusing phrases. “Orchestration” should be more explicitly defined. The difference between “cell quiescence exit” and “cell proliferation” is unclear. “Pulmonary proliferative cells” should be defined. The term “acutely” is nonspecific – does this mean rapidly? “Intriguingly” should be deleted. “AT1” should be “AT1 cells”. The abbreviation “PPCs” should be defined. The term “pharmacological mimicking” is unclear.
2. It’s unclear why in Fig. 1G the plots are different than in Fig. 1A since the data are the same. Either all the cell types listed in Fig. 1A should be identified on the 1G plots or, preferably, 1G should show Pclaf expression on the plots configured as in 1A.
3. The authors state that “Bleomycin initially damages the alveolar epithelium, including alveolar type I (AT1) and AT2 cells, followed by inflammation and interstitial fibrosis” and cite refs. 22-24. Data demonstrating that AT1 and AT2 cells are damaged prior to inflammation in this model are not immediately obvious in these references. These data should be explained or this claim should be deleted.
4. In Fig. S4, “#1” and “#2” should be defined.

Reviewer #3 (Remarks to the Author):

In this manuscript, Kim et al., described a pivotal role for PCLAF-DREAM complex in alveolar regeneration. First the authors utilized published scRNA-seq datasets from regenerating mouse lungs and human normal and IPF lungs and show that proliferating cells are enriched for PCLAF

expression. The authors went on to check PCLAF role in lung regenerating by using PCLAF KO mice and tested its role in control and bleomycin-induced injury models. The authors state that loss of PCLAF blocks AT2 to AT1 differentiation. The authors state that PCLAF knockout cells favored cell trajectory from proliferating AT2s to AT2 (Slc34a2^{high}) cells. This is in contrast to control cells where proliferating AT2s show trajectory towards AT1s. Mechanistically, the authors claim that PCLAF-DREAM activated CLIC4-TGF β signaling is necessary for AT2 to AT1 differentiation. Further, the authors claim that pharmacological mimicking of PCLAF-DREAM signaling is sufficient to depress AT2 to AT1 differentiation blockade observed in PCLAF knock out lungs.

Overall, the authors attempt to address mechanisms that control the choice of AT2s proliferation and differentiation. I think the authors attempt to address an important question. But the data presented here falls short and does not support the claims. Authors drew many conclusions from scRNA-seq data but did not provide additional validity support for them. See below for detailed comments.

Major comments:

1. One of the main claims in this manuscript is that “PPC”s are stalled in Pclaf KO lungs. If PPC are stalled, then one would expect more PPCs in these lungs. However, the quantification data presented in Fig-3G doesn’t support this claim.

2. Along the same lines, the authors used scRNA-seq to quantify the numbers of different cell populations. Such quantification is not recommended as lung tissue dissociation efficiency may vary in normal, regenerating, and perturbed conditions. Therefore, one need to perform quantification analyses on lung tissues using specific markers for different cell populations described in this manuscript.

3. The authors used “PPCs” to describe proliferating pulmonary cells. However, the authors were actually referring or at least characterized Mki67⁺ AT2s, which were previously referred to as “proliferating AT2s”. Similarly, the authors claim that they identified Lcn2^{high} AT2s. These were previously referred to as “activated AT2s”. The use of different terminology to refer to these previously described cell populations is confusing. Therefore, we suggest that authors reconsider using previously described names.

4. The authors described multiple cell populations which they referred to as “transitional cell states”, including PPCs, Krt8⁺ cells, AT2^{med}/AT1^{med}, Lcn2^{high} AT2s. Further, the authors claim

that some of these populations were dysregulated in Pclaf KO lungs. These claims need to be validated using specific markers on tissue sections.

5. What was the efficiency of Pclaf deletion in all lungs assessed? There are many “PPEs” in Pclaf KO lung scRNA-seq data. Were they not deleted for Pclaf?

6. The authors used in silico lineage prediction tools and suggest cell trajectories to and from multiple cell populations. For example, in Fig-3H the authors claim bi-directional flow between PPCs and Lcn2 high-AT2s and AT1/AT2 to Lcn2 high-AT2s. Additionally, the authors claim that bidirectional trajectory is missing between PPCs and AT2 (Lcn2 high) cells in Pclaf KO lungs. These claims need validation using lineage tracing models.

7. In lines 104 to 106, the authors state that “ While LOs with LuMSCs were mainly differentiated into the bronchiolar type,.....”. Numerous studies have shown that AT2s generate LO’s with AT2 and AT1 cells in the presence of alveolar fibroblasts. Its not clear why the authors only got bronchiolar type organoids in the presence of LuMSCs?

8. The authors used “alveolar plasticity” numerous times throughout the manuscript in different contexts. For example, in lines 114-115, the authors state “....plasticity of AT1 and AT2 cell lineage during lung regeneration”. It is not clear what type of plasticity the authors are referring to in the context of AT1 cells? Similarly, the authors used “plasticity” while referring to AT2 to AT1 differentiation. AT2 to AT1 lineage conversion is simply – differentiation. Therefore, we suggest that authors use appropriate terminology.

9. In lines 213-214, the authors discussed about markers of transient cell states. One of the citations (#25) did not describe Krt8+ cells. This should be corrected.

10. Does PCLAF-DREAM complex directly bind “PPE” targets genes to control their fate?

Responses to Reviewers' Comments

Reviewer 1

*"In their manuscript, Kim et al. examine the role of the PCLAF-DREAM axis in lung regeneration in the context of pulmonary fibrosis. They show that activation of DREAM targets induces the exit of alveolar stem progenitor proliferation and that its abrogation reduces AT1 cell numbers. This is mediated through *Clic4* which promotes TGF- β signaling and AT2 differentiation. I believe that this work is in line with the journal's mission and that it would be a valuable resource toward pharmacological interventions to boost lung regeneration in IPF patients. On the other hand, the exact mechanistic aspect of PCLAF-DREAM interactions remained elusive. I provide below my comments to improve the manuscript."*

Major comments

A1. *"The introduction section is quite short and the PCLAF-DREAM axis needs to be described in more detail. For example, what is known about the role of PCLAF and DREAM targets in the homeostatic and perturbed lung? Which cells express the components of this signaling pathway and are there human ex vivo or animal models where inhibition of this pathway has promoted disease? In addition, it would be nice if the authors added one more paragraph where they summarize their key findings."*

We greatly appreciate your valuable feedback. As suggested, we added a brief overview of the PCLAF-DREAM axis and a summary of key findings in the Introduction (see below).

Unfortunately, there have been no specific studies investigating the role of PCLAF, DREAM, or the PCLAF-DREAM complex in lung pathophysiology. In a previous study, we identified PCLAF as a binding partner of the DREAM complex in lung cancer ². Other studies have demonstrated that the DREAM complex accelerates the regeneration of β -cells ³ and muscle tissue ⁴. Furthermore, our previous research has indicated that PCLAF promotes intestinal regeneration ¹.

"The PCLAF (PCNA Clamp Associated Factor; also known as KIAA0101 or PAF) was identified as a proliferating cell nuclear antigen (PCNA)-interacting protein modulating DNA repair and replication ^{14,15}. PCLAF hyperactivates WNT/ β -catenin signaling independently of PCNA in colorectal cancer ¹⁶, promotes pancreatic tumorigenesis ¹⁷, induces cell plasticity of breast cancer ¹⁸, facilitates stem cell activation and proliferation for intestinal regeneration ¹⁹, and promotes lung tumorigenesis via the DREAM complex ²⁰.

The dimerization partner, retinoblastoma (RB)-like, E2F, and multi-vulval class B (DREAM) complex is an evolutionarily conserved multiprotein complex that orchestrates cell quiescence and the cell cycle ²¹⁻²⁴. Dissociation of the multi-vulval class B (MuvB) core complex with RBL2 (retinoblastoma-like protein 2/p130), E2F4, and DP1 (E2F dimerization partner 1) drives the MuvB complex to bind to BYMB and FOXM1, transactivating cell cycle-related DREAM target genes and leading to cell quiescence exit and cell proliferation ²⁵. PCLAF directly binds to and remodels the DREAM complex to bypass cell quiescence and promote cell proliferation ²⁰. Given the roles of the PCLAF in modulating cell proliferation, cell plasticity, or stemness in various pathophysiological conditions ¹⁶⁻²⁰, we interrogated the roles of the PCLAF-DREAM axis in lung regeneration.

Employing comprehensive approaches, including single-cell transcriptomics, organoids, and mouse models, we herein showed that the PCLAF-DREAM-mediated alveolar cell plasticity is indispensable for lung regeneration. PCLAF depletion impaired lung regeneration and led to lung fibrosis with decreased repopulation of AT1 cells. Additionally, we found that the PCLAF-DREAM complex transactivates *CLIC4* to activate TGF- β signaling for AT1 cell generation from AT2 cells. Furthermore, we identified a potential drug candidate mimicking the PCLAF-activated transcriptome for suppressing lung fibrosis."

A2. “What cells does the PPC cluster consist of? From Fig. S1A-B, I conclude that they also express alveolar epithelial cell markers. I would therefore recommend being more specific about the nomenclature to indicate that these cells are alveolar progenitors.”

We appreciate your insightful comments. As recommended, we revised the nomenclature for better clarification. Briefly, we changed the nomenclature of PPCs into proliferating alveolar progenitor cells (PAPCs).

It turns out that the PAPCs are a collection of cells derived from various sources with proliferative potential. In a bleomycin-mediated lung regeneration mouse model, it is believed that new AT2 cells arise from subsets of bronchiolar epithelial cells, including basal cells, club cells, and pre-existing AT2 cells⁵. Previous studies have demonstrated that both AT2 cells^{6,7} and club cells⁸ serve as source cells for AT2 cell generation.

We further analyzed PAPCs using the scRNA-seq datasets of mouse and human lung epithelial cells. We found that PAPCs in both mouse and human lungs expressed the markers associated with AT2 cells (*SFTPC*), club cells (*SCGB1A1*), and basal cells (*KRT5*), with *SFTPC*-expressing cells being the most abundant (Fig. R1A-D). In line with this, our scRNA-seq data showed that PAPCs predominantly expressed *Sftpc*, with some cells expressing *Scgb1a1*, which was also confirmed by immuno-staining (Fig. R1E-J). These results suggest that *PAPCs are primarily derived from AT2 cells*. However, due to the expression of club cell markers in PAPCs, we were unable to assign a specific nomenclature and referred to them as PAPCs, as kindly suggested. These new data were included in the revised manuscript (Supplementary Fig. 9).

We also revised the result section in the manuscript, accordingly, as shown below.

“Of note is that PAPCs were highly enriched with SPC, a representative marker for AT2 cells (Supplementary Fig. 9), implying that most PAPCs are likely derived from AT2 cells.”

Figure R1. PAPCs express markers of AT2 and Club cells.

A, B, Analysis of the scRNA-seq datasets shown in Figure 1A. UMAP-embedding displays cells colored by cell type identity (A). Feature plots of indicated gene expression in subset of PAPCs (B).

C, D, Analysis of the scRNA-seq datasets shown in Figure 1D. UMAP-embedding displays cells colored by cell type identity (C). Feature plots of indicated gene expression in subset of PAPCs (D).

E, F, Analysis of the scRNA-seq dataset shown in Figure 3A. UMAP-embedding displays cells colored by cell type identity (E). Feature plots of indicated gene expression in subset of PAPCs (F).

G, H, Representative images of WT lung at 7 dpi of bleomycin; immunostaining for MKI67 and SPC; MKI67 (G) and SCGB1A1; MKI67 (H).

I, J, *Pclaf-lacZ* mice were instilled with bleomycin (1.4 U/kg). After 7 dpi, lungs were collected and performed whole-mount staining for X-gal. Representative image of lung chemically immunostained for SPC (I).

Representative image of lung chemically immunostained for SCGB1A1 (J).

A3. “How does the induction of *Pclaf* expression in the Bleomycin model coincide with inflammatory and lung injury and regeneration markers across the chosen time course? This is crucial to understand the gene expression dynamics with respect to lung damage/healing.”

We agree that it is indeed crucial to understand the interplay between inflammation, apoptosis, regeneration, and the expression of PCLAF. In the context of lung regeneration, there are no established genes specifically associated with this process. Instead, we assessed the repopulation or generation of AT1 and AT2 cells in alveoli lesions. To assess lung regeneration and damage, we performed immunostaining for AT1 (RAGE+) and AT2 (SPC+) cells. We observed a significant reduction in both AT1 and AT2 cells at 3 dpi (days post injury, bleomycin intra-tracheal instillation), followed by their regeneration at 7 dpi (Fig. R2A-D). Additionally, we detected epithelial cells expressing cleaved caspase-3 (CC3), a cell death marker, at 3 dpi (Fig. R2E, F).

To further address your comments, we conducted an RT-qPCR analysis to evaluate the expression of genes associated with inflammation and apoptosis. We examined the mRNA expression levels of *Tnfa*, *Il1a*, *Il6*, and *Cox2*, markers for inflammation, and the *Bax/Bcl2* mRNA expression ratio as an indicator of apoptosis. At 3 dpi, we observed elevated mRNA expression of *Tnfa*, *Il6*, and *Cox2* and their restoration at 7 dpi (Fig. R2G). The *Bax/Bcl2* ratio showed an increase at 7 dpi followed by a decrease at 14 dpi after bleomycin instillation (Fig. R2H). In general, the onset of inflammation and apoptosis preceded *Pclaf* expression, while the regeneration process exhibited similar temporal patterns to *Pclaf* expression (Fig. R2I).

It has been believed that bleomycin-induced lung damage is followed by inflammation, as cell damage occurs prior to the inflammatory response⁹⁻¹¹. However, there is limited direct evidence regarding the precise temporal sequence of cell damage and inflammation caused by bleomycin. Nevertheless, considering the additional information that *Pclaf* expression is minimal in homeostatic lungs but increases in response to lung damage, these findings suggest that the orchestration of inflammation and cell damage may trigger *Pclaf* expression, which is associated with lung regeneration.

These new data were included in the revised manuscript (Fig. 1H, I, Fig. 2C-F, Supplementary Fig. 3D-G).

We have also revised the result section in the manuscript, as shown below.

“The expressions of inflammatory genes, *Tnf α* , *Il6*, and *Cox2*, reached their peak at 3 days post-injury (dpi) (Fig. 1H), and the ratio of *Bax/Bcl2*, which reflects cell apoptosis, also reached its maximum level at 3 dpi (Fig. 1I).”

Figure R2. Inflammation, apoptosis, and regeneration along with the *Pclaf* expression.

Pclaf WT and KO mice were treated with phosphate-buffered saline (PBS; control) or bleomycin by intratracheal instillation. At 3, 7, or 21 dpi of bleomycin or 21 dpi of PBS (control), the lungs were collected for further analysis.

A, Representative images of lungs; immunostaining for RAGE and CDH1 at indicated time points.

B, Representative images of lungs; immunostaining for RAGE and SPC at indicated time points.

C, Quantification of the ratio of RAGE⁺ area / CDH1⁺ area from the images related to Figure R2A.

D, Quantification of SPC⁺ cells from the images related to Figure R2B.

E, Representative images of lungs; immunostaining for CC3 (cleaved-caspase-3) and CDH1 at indicated time points.

F, Quantification of CC3⁺ CDH1⁺ cells.

G, RT-qPCR analysis of the lung tissues at indicated time points. *Hprt* was used as an internal control.

H, Ratio of *Bax* / *Bcl2* analyzed by RT-qPCR at indicated time points.

I, Illustration of tentative dynamics of inflammation, cell damage, PCLAF expression, regeneration, and fibrosis in the course of damage-induced lung regeneration.

A4. “The lung organoid system raises a few concerns; the fact that bronchioalveolar and bronchiolar organoids are formed suggests that the epithelial cell source is quite diverse. What was the rationale of using *Epcam+* cells? Did the authors perform flow cytometry staining of the magnetically purified epithelial cells? Have the authors tried AT2 cell sorting instead?”

Rationale: We agree with your point. It was suggested that all lung epithelial progenitor cells express EPCAM^{7,8,12}, and both alveolar progenitor cells^{7,12} and bronchiolar progenitor cells⁸ contribute to alveolar regeneration. Thus, we initially cultured lung organoids (LOs) using all lung epithelial cells (CD34-/Ter119-/CD45-/EPCAM+).

Cell sorting: As we described in the methods, we sorted lung epithelial cells (CD34-/Ter119-/CD45-/EPCAM+) by magnetic-activated cell sorting (MACS) using Myltenyl Biotec kits.

As pointed out, to gain a better understanding of alveolar regeneration, utilizing AT2 cells for organoid culture can provide more direct insights. For fluorescence-activated cell sorting of mouse AT2 cells, genetic labeling using a *Sftpc*-Cre driver mouse is commonly used⁷. Unfortunately, we did not have a *Sftpc*-Cre driver at the time of submission. Additionally, considering that AT2 cells are not the sole source of alveolar regeneration⁸, comprehensive and diverse sources could better help to understand overall cellular plasticity for alveolar regeneration. Therefore, our analysis was limited to alveolar LOs to illustrate our proposed mechanism.

To better understand the role of PCLAF in alveolar cell plasticity, we performed cell lineage tracing experiments. First, we generated *Pclaf*-fl/fl mice using *Pclaf*-LacZ-neo mice followed by confirming Cre-mediated knock-out by genotyping of *Pclaf*-fl/fl mouse embryonic fibroblasts by Ad-Cre (Fig. R3A, B). After that, we performed lineage-tracing of AT2 cells using *Sftpc*-CreERT2; LSL-Sun1-EGFP mice combined with *Pclaf* WT, *Pclaf* KO, or *Pclaf*-fl/fl (*Pclaf* conditional KO [cKO]) mice. Both *Pclaf* KO and *Pclaf* cKO mice showed that AT1 cells and KRT8+ cells from AT2 cells were decreased compared to *Pclaf* WT during lung regeneration (Fig. R3C-F). These data consistently indicate that PCLAF is crucial in alveolar cell plasticity, especially for regeneration of AT1 cells. We believe that this cell lineage tracing result strongly supports our model. These new data were included in the revised manuscript (Fig. 3G, Supplementary Fig. 10).

Nonetheless, it should be noted that our study primarily focused on understanding how PCLAF regulates cellular plasticity rather than investigating the origin of regenerated cells or their cellular trajectories. The organoid experiments demonstrated that the PCLAF-DREAM axis controls alveolar cellular plasticity, contributing to the maintenance of a balanced regeneration between AT1 and AT2 cells through the promotion of CLIC4-activated TGF- β signaling (Fig. 4, 5). Specific cellular origins regulated by PCLAF need to be investigated in further study.

We also revised the result section in the manuscript, accordingly, as shown below.

“For experimental validation of the predicted cell lineage trajectories, we also performed a cell lineage tracing analysis using genetically engineered mice. We first established *Pclaf*-fl (floxed)/fl mice for conditional KO (cKO) of *Pclaf* alleles (Supplementary Fig. 10A, B). AT2 cell-specific Cre driver (*Sftpc*^{CreERT2} [*Sftpc*-CreERT2]) and Cre-loxP recombination reporter (*Rosa26*^{Sun1GFP} [*Sun1*-GFP]) were combined with either *Pclaf* -/- (germline KO) or *Pclaf*-fl/fl (for conditional KO [cKO]) strains. Then, bleomycin was instilled into the lung, followed by tamoxifen administration for genetic labeling and lineage tracing of cells derived from AT2 cells (*Sftpc*-Cre-driven GFP+ cells). Consistent with cell lineage trajectory inference and immunostaining results (Fig. 3C-F, Supplementary Fig. 10), AT1 cells derived from AT2 cells were significantly reduced and AT2 cells were elevated in both *Pclaf* KO and cKO lung tissues compared to *Pclaf* WT (Fig. 3G, and Supplementary Fig. 10C-E). Additionally, RAGE and SPC double-positive cells (AT2^{med}/AT1^{med} cells) derived from AT2 cells were significantly decreased in *Pclaf* cKO lung, and KRT8+ cells were lessened in both *Pclaf* KO and cKO lung (Fig. 3G, and Supplementary Fig. 10C-E). In line with these results, the CytoTRACE analysis¹³ showed a relatively less differentiated cell status in activated AT2 cells and transitioning AT1^{med}/AT2^{med} cells of *Pclaf* KO lungs than of WT lungs, indicating the impaired maturation of these cells into AT1 cells in the condition of *Pclaf* ablation (Fig.

3H). These results suggest that PCLAF plays a pivotal role in alveolar cell lineage plasticity that induces AT1 cell regeneration from AT2 cells during lung regeneration.”

Figure R3. *Pclaf* KO inhibits differentiation of AT1 cells from AT2 cells.

A, Experimental scheme of generating of *Pclaf-fl/fl* mice.

B, Mouse embryonic fibroblasts (MEFs) were isolated from *Pclaf-fl/fl* mice. 1 X 10⁷ or 1 X 10⁸ PFU of Ad-Cre were introduced in *Pclaf-fl/fl* MEFs. Conditional knock-out by Ad-Cre were checked by genotyping.

C-F Sttpc-CreERT2; LSL-Sun1-EGFP; *Pclaf* WT, Sttpc-CreERT2; LSL-Sun1-EGFP; *Pclaf* KO, and Sttpc-CreERT2; LSL-Sun1-EGFP; *Pclaf-fl/fl* mice were instilled with bleomycin Then mice were injected with tamoxifen for 5 consecutive days. At 14 dpi, lungs were collected and analyzed.

C, Representative images of lungs; immunostaining for RAGE, SPC, and EGFP. White arrow; RAGE⁺/EGFP⁺ cell. Red arrow; SPC⁺/EGFP⁺ cell. Yellow arrow; RAGE⁺/SPC⁺/EGFP⁺ cell.

D, Enlarged image of RAGE and SPC double positive cell.

E, Representative images of lungs; immunostaining for KRT8 and EGFP. White arrow; KRT8⁺/EGFP⁺ cell.

F, Quantification of indicated cells in figure R3 Cand E.

A5. “Fig. 3: the message of altered differentiation process in *Pclaf* KO mice is not very straightforward. It is not intuitive how AT2 cells undergo differentiation from one phenotype (*Lcn2*) to another (*Slc34a2*) via a proliferative state. The authors should investigate further the identified alveolar epithelial populations in the two groups and provide a working model of how they think *Pclaf* deficiency affects alveolar remodeling upon injury. In addition, other trajectory inference tools (e.g. Slingshot, Diffusion maps) and experimental validation should be used to validate the model and strengthen the figure.”

We appreciate your insightful comments. We acknowledge that our data did not provide a comprehensive understanding of the exact mechanisms and status of trans-differentiation from AT2 cells to AT1 or AT2 cells.

As mentioned by reviewer #3, *Lcn2* is expressed upon activation of AT2 cells^{7,12}. Thus, we have revised the terminology in the revised manuscript. We replaced "AT2 (*Lcn2* high)" with "activated AT2" and "AT2 (*Slc34a2* high)" with "AT2 cells" to address potential confusion.

Taking into consideration your valuable comments, we realized that the original Figure 3H was overstated. In this study, we aimed to propose a model of cellular trajectory based on in-silico prediction. As kindly suggested, we performed additional trajectory inference analysis using the Slingshot package¹⁴. The results from the Slingshot analysis aligned with the RNA velocity analysis (Fig. R4A). In *Pclaf* wild-type (WT) mice, AT2 cells (*Slc34a2* high) showed a trajectory into AT1 cells through PAPC-activated AT2 (*Lcn2* high)-AT2^{med}/AT1^{med} or through PAPC-Krt8⁺ cells (Fig. R4A). However, in *Pclaf* KO mice, AT2 cells did not transition into AT1 cells (Fig. R4A). Moreover, activated AT2 cells in *Pclaf* WT differentiated into both AT2 and AT1 cells, while activated AT2 cells in the KO group primarily transitioned into AT2 cells (Fig. R4A).

These findings were consistent across two different trajectory inference analyses and were further supported by in vivo (Fig. R4B, C) and organoid experiments (Fig. R4D, E), which demonstrated reduced AT1 cells and increased AT2 cells in *Pclaf* KO. Moreover, the lineage tracing of AT2 cells using *Sftpc*-CreERT2 mice showed the impaired transition of AT1 cells from AT2 cells by *Pclaf* KO, consistently (Fig. R3). Collectively,

these results suggest that PCLAF acts as a key regulatory factor balancing cell fate between AT1 and AT2 cells. We have included these new data in the revised manuscript (Fig. 3E).

Although AT2 cells are known to possess self-regeneration properties⁵, the underlying mechanisms are not yet fully understood. In our study, we predicted and suggested that AT2 cells are activated under proliferative conditions and subsequently contribute to the regeneration of AT2 cells. However, this aspect requires validation in further studies. The lack of specific markers for activated AT2 cells presents a challenge for validation using lineage tracing. Therefore, the identification of specific markers for activated AT2 cells should be a priority.

We have revised the tone related to cellular trajectories in the revised manuscript, as shown below.

"In line with previous studies^{26,31,33}, a cell lineage trajectory inference using Slingshot³⁴ and RNA velocity³⁵ predicted the cellular trajectory from AT2 cells into AT1 cells in *Pclaf* WT (Fig. 3C-E, Supplementary Fig. 8)."

A6. “A major limitation of the manuscript is that it evades elucidating the mechanism via which Pclaf-DREAM interactions regulate *Clc4*-TGF- β signaling. Is it a direct interaction between the DREAM complex and *Clc4* or does it occur via other mediators? Unfortunately, the authors did not comment on this. This should be a point of further exploration in this study.”

We appreciate your acknowledgment and apologize for the previous lack of detailed information regarding the mechanism by which PCLAF-DREAM regulates *CLIC4*. To address this, we conducted additional experiments to elucidate the interaction between PCLAF-DREAM and the *CLIC4* gene.

Based on previous studies that identified *CLIC4* as a direct target gene of the DREAM complex through ChIP-sequencing¹⁵, we predicted *CLIC4* to be a direct target of the PCLAF-DREAM complex in our previous study². To validate this, we performed chromatin immunoprecipitation (ChIP) experiments. Utilizing the ChIP-Atlas database (chip-atlas.org), we found that several components of the DREAM complex, including RBBP4, RBBP5, RPPB7, Lin9, E2F1, E2F3, E2F4, E2F6, E2F7, and MYBL2, were observed to bind to the *CLIC4* promoter, indicating the potential interaction of the PCLAF-DREAM complex with the *CLIC4* promoter (Fig. R5A). Indeed, ChIP-PCR showed that ectopically expressed 3FLAG-PCLAF bound to two regions of the *CLIC4* proximal promoters in human lung adenocarcinoma cells (H358 cells) (Fig. R5B). Consistently, endogenous PCLAF was observed to bind to the *CLIC4* proximal promoter region (Fig. R5C), indicating that PCLAF directly transactivates *CLIC4* expression.

As we performed rescue experiments for the functional implications in the initial manuscript, the ectopic expression of *CLIC4* significantly rescued the decreased AT1 regeneration (Fig. R5D-F) and phosphorylated SMAD3 levels (Fig. R5G, H) observed in *Pclaf* KO LOs. Additionally, external stimulation with TGF- β 1 rescued the AT1/AT2 cell ratio in *Pclaf* KO LOs (Fig. R5I-K). These findings suggest that *CLIC4*, directly promoted by PCLAF, plays a

crucial role in regulating alveolar cell plasticity by promoting TGF- β signaling.

We have added these new data to Figure 5C and Supplementary Figure 14B and C of the revised manuscript to provide a comprehensive understanding of the mechanisms underlying PCLAF-DREAM regulation of CLIC4 and its implications for alveolar cell plasticity.

H, Quantification of the p-SMAD3⁺ cells.

I-K, LOs treated with TGF- β 1 for the indicated time before the end of culture, were fluorescently immunostained for HOPX (red) and SPC (light blue) (I). Representative images are shown. Quantification of the HOPX⁺ (J) and SPC⁺ (K) cells.

We have also revised the result section in the manuscript, as shown below.

“CLIC4 has been predicted as a direct target gene of the DREAM complex¹⁵ and PCLAF², and various DREAM components were recognized to bind the proximal promoter of CLIC4 (Supplementary Fig. 14B). Thus, we tested whether PCLAF directly transactivates CLIC4 by chromatin immunoprecipitation (ChIP) assay. Indeed, PCLAF bound to the proximal promoter of CLIC4 (Fig. 5C, and Supplementary Fig. 14C).”

A7. “I would strongly recommend the authors to repeat the Phenelzine effects on a human ex vivo model, such as precision-cut lung slices to provide the human perspective, as well. Does Phenelzine treatment enhance regeneration and AT2 differentiation?”

We appreciate the highly insightful comment. We strongly agree that using human-related model systems may better represent the pathophysiology of human pulmonary fibrosis. Moreover, employing more than one model system would be preferable.

Utilizing PCLS (Precision-Cut Lung Slices) could serve as a promising model system for evaluating the impact of phenelzine on human lung regeneration. Unfortunately, to the best of our knowledge, the assessment of regeneration using PCLS as a model system has not yet been fully established. Therefore, we are afraid that it is not feasible to test the impact of phenelzine on lung fibrosis or AT2 cell differentiation using the PCLS. In addition, while a lung fibrosis model has been developed through the administration of a fibrotic cytokine cocktail¹⁶, this model fails to fully replicate the pathophysiology of damage-induced lung fibrosis, including the regenerative processes. Recently, we have submitted a new grant proposal including the following topic: “Establishing a damage-induced lung regeneration and fibrosis model using PCLS and assessing the efficacy of phenelzine.” This proposed project will serve as our upcoming endeavor in this field.

We would like to highlight that our study suggests that manipulating cell plasticity suppresses fibrosis and promotes regeneration, laying a novel foundation for developing new therapies for pulmonary fibrosis or other lung diseases.

We added the PCLS and the information mentioned above in the Discussion accordingly. Again, thank you for your intuitive comments.

A8. “The transcriptomics section in the Methods severely lacks detail. The authors should revise accordingly to include missing aspects (e.g. data integration, combined signature enrichment, CLUE platform, etc.) of their analysis. Furthermore, following the code availability statement, I was not able to locate the data on github.”

We sincerely apologize for the poor description of the method. We have added detailed methods in the revised manuscript. Additional information regarding gene set scoring and drug identification was also included, as shown below.

“Gene set expression score analysis

The scores of gene sets were analyzed using the “AddModuleScore” function of Seurat. The SMAD3 target genes^{44,45}, DREAM-target genes³⁹, and SOX9⁺ progenitor-related genes³⁸ were utilized for

module score analysis. A detailed list of genes for each module score was listed in Supplementary Table 9.”

“Drug identification for mimicking of PCLAF-mediated transcriptome

CLUE platform (<https://clue.io/>) containing L1000-based Connectivity Map ⁵⁰, was used for drug identification. We identified drug candidates using the three gene lists: (a) genes highly elevated in *Pclaf* WT PAPCs (compared to *Pclaf* KO PAPCs), (b) genes specifically downregulated in human LUAD cells, H1792 cells, by shPCLAF (compared to control shRNA), and (c) genes decreased genes in mouse LUAD cells, KP-1 cells, by shPclaf (compared to control shRNA). We selected each drug candidate above a connectivity score of 1.5 and isolated PAPC-specific drugs using a Venn diagram.”

We have also made the data on GitHub publicly available (https://github.com/jaeilparklab/Pclaf_lung_regeneration).

A9. “Again, in the Discussion, the authors should comment more on the mechanism via which PCLAF-DREAM regulates *CLIC4*-TGF- β and alveolar regeneration. Currently, this aspect is underrepresented. Additionally, the authors did not discuss the outcome of the Phenezine treatment experiments as a potential treatment for pulmonary fibrosis.”

We appreciate the helpful comments. As kindly suggested, we revised the Discussion by adding the mechanisms of how PCLAF-DREAM modulates alveolar regeneration and by highlighting the unexpected role of the PCLAF-DREAM axis in activating TGF- β signaling via *CLIC4* transactivation (Fig. R5). In addition, we further discussed the potential significance of phenelzine and manipulating cell plasticity for pulmonary fibrosis treatment. Again, thank you for your constructive suggestions.

We have also revised the Discussion section in the manuscript, as shown below.

“TGF- β signaling has been proposed as one of the driving factors for lung fibrosis ⁵⁷ by promoting the epithelial-mesenchymal transition of AT2 cells ⁵⁸. Other studies suggest that TGF- β signaling is required for alveolar cell plasticity ⁴¹ and alveolar regeneration ⁴². Our results showed that TGF- β signaling is necessary for AT1 cell regeneration in the context of PAPCs. In *Pclaf* KO lung, p-SMAD3 was significantly decreased in regenerative alveoli lesions (Fig. 5E, F) but elevated in inflamed fibrotic lesions (Supplementary Fig. 17). In line with this, SMAD3-target genes were downregulated in IPF PAPCs while upregulated in IPF fibroblasts compared to normal lungs (Fig. 5N, O, and Supplementary Fig. 18). Thus, it is highly likely that TGF- β signaling spatiotemporally contributes to lung fibrosis or regeneration, depending on cell types or lesions. For the mechanism of TGF- β signaling activation, we notified *CLIC4*, one of the PCLAF-DREAM target genes ^{20,40} and a positive regulator for TGF- β signaling ⁴³. Indeed, PCLAF is directly bound to *CLIC4* proximal promoters, and ectopic expression of *CLIC4* markedly rescued p-SMAD3 levels in *Pclaf* KO LOs (Fig.5C and Supplementary Fig. 14C).

The current therapeutic strategy for lung fibrosis has focused on inhibiting fibroblasts using pirfenidone and nintedanib ⁵⁹. Recent studies suggested that failure of lung regeneration is one of the fundamental pathogenesis of lung fibrosis ^{7,9-13}. Intriguingly, disrupted AT2 cell lineage plasticity by *Pclaf* KO drove lung fibrosis rather than lung regeneration (Fig. 2). Based on our findings, we identified PCLAF-activation-mimicking drugs and tested their therapeutic potential on lung regeneration in vivo and in vitro (Fig. 6). This strategy, which facilitates the repopulation of functional lung epithelial cells, may be an alternative regimen or preventive or therapeutic measures for patients with potential lung fibrosis, such as those caused by thoracic radiotherapy in lung cancer patients.”

Minor comments

A10. “Fig. 1G: please stretch the panel, it is difficult to evaluate the gene expression differences.”

We apologize for the previous oversight regarding the visibility of the shrunk images. We acknowledge your recommendation and have accordingly revised Figure 1G.

A11. “The contrast in Fig. 1 K is not optimal and the data cannot be properly evaluated. Since the mouse strain is not published anywhere else, the authors must provide the validation together with the strategy in a supplementary figure for the interested reader.”

We apologize for the poor contrast. We revised Figure 1K in the revised manuscript.

As suggested, we have included the overall strategy for generating *Pclaf-LacZ* mice in the revised manuscript (Fig. R6A). We previously observed that PCLAF is expressed in the stem and progenitor cells, the crypt base columnar cells (CBCs), of the small intestine¹. Consistently, we performed X-gal staining and found positive staining in the CBCs (Fig. R6B). Unfortunately, the PCLAF antibody we previously used (ab56773) has been discontinued, and the replacement antibody (ab226255) did exhibit reactivity towards human PCLAF but not mouse PCLAF. Consequently, for the revised study, we relied on X-gal staining to detect PCLAF expression. PCLAF was expressed in epithelial, immune, and mesenchymal cells (Fig. R6C).

Figure.1

These new data were included in the revised manuscript (Supplementary Fig. 2).

We have also revised the Result section in the manuscript, as shown below.

“In addition, *Pclaf-lacZ* knock-in mice validated by LacZ expression in the intestinal crypt base columnar cells as previously identified¹⁹ (Supplementary Fig. 2A, B) displayed a similar increase in PCLAF⁺ cells in the regenerating lungs (Fig. 1K-M). Of note, PCLAF⁺ cells were also found in the immune and mesenchymal cells (Supplementary Fig. 2C-F).”

Figure R6. Establishment and validation of *Pclaf-LacZ* mice

A, Experimental scheme of generating *Pclaf-LacZ* mice.

B, Representative image of X-gal staining of small intestine (a positive control). Consistent with our previous reports ¹, *Pclaf* was expressed in crypt base columnar cells.

C, *Pclaf-lacZ* mice were instilled with bleomycin (1.4 U/kg). After 7 dpi, lungs were collected and performed whole-mount staining for X-gal. Each X-gal-stained slide were chemically immuno-stained for CDH1 (epithelial cells), CD45 (immune cells), CD44 (mesenchymal cells), or FN (fibronectin; mesenchymal cells). Representative images are shown.

A12. “The PBS WT/KO control is missing from Fig. 2D. Please add.”

We apologize for the confusion caused by the incorrect labeling of 0 dpi. As per your clarification, we have revised the labeling of the revised manuscript (Fig. 2E, F). The correct labeling now indicates that 0 dpi represents the PBS control samples, collected at the same time as the 21 dpi samples.

A13. “What was the rationale for the two experimental designs in lung organoids? Why did the authors prompt for the intermediate and withdrawal models? This was not very clear in the manuscript, especially given the fact that they may yield contradictory results (e.g. Fig. S11).”

We apologize for not providing sufficient information regarding the rationale of experimental designs.

Rationale 1: Previous studies have shown that TGF- β signaling plays a dual role in the trans-differentiation of AT2 cells into AT1 cells for lung regeneration¹⁷ and in driving EMT of AT2 cells for lung fibrosis¹⁸. Unlike the binary modes of simple signal circuits, e.g., On and Off switch, many signaling pathways elicit various outcomes depending on the spatiotemporal and dosage of signaling cues¹⁹⁻²¹. For example, intermittent or continuous stimulation of parathyroid hormone (PTH) can result in either increased or decreased bone mass, respectively²². Moreover, different dosages of cytokines induce different types of differentiated cells from embryonic stem cells²³. Therefore, given the biphasic functions of TGF- β signaling in lung regeneration/fibrosis, we hypothesized that the dosage and timing of TGF- β signaling activation are crucial for promoting lung regeneration via the trans-differentiation of AT2 cells into AT1 cells.

Rationale 2: The balance between proliferation and differentiation is critical in organoid culture in general. For instance, many organoids (intestine, stomach, and esophagus) are cultured in two different conditions: organoid maintenance (cell proliferation) and organoid differentiation (cell differentiation). Moreover, it is noteworthy that TGF- β signaling has been shown to inhibit the proliferation of alveolar cells¹⁷. Thus, we thought that culture of the LOs under TGF- β 1 stimulation during the whole culture period severely inhibits lung alveolar cell proliferation, generating no organoid formation.

Based on the rationale mentioned above, we tested two different stimulation models: intermittent (INT; two days with TGF- β 1 + two days without TGF- β 1) and withdrawal (WD: one-week treatment followed by one-week withdrawal of TGF- β 1) administration of TGF- β 1 ligand to LO culture. The INT model aimed to continuously alter the organoid culture condition between TGF- β 1-mediated cell differentiation and proliferation, while the WD model aimed to provide continuous TGF- β 1 stimulation at the early stage of LO culture. The INT model exhibited increased AT1 differentiation, which rescues *Pclaf* KO-depleted AT1 cells. However, the WD model displayed complete inhibition of AT1 cell differentiation regardless of *Pclaf* WT or KO (Supplementary Fig. 15), which led us also to test the impact of TGF- β 1 on the late stage of LO culture. Interestingly, the treatment of LOs with TGF- β 1 at a later time (on day 10 or day 12 of a total of 14 days of culture) showed increased AT1 differentiation (Fig. 5J-M).

These results suggest that, like other developmental signaling pathways, temporal regulation of TGF- β signaling is crucial for AT2 to AT1 cell differentiation.

We added this information and results to Supplementary Figure 16.

“Of note, this experiment was designated by two rationales. First, signaling pathways elicit various outcomes depending on the spatiotemporal and dosage of signaling cues²⁻⁴. For example, intermittent or continuous stimulation of parathyroid hormone (PTH) can result in either increased or decreased bone mass, respectively⁵. Second, the balance between proliferation and differentiation is critical in organoid culture in general, and TGF- β signaling has been shown to inhibit the proliferation of alveolar cells⁶. The INT model aimed to continuously alter the organoid culture condition between TGF- β 1-mediated cell differentiation and proliferation, while the WD model aimed to provide continuous TGF- β 1 stimulation at the early stage of LO culture. LO culture under

TGF- β 1 stimulation during the whole period of culture was avoided due to the possibility of severe proliferating inhibition.”

A14. “Overall, the authors should consider better color codes in their figures. The default hue color code is not optimal for labeling several cell types on UMAP plots.”

As suggested, we revised all the UMAP plots with the polychrome colors using the scCustomize packages (Fig. 1, 3).

Figure.1

Figure.3

A15. “Fig. 6: as stated elsewhere, since the focus of the manuscript is the alveolar regeneration, I do not follow panels C-D bottom.”

A previous study showed that the bronchiolar cells, especially, the club cells, also contribute to alveolar regeneration in part ⁸. Thus, we tested whether phenelzine could also affect bronchiolar cell plasticity, even if the cell plasticity of bronchiolar cells is out of the focus of our study.

We have revised the result section in the manuscript, as shown below.

“ Additionally, phenelzine altered the cell plasticity of bronchiolar cells (Fig. 6E), which partly contributes to alveolar regeneration ³⁹, resulting in the reduced numbers of ciliated and elevated club cells in LOs (Fig. 6E, F).”

We sincerely appreciate your insightful comments.

Reviewer 2

"This manuscript addresses an important problem, the mechanism of lung regeneration and fibrosis. However, the authors use a global knockout mouse to make conclusions about PCLAF function in epithelial cells, though it is expressed in other cell types. The methods used to measure proliferation and AT1 transition are problematic. PCLAF induces the cell cycle. The most convincing data is that PCLAF is necessary for AT2 cell proliferation in the organoids, resulting in enhanced CFE. The quantitation of AT2 and AT1 cell numbers is not convincing and the conclusions about the role of PCLAF in AT1 transition is based on this quantitation as well as pseudotime data. The organoid data is helpful to exclude the role of PCLAF in other cell types; however, the dramatic effect of PCLAF on the CFE confounds interpretation of AT2 to AT1 transition data. The therapeutic potential of PCLAF mimicking drugs is a notable strength."

Major Comments

B1. "Fig. 11-L: Other cell types besides epithelial cells proliferate in this model. Costaining with epithelial cell markers should be performed. Representative images and quantitation of PCLAF+ epithelial cells should be shown. Sections should be costained for immune and mesenchymal markers, and PCLAF expression in these cells should be shown."

We appreciate your understanding and recognition of the diverse expression of PCLAF across various cell types. As you correctly mentioned, our findings demonstrate that PCLAF is expressed not only in epithelial cells but also in immune cells and fibroblasts (Fig. R6). However, the observed phenotype in *Pclaf* KO mice and the results from lung organoids lacking immune cells and fibroblasts indicate that the role of the PCLAF-DREAM axis in lung epithelial cell plasticity operates independently of immune cells and fibroblasts.

Nevertheless, we acknowledge the limitations of our study. In the revised manuscript, we discuss these limitations, specifically addressing the need for further investigations to fully understand the involvement of immune cells and fibroblasts in the processes regulated by the PCLAF-DREAM axis.

Additionally, we have made the necessary revisions to ensure clarity and scientific integrity in the result and discussion of the revised manuscript.

"Of note, PCLAF+ cells were also found in the immune and mesenchymal cells (Supplementary Fig. 2C-F)."

"Using *Pclaf-LacZ* mice, we observed that PCLAF was also expressed in non-epithelial cells such as immune cells and fibroblasts (Supplementary Fig. 2). *Pclaf* KO mice showed slight alteration of immune cell status⁶⁰. Since immune cells⁶¹ and fibroblasts^{61,62}

Figure R6. Establishment and validation of *Pclaf-LacZ* mice
A, Experimental scheme of generating *Pclaf-LacZ* mice.
B, Representative image of X-gal staining of small intestine (a positive control). Consistent with our previous reports¹, *Pclaf* was expressed in crypt base columnar cells.
C, *Pclaf-LacZ* mice were instilled with bleomycin (1.4 U/kg). After 7 dpi, lungs were collected and performed whole-mount staining for X-gal. Each X-gal-stained slide were immuno-stained for CDH1 (epithelial cells), CD45 (immune cells), CD44 (mesenchymal cells), or FN (fibronectin; mesenchymal cells). Representative images are shown.

play an essential role in lung regeneration, the impact of *Pclaf* KO on these cells might also contribute to the lung regeneration phenotype. However, consistent with the *Pclaf* KO mouse and LO phenotypes, *Pclaf* cKO in AT2 cells also showed impaired regeneration of AT1 cells from AT2 cells (Fig. 3, Supplementary Fig. 10). Therefore, it is highly likely that the PCLAF-DREAM axis-driven lung epithelial cell plasticity is independent of immune cells and fibroblasts. Nonetheless, our finding in murine systems remains to be tested in a human-relevant system, such as human lung organoids.”

B2. “A major point of this manuscript is the effect of PCLAF on AT2 cell proliferation and the transition to AT1 cells. However, the data in Fig. 2C D are problematic. First, the RAGE staining at 0 dpi does not appear to be specific for AT1 cells, as there are large, round RAGE+ entities. An enlarged image should be shown to convince that the RAGE staining is working. More importantly, there are concerns about the quantitative data. Methods for the quantitation in Fig. 2D should be explained. Specifically, the denominator for the percent calculation should be stated. If the denominator is total cells (DAPI+), this is problematic because the injured lungs have many more cells due to immune cell recruitment to the lung, which will artificially decrease the percent of epithelial cells. In addition, it is not possible to count the number of RAGE+ cells since individual cells cannot be identified. Many other papers have demonstrated that there are less AT1 cells at 21 dpi than at 0 dpi in WT lungs in the bleomycin model, so the quantitation performed here is highly inconsistent with established literature. A marker of S phase may be helpful. Lineage tracing should be performed because the number of AT1 cells, even if appropriately quantified, may reflect a role of PCLAF in cell death instead of regeneration.”

We appreciate your critical comments. We acknowledge that the large and round entities stained positive for RAGE may have resulted from low magnification or non-specific staining. To address this concern, we conducted additional high-magnification imaging to ensure accurate visualization of RAGE staining.

Regarding the quantification of AT1 cells, we agree that our previous approach of calculating the percentage of AT1 cells based on their connection to DAPI staining²⁴⁻²⁷ may not provide the most accurate results. We also recognize the importance of considering the proportions of epithelial cells within all cell types to effectively compare AT1 cells. Therefore, in the revised manuscript, we adopted a new approach for analyzing AT1 regeneration. We measured the ratio of the RAGE-stained area to the CDH1 (E-cadherin)-stained area, using high magnification images focused on regions abundant in epithelial cells. As shown in Figure R6A, the RAGE staining was performed effectively, and the ratio of RAGE to CDH1 demonstrated that AT1 cells were not fully restored compared to the PBS control (Fig. R7A and B). Furthermore, the regeneration of AT1 cells was significantly decreased in *Pclaf* KO lungs at 7 and 21 dpi compared to WT (Fig. R7A and B).

For the analysis of cell death, we performed double staining of Cleaved-CASPASE-3 (CC3) and CDH1 to assess apoptotic epithelial cells. Interestingly, we did not observe any differences in apoptotic epithelial cells between WT and KO at 3 dpi, implying that *Pclaf* KO does not affect bleomycin-mediated cell death (Fig. R7C and D).

In this study, our focus is to highlight that the PCLAF-DREAM axis regulates alveolar cell plasticity during lung regeneration rather than activating or promoting the proliferation of progenitor cells. Although PCLAF has previously been identified as a mitogen through its binding with PCNA^{28,29}, the functions of PCLAF are context-dependent. In our previous studies, we found that PCLAF is required to maintain the stemness of intestinal stem cells¹. and breast cancer cells³⁰ without influencing cell proliferation in a PCNA-independent manner. Interestingly, *Pclaf* KO lungs after bleomycin damage showed an increased population of MKI67+ epithelial cells (Fig. R7E and F). Additionally, *Pclaf* KO lung organoids exhibited an increase in MKI67+ cells (Fig. R7G and H). These data suggest that PCLAF plays a role in controlling cell plasticity rather than cell proliferation during lung regeneration.

Additionally, we also performed staining of RAGE and CDH1 using lung tissue sections from mice under DMSO or phenelzine treatment, with a high dose of bleomycin instillation. The regeneration of AT1 cells was significantly increased in phenelzine-treated lungs at 13 dpi compared to DMSO-treated lungs (Fig. R7I and J).

Moreover, **the lineage tracing experiments** showed that Both *Pclaf* KO and *Pclaf* cKO mice showed that AT1 cells and KRT8+ cells from AT2 cells were decreased compared to *Pclaf* WT during lung regeneration (Fig. R3). These data consistently indicate that PCLAF is crucial in alveolar cell plasticity, especially for regeneration of AT1 cells.

We have included these new data in the revised manuscript (Fig. 2C, E, Fig.3G, Fig. 6I, J, Supplementary Fig. 3B-F, and Supplementary Fig. 5F, G, Supplementary Fig. 10).

Thank you for your valuable input, and we have incorporated these clarifications and explanations into the revised manuscript.

“We initially hypothesized that PCLAF is involved in activating lung progenitor cells based on our previous finding that PCLAF drives cell quiescence exit²⁰. Unexpectedly, *Pclaf* KO lung showed increased proliferating epithelial cells at the regeneration stage (7 dpi, Supplementary Fig. 3B, C). Cell apoptosis was not altered by *Pclaf* KO at the acutely damaged stage (3 dpi, Supplementary Fig. 3D, E).”

Figure R7. *Pclaf* depletion inhibits damage-induced regeneration of AT1 cells.

A-F, *Pclaf* WT and KO mice were treated with phosphate-buffered saline (PBS; control) or bleomycin by intratracheal instillation. At 3, 7, or 21 dpi of bleomycin, or 21 dpi of PBS, the lung tissues were collected for further analysis. Representative images of lungs that fluorescently immunostained for RAGE and CDH1 at indicated time points (A). Quantification of the ratio of RAGE⁺ area / CDH1⁺ area (B). Representative images of lungs that fluorescently immunostained for CC3 (cleaved-caspase-3) and CDH1 at indicated time points (C). Quantification of CC3⁺ CDH1⁺ cells (D). Representative images of lungs that fluorescently immunostained for MKI67 and CDH1 at indicated time points (E). Quantification of MKI67⁺ CDH1⁺ cells (F).

G, H, Representative images of LOs derived from *Pclaf* WT or *Pclaf* KO lung epithelial cells, that fluorescently immunostained for MKI67 (G) and quantification of MKI67⁺ cells (H).

I, J, Mice were treated with bleomycin (2.8 U/kg) by intratracheal instillation. The vehicle control (DMSO, n = 10) or phenelzine (n = 10; 750 μg/head) were administered by intraperitoneal injection at -1, 1, and 3 dpi. At 13 dpi (n = 4 for each group), lungs were collected for further analysis. Representative images of lungs that fluorescently immunostained for RAGE and CDH1 at indicated time points (I). Quantification of the ratio of RAGE⁺ area / CDH1⁺ area (J).

B3. “In Fig. 2E-G, the authors claim to show that KO mice have more fibrosis. The data are fairly convincing, but 0 dpi data should be added to the quantitation in 2H. The time point for the data in 2H should be stated; if it is 21 dpi, the amount of fibrosis in WT mice appears to be very low compared to the literature. Hydroxyproline assay is a less biased method to measure the degree of fibrosis than staining and image analysis. Finally, since a global KO is used, the authors must demonstrate that PCLAF is not expressed in other cell types, particularly fibroblasts and immune cells. Review of the dataset that the authors used shows that *Pclaf* (2810417H13Rik) is highly expressed in immune cells. To definitively support their claim, a conditional knockout in epithelial progenitors and lineage tracing should be performed.”

We appreciate your consideration and the feedback provided. We understand the importance of including PBS (0 dpi) data in the quantification of picosirius red staining, and we apologize for the oversight. We have now added the PBS data for picosirius red area quantification (Fig. R8A).

Additionally, we conducted another round of experiments involving bleomycin-induced lung fibrosis in both *Pclaf* WT and *Pclaf* KO mice. At 25 dpi, we performed a hydroxyproline assay to assess the extent of fibrosis. Consistently, *Pclaf* KO lungs exhibited significantly increased hydroxyproline by bleomycin instillation (Fig. R8B).

Regarding the inconsistency of lung fibrosis observed in our study compared to the existing literature, we acknowledge that there may be experimental variations in the bleomycin-induced lung fibrosis model. As part of our study optimization, we conducted pilot experiments to determine the most suitable model for assessing lung regeneration. To accomplish this, we assessed the functional lung status using oximetry across various concentrations of bleomycin that had been used in previous studies. The results of the oximetry analysis demonstrated that mice treated with 1.4U/kg of bleomycin showed full restoration of lung function by 14 dpi (Fig. R8C). Therefore, we selected a dose of 1.4U/kg of bleomycin for our regeneration model.

On the other hand, we found that a dose of 2.8U/kg of bleomycin significantly increased lung fibrosis by 21 dpi, which is consistent with the findings reported in the literature (Fig. R8D and E). Consequently, we utilized a dose of 2.8U/kg of bleomycin for the lung fibrosis model, particularly when assessing the therapeutic efficacy of phenelzine.

Thank you for bringing up these points, and we have now addressed them by including the relevant data and explanations in the revised manuscript (Fig. 2H, J, L, Supplementary Fig. 3A, and Supplementary Fig. 19C, D).

Figure R8. *Pclaf* KO promotes damage-induced lung fibrosis.

- A,** Quantification of picrosirius red* area in the bleomycin or PBS (control) instilled lung at 21 dpi.
- B,** Quantification of hydroxyproline contents in the left lobe of bleomycin or PBS instilled lung at 25 dpi.
- C,** The dynamics of SpO₂ levels measured by pulse-oximetry at the indicated time points after 1.4 U/kg, 2.8 U/kg, or 7 U/kg of bleomycin instillation.
- D,** Representative images of H&E staining of lungs at indicated time points after 2.8 U/kg of bleomycin instillation.
- E,** Representative images of picrosirius red staining of lungs at indicated time points after 2.8 U/kg of bleomycin instillation.

To better understand the role of PCLAF in alveolar cell plasticity, we performed lineage tracing experiments. First, we generated *Pclaf-fl/fl* mice using *Pclaf-LacZ-neo* mice followed by confirming Cre-mediated knock-out by genotyping of *Pclaf-fl/fl* mouse embryonic fibroblasts by Ad-Cre (Fig. R3A, B). After that, we performed lineage-tracing of AT2 cells using *Sftpc-CreERT2; LSL-Sun1-EGFP* mice combined with *Pclaf* WT, *Pclaf* KO, or *Pclaf-fl/fl* (*Pclaf* cKO) mice. Both *Pclaf* KO and *Pclaf* cKO mice showed that AT1 cells and KRT8+ cells from AT2 cells were decreased compared to *Pclaf* WT during lung regeneration (Fig. R3C-F). These data consistently indicate that PCLAF is crucial in alveolar cell plasticity, especially for regeneration of AT1 cells.

We have revised the result section in the manuscript.

“Next, to determine the role of PCLAF in lung regeneration, we tested several concentrations of bleomycin (1.4 U/kg, 2.8 U/kg, and 7 U/kg) in the lung and monitored blood oxygen levels (peripheral oxygen saturation [SpO₂]). Mice with a lower concentration of bleomycin (1.4 U/kg) displayed a restoration of SpO₂ levels at 14 dpi, whereas those with 2.8 U/kg and 7 U/kg exhibited a decrease in SpO₂ levels until 14 dpi (Supplementary Fig. 3A). Thus, we administered a low dose of bleomycin (1.4 U/kg) to *Pclaf* wild-type (WT) and knock-out (KO) mice to examine lung regeneration (Fig. 2A).”

“ Unlike WT lung tissues, *Pclaf* KO lungs exhibited more inflamed and condensed tissue (Fig. 2E), with severe fibrotic features confirmed by picrosirius red (a dye staining collagen), α SMA/ACTA2 (a marker for myofibroblasts) staining, and hydroxyproline assay (exclusively in collagen) (Fig. 2F-J). These results suggest that *Pclaf* is indispensable for bleomycin-induced lung regeneration.”

“For experimental validation of the predicted cell lineage trajectories, we also performed a cell lineage tracing analysis using genetically engineered mice. We first established *Pclaf-fl* (*floxed*)/*fl* mice for conditional KO (cKO) of *Pclaf* alleles (Supplementary Fig. 10A, B). AT2 cell-specific Cre driver (*Sftpc*^{CreERT2} [*Sftpc-CreERT2*]) and Cre-loxP recombination reporter (*Rosa26*^{Sun1GFP} [*Sun1-GFP*]) were combined with either *Pclaf* *-/-* (germline KO) or *Pclaf-fl/fl* (for conditional KO [cKO]) strains. Then, bleomycin was instilled into the lung, followed by tamoxifen administration for genetic labeling and lineage tracing of cells derived from AT2 cells (*Sftpc*-Cre-driven GFP+ cells). Consistent with cell lineage trajectory inference and immunostaining results (Fig. 3C-F, Supplementary Fig. 10), AT1 cells derived from AT2 cells were significantly reduced and AT2 cells were elevated in both *Pclaf* KO and cKO lung tissues compared to *Pclaf* WT (Fig. 3G, and Supplementary Fig. 10C-E). Additionally, RAGE and SPC double-positive cells (AT2^{med}/AT1^{med} cells) derived from AT2 cells were significantly decreased in *Pclaf* cKO lung, and KRT8+ cells were lessened in both *Pclaf* KO and cKO lung (Fig. 3G, and Supplementary Fig. 10C-E). In line with these results, the CytoTRACE analysis¹³ showed a relatively less differentiated cell status in activated AT2 cells and transitioning AT1^{med}/AT2^{med} cells of *Pclaf* KO lungs than of WT lungs, indicating the impaired maturation of these cells into AT1 cells in the condition of *Pclaf* ablation (Fig. 3H). These results suggest that PCLAF plays a pivotal role in alveolar cell lineage plasticity that induces AT1 cell regeneration from AT2 cells during lung regeneration.”

Figure R3. *Pclaf* KO inhibits differentiation of AT1 cells from AT2 cells.

A, Experimental scheme of generating of *Pclaf*-fl/fl mice.

B, Mouse embryonic fibroblasts (MEFs) were isolated from *Pclaf*-fl/fl mice. 1 X 10⁷ or 1 X 10⁸ PFU of Ad-Cre were introduced in *Pclaf*-fl/fl MEFs. Conditional knock-out by Ad-Cre were checked by genotyping.

C-F *Sftpc*-CreERT2; LSL-Sun1-EGFP; *Pclaf* WT, *Sftpc*-CreERT2; LSL-Sun1-EGFP; *Pclaf* KO, and *Sftpc*-CreERT2; LSL-Sun1-EGFP; *Pclaf*-fl/fl mice were instilled with bleomycin Then mice were injected with tamoxifen for 5 consecutive days. At 14 dpi, lungs were collected and analyzed.

C, Representative images of lungs; immunostaining for RAGE, SPC, and EGFP. White arrow; RAGE⁺/EGFP⁺ cell. Red arrow; SPC⁺/EGFP⁺ cell. Yellow arrow; RAGE⁺/SPC⁺/EGFP⁺ cell.

D, Enlarged image of RAGE and SPC double positive cell.

E, Representative images of lungs; immunostaining for KRT8 and EGFP. White arrow; KRT8⁺/EGFP⁺ cell.

F, Quantification of indicated cells in figure R3C and E.

B4. “The immunostaining in Fig. S4 is unclear. High resolution images and isotype control staining should be shown. It is concerning that there are no SPC+ cells in the WT organoids. The impaired colony forming efficiency shown in Fig. S4 is dramatic but not surprising since the function of proliferating cell nuclear antigen-associated factor (PCLAF) is to promote cell proliferation. Moreover, the dramatic inability to form organoids confounds the AT1 transition data. It is difficult to draw conclusions about AT1 cell transition from the limited remaining organoids.”

We appreciate your comments on the image quality and the steps taken to address it.

Immunofluorescence staining:

It is understood that the visibility of SPC+ (green) cells in WT organoids was hindered by excessive red fluorescence. To improve image quality and quantification, we have repeated staining using an anti-HOPX antibody instead of anti-RAGE antibody. Additionally, as kindly suggested, we have included isotype control staining images and high-resolution images (Fig. R9A-C). These new data were added to the revised manuscript (Supplementary Fig. 5).

Impact of PCLAF depletion on cell proliferation vs. cell differentiation/ plasticity:

We agree that *Pclaf* KO-reduced organoid forming efficiency could confound our interpretation that *Pclaf* KO impairs AT2-to-AT1 cell differentiation. Nonetheless,

Figure R9. *Pclaf* KO inhibits alveolar organoid formation.

A, Representative images of LOs immunostained for HOPX, SPC, SCGB1A1, Ac-TUB, and each isotype control. a; alveolar, b; bronchiolar; ba; bronchioalveolar. LOs were stained using serial sections.

B, Quantification of organoid types. #1 and #2 represents each independent experiment.

C, High-resolution images of each organoid type using the stained images shown in Figure R8A.

- Consistent with the results from organoids, *Pclaf* KO also impairs AT2-to-AT1 cell differentiation in mice (Fig. 2).
- Pclaf* KO-impaired AT2-to-AT1 cell differentiation is **rescued** by Harmine, Lin52^{S28A}, Clic4, or TGF- β 1 even in organoids (Fig. 4, 5).
- Several pieces of evidence indicate that organoid-forming efficiency is not solely dependent on cell proliferation³¹. In our previous study, we observed a significant decrease in the organoid-forming efficiency of *Pclaf* KO intestinal stem cells without affecting cell proliferation¹.
- PCNA-independent roles of PCLAF:** As a leading group in the PCLAF study, we previously have revealed that PCLAF hyperactivates WNT/ β -catenin signaling independently of PCNA in colorectal cancer^{1,32}, promotes pancreatic tumorigenesis via MAPK signaling³³, induces cell plasticity and increases cell stemness of breast cancer³⁰, facilitates stem cell activation and proliferation for intestinal regeneration¹, and promotes lung tumorigenesis via the DREAM complex². PCLAF directly

binds to EZH2 to hyperactivate β -catenin-mediated gene transactivation independently of PCNA binding. Similarly, PCLAF binds to RBBP4/7, core components of the DREAM complex, and inhibits the recruitment of RBL2/p130 to the DREAM/MuvB complex, which is also independent of PCNA binding². We also quantified the amount of PCLAF protein in HeLa cells. Although most PCLAF protein shows nuclear localization, approximately 50% of PCLAF binds to PCNA with a one-to-one molar ratio. However, the other 50% of PCLAF protein is not associated with PCNA but with EZH2 or nuclear proteins³². **Therefore, PCLAF is not exclusively associated with PCNA or cell proliferation.**

(e) As described in the initially submitted manuscript, *Pclaf* KO lung organoids showed an elevated number of MKI67+ cells in organoids compared to *Pclaf* WT, whereas *Pclaf* KO lung organoids exhibited a significant reduction in organoid-forming efficiency compared to WT organoids (Fig. R7G, H).

Consistently, immunostaining of *Pclaf* WT and KO lung tissues exhibited similar results (Fig. R7E, F).

Therefore, it is reasonable to conclude that PCLAF plays a role in promoting alveolar cell plasticity. However, our results do not fully exclude any involvement of PCLAF-mediated cell cycle reentry or cell proliferation. These new data and interpretations were added to the revised manuscript (Supplementary Fig. 3, 5).

Figure R7. Pclaf depletion inhibits damage-induced regeneration of AT1 cells.

E, F, *Pclaf* WT and KO mice were treated with phosphate-buffered saline (PBS; control) or bleomycin by intratracheal instillation. At 3, 7, or 21 dpi of bleomycin, or 21 dpi of PBS, the lung tissues were collected for further analysis. Representative images of lungs that fluorescently immunostained for MKI67 and CDH1 at indicated time points (E). Quantification of MKI67⁺ CDH1⁺ cells (F).

G, H, Representative images of LOs derived from *Pclaf* WT or *Pclaf* KO lung epithelial cells, that fluorescently immunostained for MKI67 (G) and quantification of MKI67⁺ cells (H).

Again, we appreciate the insightful comments.

B5. “scRNAseq cannot be used to quantify cell numbers (Fig. 3) since enzymatic dissociation results in cell loss, particularly of AT1 cells. Pseudotime analysis is predictive but lineage tracing is needed to validate the predictions.”

We appreciate your critical comments. We fully understand your concern, and we agree with that. Thus, we removed cell proportion analysis using the scRNA-seq dataset. Instead, we performed immune staining using specific markers for each cell type, such as SPC (AT2 cell), RAGE (AT1 cell), LCN2 (Activated AT2 cell), and KRT8 (KRT8+ cell). Unfortunately, there are no specific markers for AT1med/AT2med cells. As shown in Figure R10, *Pclaf*-deficient decreased AT1 cells and elevated AT2 cells upon bleomycin damage (Fig. R10A, B, E). Intriguingly, *Pclaf* KO significantly increased the number of KRT8+ cells during lung regeneration compared to WT, while activated AT2 cells were not altered (Fig. R10C, D, E). Considering that KRT8+ cells are intermediate cells from AT2 into AT1 cells

7,12, the increased number of KRT8+ cells by *Pclaf* KO indicates that *Pclaf* is required for final differentiation into AT1 cells. Moreover, unaltered numbers of activated AT2 cells by *Pclaf* KO indicate that the role of *Pclaf* in AT2 activation may not exist or can be compensated.

Figure R10. *Pclaf* depletion inhibits AT1 cells and elevates AT2 cells and KRT8+ cells.

A-E, *Pclaf* WT and KO mice were treated with phosphate-buffered saline (PBS; control) or bleomycin by intratracheal instillation. At 7 dpi of bleomycin the lung tissues were collected for further analysis. Representative images of lungs that fluorescently immunostained for RAGE and CDH1 (A), RAGE and SPC (B), and KRT8 and CDH1 (C), and that chemically immunostained for LCN2 (D). Quantification of relative cell types (vs. *Pclaf* WT).

As we described above, lineage tracing experiments of AT2 cells using *Sftpc-CreERT2* showed that both *Pclaf* KO and *Pclaf* cKO inhibited the regeneration of AT1 cells from AT2 cells compared to *Pclaf* WT (Fig. R3). Two predictive analyses by RNA-velocity and Slingshot and lineage tracing results consistently showed that the PCLAF is required for proper regeneration of AT1 cells upon damage.

We have addressed these points by including the relevant data and explanations in the revised manuscript (Fig. 3F and Supplementary Fig. 7F).

We have revised the result section in the manuscript.

“Consistently, immunostaining of *Pclaf* KO lung tissues showed the marked decreased AT1 cells and increased AT2 cells (Fig. 3F), consistent with in silico results. In *Pclaf* KO lung, KRT8+ cells (intermediate cells) were significantly reduced, while activated AT2 cells (LCN2+) were not changed, compared to WT (Fig. 3F and Supplementary Fig. 7F). Of note is that PAPCs were highly enriched with SPC, a representative marker for AT2 cells (Supplementary Fig. 9), implying that most PAPCs are likely derived from AT2 cells.”

Minor Comments

B6. *“The Abstract should be edited for language. For example, there are several confusing phrases. “Orchestration” should be more explicitly defined. The difference between “cell quiescence exit” and “cell proliferation” is unclear. “Pulmonary proliferative cells” should be defined. The term “acutely” is nonspecific – does this mean rapidly? “Intriguingly” should be deleted. “AT1” should be “AT1 cells”. The abbreviation “PPCs” should be defined. The term “pharmacological mimicking” is unclear.”*

As kindly suggested, we revised the abstract. Briefly, the words “Orchestration”, “Acutely”, “Intriguingly”, and “Pharmacological mimicking” were removed. The “AT1” changed into “AT1 cells” and the “Cell quiescence exit” was revised into “Re-entering into cell cycle”. As described in the response to reviewer #1, we have changed the term PPC into PAPC (proliferating alveolar progenitor cell), and we removed PAPC from the abstract.

“Spatiotemporal control of stem/progenitor cells is essential for lung regeneration, the failure of which leads to lung disease. However, the mechanism of alveolar cell plasticity during regeneration remains elusive. We previously found that PCLAF remodels the DREAM complex for re-entering into the cell cycle. PCLAF is expressed explicitly in proliferating lung progenitor cells, along with the DREAM target genes by lung damage. Depletion of *Pclaf* inhibited alveolar type I (AT1) cell regeneration, inducing lung fibrosis. The single-cell transcriptome and organoid analyses showed that the PCLAF-DREAM complex increased a direct target gene, *CLIC4*, promoting TGF- β signaling, which regulates the balance of regeneration between AT1 and AT2 cells. Furthermore, a drug candidate was identified using a perturbation database and in silico dataset for lung regeneration. Our study unveils an unexpected role of the PCLAF-DREAM axis in controlling alveolar cell plasticity for lung regeneration and proposes a viable option for lung fibrosis prevention..”

B7. *“It’s unclear why in Fig. 1G the plots are different than in Fig. 1A since the data are the same. Either all the cell types listed in Fig. 1A should be identified on the 1G plots or, preferably, 1G should show *Pclaf* expression on the plots configured as in 1A.”*

We apologize for any confusion. The UMAP plots shown in Figure 1A and Figure 1G were generated using the same scRNA-seq dataset. We reduced the horizontal size for fitting in the figure. For clarity, we have revised Figure 1G by adjusting the aspect ratio for consistency.

Fig. 1

B8. *“The authors state that “Bleomycin initially damages the alveolar epithelium, including alveolar type I (AT1) and AT2 cells, followed by inflammation and interstitial fibrosis” and cite refs. 22-24. Data demonstrating that AT1 and AT2 cells are damaged prior to inflammation in this model are not immediately obvious in these references. These data should be explained or this claim should be deleted.”*

We appreciate your comments. We agree that the sequence of cell damage and inflammation in the bleomycin-induced lung regeneration model has not been definitively demonstrated. While it is generally believed that cell damage occurs prior to the inflammatory response in this model⁹⁻¹¹, there is limited direct evidence regarding the precise temporal sequence of these events caused by bleomycin. Therefore, we revised the sentence as shown below.

“Bleomycin damages the alveolar epithelium, including AT1 and AT2 cells, followed by interstitial fibrosis²⁸⁻³⁰”

B9. *“In Fig. S4, “#1” and “#2” should be defined.”*

Our apologies. #1 and #2 means two independent experiments (each n = 3). We have added detailed information in the revised manuscript (Supplementary Fig. 5).

We sincerely appreciate the valuable input, which has enhanced the scientific rigor and clarity of our research.

Reviewer 3

“In this manuscript, Kim et al., described a pivotal role for PCLAF-DREAM complex in alveolar regeneration. First the authors utilized published scRNA-seq datasets from regenerating mouse lungs and human normal and IPF lungs and show that proliferating cells are enriched for PCLAF expression. The authors went on to check PCLAF role in lung regenerating by using PCLAF KO mice and tested its role in control and bleomycin-induced injury models. The authors state that loss of PCLAF blocks AT2 to AT1 differentiation. The authors state that PCLAF knockout cells favored cell trajectory from proliferating AT2s to AT2 (*Slc34a2*^{high}) cells. This is in contrast to control cells where proliferating AT2s show trajectory towards AT1s. Mechanistically, the authors claim that PCLAF-DREAM activated *CLIC4-TGFβ* signaling is necessary for AT2 to AT1 differentiation. Further, the authors claim that pharmacological mimicking of PCLAF-DREAM signaling is sufficient to depress AT2 to AT1 differentiation blockade observed in PCLAF knock out lungs.

Overall, the authors attempt to address mechanisms that control the choice of AT2s proliferation and differentiation. I think the authors attempt to address an important question. But the data presented here falls short and does not support the claims. Authors drew many conclusions from scRNA-seq data but did not provide additional validity support for them. See below for detailed comments.”

Major comments

C1. “One of the main claims in this manuscript is that “PPC”s are stalled in *Pclaf* KO lungs. If PPC are stalled, then one would expect more PPCs in these lungs. However, the quantification data presented in Fig-3G doesn’t support this claim.”

We appreciate your valuable comments. We agree that we did not provide sufficient evidence for the characterization of stalled PAPCs (proliferating alveolar progenitor cells; revised from PPCs according to the comments from reviewer #1). To address this, we conducted co-immunostaining of lung tissues (*Pclaf* WT vs. KO; 7 dpi) for MKI67 and CDH1 and observed a significant increase in MKI67⁺ cells in *Pclaf* KO lungs at 7 dpi (Fig. R7E, F) and lung organoids (Fig. R7G, H). These results suggest that *Pclaf* deficiency leads to the stalling of PAPCs.

Please note that Figure 3G represents the results of **CytoTRACE**, which predicts the developmental potential of cells (cell differentiation vs. de-differentiation) and is not directly related to cell proliferation. We have revised the result section in the manuscript by adding the new results (Supplementary Fig. 3B, C, Supplementary Fig. 5F, G).

Fig. R7. *Pclaf*-deficiency inhibits damage-induced regeneration of AT1 cells.

(E-F) *Pclaf* WT and KO mice were treated with phosphate-buffered saline (PBS) or bleomycin by intratracheal instillation. At 3 dpi, 7 dpi, or 21 dpi of bleomycin, or 21 dpi of PBS (control), the lungs were collected for further analysis.

(E) Representative images of lungs that fluorescently immunostained for MKI67 and CDH1 at indicated time points.

(F) Quantification of MKI67⁺ CDH1⁺ cells.

(G) Representative images of LOs using *Pclaf* WT or *Pclaf* KO lung epithelial cells, that fluorescently immunostained for MKI67.

(H) Quantification of MKI67⁺ cells.

C2. “Along the same lines, the authors used scRNA-seq to quantify the numbers of different cell populations. Such quantification is not recommended as lung tissue dissociation efficiency may vary in normal, regenerating, and perturbed conditions. Therefore, one need to perform quantification analyses on lung tissues using specific markers for different cell populations described in this manuscript.”

We appreciate your insightful comments. As also suggested by reviewer #2, we deleted the cell proportion assay data generated from the scRNA-seq dataset. Instead, as kindly suggested, we quantified each cell population of AT1 (RAGE+), AT2 (SPC+), Krt8+ (KRT8+), and activated AT2 (LCN2+) using specific markers (Fig. R10). Unfortunately, there are no specific markers for AT1^{med}/AT2^{med} cells. As shown in Figure R10, *Pclaf*-deficient decreased AT1 cells and elevated AT2 cells upon bleomycin damage (Fig. R10A, B, E). Intriguingly, *Pclaf* KO significantly increased the number of KRT8+ cells during lung regeneration compared to WT, while activated AT2 cells were not altered (Fig. R10C, D, E). Considering that KRT8+ cells are intermediate cells from AT2 into AT1 cells^{7,12}, the increased number of KRT8+ cells by *Pclaf* KO indicates that *Pclaf* is required for final differentiation into AT1 cells. Moreover, unaltered numbers of activated AT2 cells by *Pclaf* KO indicate that the role of *Pclaf* in AT2 activation may not exist or can be compensated.

These new data were included in the revised manuscript (Fig. 3F, Supplementary Fig. 7F).

We also revised the result section in the manuscript, accordingly, as shown below.

“Next, we validated such cell lineage trajectory inference results. Immunostaining of *Pclaf* KO lung tissues showed the marked decreased AT1 cells and increased AT2 cells (Fig. 3F), consistent with in silico results. In *Pclaf* KO lung, KRT8⁺ cells (intermediate cells) were significantly reduced, while activated AT2 cells (LCN2⁺) were not changed, compared to WT (Fig. 3F and Supplementary Fig. 7F). Of note is that PAPCs were highly enriched with SPC, a representative marker for AT2 cells (Supplementary Fig. 9), implying that most PAPCs are likely derived from AT2 cells.”

Figure R10. *Pclaf* depletion inhibits AT1 cells and elevates AT2 cells and KRT8⁺ cells.

A-E, *Pclaf* WT and KO mice were treated with phosphate-buffered saline (PBS; control) or bleomycin by intratracheal instillation. At 7 dpi of bleomycin the lung tissues were collected for further analysis. Representative images of lungs that fluorescently immunostained for RAGE and CDH1 (A), RAGE and SPC (B), and KRT8 and CDH1 (C), and that chemically immunostained for LCN2 (D). Quantification of relative cell types (vs. *Pclaf* WT).

C3. “The authors used “PPCs” to describe proliferating pulmonary cells. However, the authors were actually referring or at least characterized *Mki67*⁺ AT2s, which were previously referred to as “proliferating AT2s”. Similarly, the authors claim that they identified *Lcn2* high AT2s. These were previously referred to as “activated AT2s”. The use of different terminology to refer to these previously described cell populations is confusing. Therefore, we suggest that authors reconsider using previously described names.”

We appreciate your valuable comments. We respectfully agree with your suggestion in part. For *Lcn2* high AT2 cells, we have revised it into “activated AT2 cells” as you kindly suggested and according to the previous findings^{7,12}. For the term “PPC”, the *Mki67*⁺ cell cluster contained proliferating AT2 cells in major, as mentioned (Fig. R1). However, *SCGB1A1*⁺ club cells also belonged to this cell cluster (Fig. R1H, J). Thus, the term “proliferating AT2 cell” is not sufficient to represent the whole *Mki67*⁺ cells. Therefore, we changed the nomenclature of PPCs into proliferating alveolar progenitor cells (**PAPCs**), as also suggested by reviewer #1.

Figure R1. PAPCs express markers of AT2 and Club cells.

A, B, Analysis of the scRNA-seq datasets shown in Figure 1A. UMAP-embedding displays cells colored by cell type identity (A).

Feature plots of indicated gene expression in subset of PAPCs (B).

C, D, Analysis of the scRNA-seq datasets shown in Figure 1D. UMAP-embedding displays cells colored by cell type identity (C).

Feature plots of indicated gene expression in subset of PAPCs (D).

E, F, Analysis of the scRNA-seq dataset shown in Figure 3A. UMAP-embedding displays cells colored by cell type identity (E).

Feature plots of indicated gene expression in subset of PAPCs (F).

G, H, Representative images of WT lung at 7 dpi of bleomycin; immunostaining for MKI67 and SPC; MKI67 (G) and SCGB1A1; MKI67 (H).

I, J, *Pclaf-lacZ* mice were instilled with bleomycin (1.4 U/kg). After 7 dpi, lungs were collected and performed whole-mount staining for X-gal. Representative image of lung chemically immunostained for SPC (I).

Representative image of lung chemically immunostained for SCGB1A1 (J).

C4. “The authors described multiple cell populations which they referred to as “transitional cell states”, including PPCs, Krt8+ cells, AT2med/AT1med, Lcn2high AT2s. Further, the authors claim that some of these populations were dysregulated in *Pclaf* KO lungs. These claims need to be validated using specific markers on tissue sections.”

We appreciate your comments. Validating transitional cell states using tissue sections would strengthen our message. Thus, we quantified these transitional cells by immune staining with lung sections of *Pclaf* KO. We used specific markers for each cell type: PAPC (MKI67+), activated AT2 (Lcn2high; LCN2+), and Krt8+ (KRT8+). Unfortunately, there were no specific markers for AT2med/AT1med cells, and double staining of RAGE and SPC was not possible to distinguish these cells due to unclear clear boundaries of RAGE staining and non-epithelial cell types. The immunostaining results of lung tissue showed increased PAPCs (Fig. R7E, F) and KRT8+ cells (Fig. R10C, E) in *Pclaf* KO lung compared to *Pclaf* WT. The LCN2+ activated AT2 cells were not altered by *Pclaf* KO (Fig. R10D, E). Considering that KRT8+ cells are intermediate cells from AT2 into AT1 cells^{7,12} and PAPCs are predicted intermediates in this study, the increased number of KRT8+ cells and PAPCs by *Pclaf* KO indicates that *Pclaf* is required for final differentiation into AT1 cells. Moreover, unaltered numbers of activated AT2 cells by *Pclaf* KO indicate that the role of *Pclaf* in AT2 activation may not exist or can be compensated.

Figure R7. *Pclaf* depletion inhibits damage-induced regeneration of AT1 cells.

E, F, *Pclaf* WT and KO mice were treated with phosphate-buffered saline (PBS; control) or bleomycin by intratracheal instillation. At 3, 7, or 21 dpi of bleomycin or 21 dpi of PBS, the lung tissues were collected for further analysis. Representative images of lungs that fluorescently immunostained for MKI67 and CDH1 at indicated time points (E). Quantification of MKI67+ CDH1+ cells (F).

Figure R10. *Pclaf* depletion inhibits AT1 cells and elevates AT2 cells and KRT8+ cells.

A-E, *Pclaf* WT and KO mice were treated with phosphate-buffered saline (PBS; control) or bleomycin by intratracheal instillation. At 7 dpi of bleomycin the lung tissues were collected for further analysis. Representative images of lungs that fluorescently immunostained for RAGE and CDH1 (A), RAGE and SPC (B), and KRT8 and CDH1 (C), and that chemically immunostained for LCN2 (D). Quantification of relative cell types (vs. *Pclaf* WT).

C5. “What was the efficiency of *Pclaf* deletion in all lungs assessed? There are many “PPEs” in *Pclaf* KO lung scRNA-seq data. Were they not deleted for *Pclaf*?”

We apologize for the confusion. We established *Pclaf* KO *germline* mice using the CRISPR system^{1,2}, which delivers complete KO of the *Pclaf* alleles. Our previous studies showed that *Pclaf* KO did not show any discernible phenotypes^{1,2}. The PCR and sequencing of genomic DNA validated the complete *Pclaf* KO (Fig. R11)^{1,2}.

As you pointed out, we initially expected reduced or absence of PAPCs in *Pclaf* KO lung after damage. However, unexpectedly, the *Pclaf* KO lung showed an elevated number of MKI67+ PAPCs along with inhibited AT1 regeneration during lung regeneration compared to WT in both organoids and mice (Fig. R6; please see the response C1). PCLAF was initially identified as a PCNA-interacting protein to promote cell proliferation. And we found that PCLAF is highly expressed in PAPCs. Therefore, it could be expected that proliferative cells are significantly reduced or lost in *Pclaf* KO lung or organoids. However, it should also be noted that PCLAF modulates various cellular processes independently of PCNA or cell proliferation, especially for the S phase, as described below.

Figure R11. Genotyping of *Pclaf* KO mice.
Genotyping results of littermates for *Pclaf* WT, KO, or heterozygous KO (Het).

PCNA-independent roles of PCLAF

As a leading group in the PCLAF study, we previously have revealed that PCLAF hyperactivates WNT/ β -catenin signaling independently of PCNA in colorectal cancer^{1,32}, promotes pancreatic tumorigenesis via MAPK signaling³³, induces cell plasticity and increases cell stemness of breast cancer³⁰, facilitates stem cell activation and proliferation for intestinal regeneration¹, and promotes lung tumorigenesis via the DREAM complex². PCLAF directly binds to EZH2 to hyperactivate β -catenin-mediated gene transactivation independently of PCNA binding. Similarly, PCLAF binds to RBBP4/7, core components of the DREAM complex, and inhibits the recruitment of RBL2/p130 to the DREAM/MuvB complex, which is also independent of PCNA binding². We also quantified the amount of PCLAF protein in HeLa cells. Although most PCLAF protein shows nuclear localization, approximately 50% of PCLAF binds to PCNA with a one-to-one molar ratio. However, the other 50% of PCLAF protein is not associated with PCNA but with EZH2 or nuclear proteins³². Therefore, PCLAF is not exclusively associated with PCNA or cell proliferation.

Our data suggest that PCLAF plays a role in modulating the cell fate of PAPCs toward AT1 cells rather than proliferation. Thus, PCLAF depletion or inactivation may cause stalled PAPCs and drive toward AT2 cells.

C6. *“The authors used in silico lineage prediction tools and suggest cell trajectories to and from multiple cell populations. For example, in Fig-3H the authors claim bi-directional flow between PPCs and Lcn2 high-AT2s and AT1/AT2 to Lcn2 high-AT2s. Additionally, the authors claim that bidirectional trajectory is missing between PPCs and AT2 (Lcn2 high) cells in Pclaf KO lungs. These claims needs validation using lineage tracing*

models.”

We appreciate your comments. We agree that validation of lineage trajectory will strengthen our message more robustly. To this end, we performed lineage tracing experiments. First, we generated *Pclaf-fl/fl* mice using *Pclaf-LacZ-neo* mice followed by confirming Cre-mediated knock-out by genotyping of *Pclaf-fl/fl* mouse embryonic fibroblasts by Ad-Cre (Fig. R3A, B). Then, we performed lineage-tracing of AT2 cells using *Sftpc-CreERT2*; *LSL-Sun1-EGFP* mice combined with *Pclaf* WT, *Pclaf* KO, or *Pclaf-fl/fl* (*Pclaf* cKO) mice. Both *Pclaf* KO and *Pclaf* cKO mice showed that AT1 cells and KRT8+ cells from AT2 cells were decreased compared to *Pclaf* WT during lung regeneration (Fig. R3C-F). These data consistently indicate that PCLAF is crucial in alveolar cell plasticity, especially for regeneration of AT1 cells. Even though we did not show the exact cellular trajectory in bi-directional flow using a lineage tracing experiment, the increased AT2 cells and decreased AT1 cells traced from AT2 cells support our claim in part. To better understand this bi-directional trajectory, more specific lineage tracing using markers of LCN2+ cells and PAPCs needs to be performed in further study.

Figure R3. *Pclaf* KO inhibits the differentiation of AT1 cells from AT2 cells.

A, Experimental scheme of generating *Pclaf-fl/fl* mice.

B, Mouse embryonic fibroblasts (MEFs) were isolated from *Pclaf-fl/fl* mice. 1 X 10⁷ or 1 X 10⁸ PFU of Ad-Cre were introduced in *Pclaf-fl/fl* MEFs.

Conditional knock-out by Ad-Cre was checked by genotyping.

C-F *Sftpc-CreERT2*; *LSL-Sun1-EGFP*; *Pclaf* WT, *Sftpc-CreERT2*; *LSL-Sun1-EGFP*; *Pclaf* KO, and *Sftpc-CreERT2*; *LSL-Sun1-EGFP*; *Pclaf-fl/fl* mice were instilled with bleomycin. Then mice were injected with tamoxifen for 5 consecutive days. At 14 dpi, lungs were collected and analyzed.

C, Representative images of lungs; immunostaining for RAGE, SPC, and EGFP. White arrow; RAGE+/EGFP+ cell. Red arrow; SPC+/EGFP+ cell. Yellow arrow; RAGE+/SPC+/EGFP+ cell.

D, Enlarged image of RAGE and SPC double-positive cell.

E, Representative images of lungs; immunostaining for KRT8 and EGFP.

White arrow; KRT8+/EGFP+ cell.

F, Quantification of indicated cells in figure R3 C and E.

C7. *“In lines 104 to 106, the authors state that “ While LOs with LuMSCs were mainly differentiated into the bronchiolar type,”. Numerous studies have shown that AT2s generate LO’s with AT2 and AT1 cells in the presence of alveolar fibroblasts. Its not clear why the authors only got bronchiolar type organoids in the presence of LuMSCs?”*

We appreciate your critical comments. As you mentioned, the method of LO culture with LuMSC^{7,34-37} was well established to generate alveolar organoids along with the LO culture with LuEC³⁸. Thus, we tried to culture LOs using both methods. Somehow, LO culture with LuMSC did not generate alveolar organoids *in our hands*. Perhaps it was due to some experimental or technical variation or limitations in our laboratories. Nonetheless, the LO culture with LuEC was well generated in our hands, and they undergo both alveolar and bronchiolar organoids. Thus, we choose to culture LOs with LuEC in this study.

As pointed out, to avoid confusion, we revised the manuscript by deleting the texts and Figures related to LOs with LUMSCs.

C8. *“The authors used “alveolar plasticity” numerous times throughout the manuscript in different contexts. For example, in lines 114-115, the authors state “....plasticity of AT1 and AT2 cell lineage during lung regeneration”. It is not clear what type of plasticity the authors are referring to in the context of AT1 cells? Similarly, the authors used “plasticity” while referring to AT2 to AT1 differentiation. AT2 to AT1 lineage conversion is simply – differentiation. Therefore, we suggest that authors use appropriate terminology.”*

We apologize for confusing terminology. As you mentioned, cell transition from AT2 cells into AT1 cells can be expressed by a simple term, differentiation. However, this simple terminology, differentiation, is widely used in the development area representing the cell transition from stem and/or progenitor cells into mature or terminally differentiated cells³⁹. In contrast, cell plasticity could be a more comprehensive term including de-differentiation, trans-differentiation, phenotypic transition of differentiated cells, interconversion of different stem cell pools, and activation of facultative stem or progenitor cells³⁹⁻⁴³. Thus, we think that “plasticity” could better represent the complex physiology of regeneration rather than differentiation. Lung regeneration is a complex biological process that contains several cellular events, such as the activation of cells that have the potential for stemness, including AT2 cells, de-differentiation, and trans-differentiation^{44,45}.

Although, as suggested, we revised the manuscript to use the term, cell plasticity, in a more consistent manner.

C9. *“In lines 213-214, the authors discussed about markers of transient cell states. One of the citations (#25) did not describe Krt8+ cells. This should be corrected.”*

Our apologies. We misplaced the citation (#25). We have revised the citation (#25) to refer to the right sentence in the revised manuscript.

C10. “Does PCLAF-DREAM complex directly bind “PPC” targets genes to control their fate?”

We appreciate your insightful comment. In this study, we found that the PCLAF-DREAM complex is required for AT1 regeneration through the CLIC4-activated TGFβ signaling pathway. The CLIC4 was identified as a direct target gene of DREAM complex¹⁵ and predicted as a direct target gene of PCLAF-DREAM complex².

To determine whether PCLAF directly transactivates *CLIC4*, we performed chromatin immunoprecipitation (ChIP) assays. According to the ChIP-Atlas database (chp-atlas.org), several components of the DREAM complex, RBBP4, RBBP5, RPPB7, Lin9, E2F1, E2F3, E2F4, E2F6, E2F7, and MYBL2, binds to the *CLIC4* promoter (Fig. R5A). Indeed, ChIP experiments showed that ectopically expressed 3FLAG-PCLAF bound to two regions of proximal promoters of the *CLIC4* in human lung adenocarcinoma cells, H358 cells (Fig. R5B). Consistently, endogenous PCLAF bound to the *CLIC4* proximal promoter (Fig. R5C). Of note is that due to a relatively small number of PCLAF+ cells in the lung, performing ChIP assays of the lung tissues with anti-PCLAF antibodies was not technically feasible.

As we performed rescue experiments for the functional implications in the initial manuscript, the ectopic expression of CLIC4 significantly rescued the decreased AT1 regeneration (Fig. R5D-F) and phosphorylated SMAD3 levels (Fig. R5G, H) observed in *Pclaf* KO LOs. Additionally, external stimulation with TGF-β1 rescued the AT1/AT2 cell ratio in *Pclaf* KO LOs (Fig. R5I-K). Together, these results suggest that the PCLAF-DREAM complex directly binds to the promoters and transactivates *CLIC4* for cell fate change by promoting TGF-β signaling.

Figure R5. The PCLAF-DREAM directly transactivates *CLIC4* to activate TGF-β signaling.

- A**, The DREAM complex components bind to the *CLIC4* promoter. ChIP-Atlas database (chp-atlas.org).
- B**, qPCR analysis using indicated primer sets targeting proximal promoter of DREAM target genes, *CLIC4*, or *ACTB*. ChIP was performed using anti-FLAG antibody. H358 cells ectopically expressing H358 were used for ChIP.
- C**, qPCR analysis using indicated primer sets targeting proximal promoter of DREAM target genes, *CLIC4*, or *ACTB*. ChIP was performed using anti-PCLAF antibody. H358 cells were used for ChIP.
- D-F**, Isolated lung epithelial cells were transduced with RFP- or CLIC4-expressing lentiviruses and cultured into LOs. Images of alveolar type organoids fluorescently immunostained for HOPX (red) and SPC (light blue) (D). Quantification of the HOPX⁺ (E) and SPC⁺ (F) cells.
- G**, Images of alveolar type organoids chemically immunostained for p-SMAD3.

We added these new results in the revised manuscript (Fig. 5 and Supplementary Fig. 13).

We would like to express our gratitude for the insightful comments, which have contributed to enhancing the overall quality and scientific rigor of our study.

References

- 1 Kim, M. J. *et al.* PAF-Myc-Controlled Cell Stemness Is Required for Intestinal Regeneration and Tumorigenesis. *Dev. Cell* **44**, 582-+, doi:10.1016/j.devcel.2018.02.010 (2018).
- 2 Kim, M. J. *et al.* PAF remodels the DREAM complex to bypass cell quiescence and promote lung tumorigenesis. *Mol Cell* **81**, 1698-1714.e1696, doi:10.1016/j.molcel.2021.02.001 (2021).
- 3 Wang, P. *et al.* Disrupting the DREAM complex enables proliferation of adult human pancreatic β cells. *J Clin Invest* **132**, doi:10.1172/jci157086 (2022).
- 4 Chen, Z. *et al.* A Cdh1-FoxM1-Apc axis controls muscle development and regeneration. *Cell Death Dis* **11**, 180, doi:10.1038/s41419-020-2375-6 (2020).
- 5 Barkauskas, C. E. *et al.* Type 2 alveolar cells are stem cells in adult lung. *J Clin Invest* **123**, 3025-3036, doi:10.1172/jci68782 (2013).
- 6 Chung, M. I., Bujnis, M., Barkauskas, C. E., Kobayashi, Y. & Hogan, B. L. M. Niche-mediated BMP/SMAD signaling regulates lung alveolar stem cell proliferation and differentiation. *Development* **145**, doi:10.1242/dev.163014 (2018).
- 7 Choi, J. *et al.* Inflammatory Signals Induce AT2 Cell-Derived Damage-Associated Transient Progenitors that Mediate Alveolar Regeneration. *Cell Stem Cell* **27**, 366-382.e367, doi:10.1016/j.stem.2020.06.020 (2020).
- 8 Kathiriya, J. J., Brumwell, A. N., Jackson, J. R., Tang, X. & Chapman, H. A. Distinct Airway Epithelial Stem Cells Hide among Club Cells but Mobilize to Promote Alveolar Regeneration. *Cell Stem Cell* **26**, 346-358.e344, doi:10.1016/j.stem.2019.12.014 (2020).
- 9 Tashiro, J. *et al.* Exploring Animal Models That Resemble Idiopathic Pulmonary Fibrosis. *Front Med (Lausanne)* **4**, 118, doi:10.3389/fmed.2017.00118 (2017).
- 10 Hay, J., Shahzeidi, S. & Laurent, G. Mechanisms of bleomycin-induced lung damage. *Arch Toxicol* **65**, 81-94, doi:10.1007/bf02034932 (1991).
- 11 Jenkins, R. G. *et al.* An Official American Thoracic Society Workshop Report: Use of Animal Models for the Preclinical Assessment of Potential Therapies for Pulmonary Fibrosis. *Am J Respir Cell Mol Biol* **56**, 667-679, doi:10.1165/rcmb.2017-0096ST (2017).
- 12 Strunz, M. *et al.* Alveolar regeneration through a Krt8+ transitional stem cell state that persists in human lung fibrosis. *Nat Commun* **11**, 3559, doi:10.1038/s41467-020-17358-3 (2020).
- 13 Gulati, G. S. *et al.* Single-cell transcriptional diversity is a hallmark of developmental potential. *Science* **367**, 405-411, doi:10.1126/science.aax0249 (2020).
- 14 Street, K. *et al.* Slingshot: cell lineage and pseudotime inference for single-cell transcriptomics. *BMC Genomics* **19**, 477, doi:10.1186/s12864-018-4772-0 (2018).
- 15 Fischer, M., Grossmann, P., Padi, M. & DeCaprio, J. A. Integration of TP53, DREAM, MMB-FOXM1 and RB-E2F target gene analyses identifies cell cycle gene regulatory networks. *Nucleic Acids Res.* **44**, 6070-6086, doi:10.1093/nar/gkw523 (2016).
- 16 Alsafadi, H. N. *et al.* An ex vivo model to induce early fibrosis-like changes in human precision-cut lung slices. *Am J Physiol Lung Cell Mol Physiol* **312**, L896-L902, doi:10.1152/ajplung.00084.2017 (2017).
- 17 Riemondy, K. A. *et al.* Single cell RNA sequencing identifies TGF β as a key regenerative cue following LPS-induced lung injury. *JCI Insight* **5**, doi:10.1172/jci.insight.123637 (2019).
- 18 Kim, K. K. *et al.* Alveolar epithelial cell mesenchymal transition develops in vivo during pulmonary fibrosis and is regulated by the extracellular matrix. *Proc Natl Acad Sci U S A* **103**, 13180-13185, doi:10.1073/pnas.0605669103 (2006).
- 19 Fowell, D. J. & Kim, M. The spatio-temporal control of effector T cell migration. *Nat Rev Immunol* **21**, 582-596, doi:10.1038/s41577-021-00507-0 (2021).
- 20 Araque, A. *et al.* Gliotransmitters travel in time and space. *Neuron* **81**, 728-739, doi:10.1016/j.neuron.2014.02.007 (2014).
- 21 Barrientos, S., Stojadinovic, O., Golinko, M. S., Brem, H. & Tomic-Canic, M. Growth factors and cytokines in wound healing. *Wound Repair Regen* **16**, 585-601, doi:10.1111/j.1524-475X.2008.00410.x (2008).
- 22 Silva, B. C. & Bilezikian, J. P. Parathyroid hormone: anabolic and catabolic actions on the skeleton. *Curr Opin Pharmacol* **22**, 41-50, doi:10.1016/j.coph.2015.03.005 (2015).
- 23 Keller, G. Embryonic stem cell differentiation: emergence of a new era in biology and medicine. *Genes Dev* **19**, 1129-1155, doi:10.1101/gad.1303605 (2005).

- 24 Liu, Q. *et al.* Lung regeneration by multipotent stem cells residing at the bronchioalveolar-duct junction. *Nat Genet* **51**, 728-738, doi:10.1038/s41588-019-0346-6 (2019).
- 25 Kobayashi, Y. *et al.* Persistence of a regeneration-associated, transitional alveolar epithelial cell state in pulmonary fibrosis. *Nature Cell Biology* **22**, 934-946, doi:10.1038/s41556-020-0542-8 (2020).
- 26 Rock, J. R. *et al.* Multiple stromal populations contribute to pulmonary fibrosis without evidence for epithelial to mesenchymal transition. *Proc Natl Acad Sci U S A* **108**, E1475-1483, doi:10.1073/pnas.1117988108 (2011).
- 27 Nguyen, T. M. *et al.* Stretch increases alveolar type 1 cell number in fetal lungs through ROCK-Yap/Taz pathway. *Am J Physiol Lung Cell Mol Physiol* **321**, L814-L826, doi:10.1152/ajplung.00484.2020 (2021).
- 28 Emanuele, M. J., Ciccia, A., Elia, A. E. & Elledge, S. J. Proliferating cell nuclear antigen (PCNA)-associated KIAA0101/PAF15 protein is a cell cycle-regulated anaphase-promoting complex/cyclosome substrate. *Proc Natl Acad Sci U S A* **108**, 9845-9850, doi:10.1073/pnas.1106136108 (2011).
- 29 De Biasio, A. *et al.* Structure of p15(PAF)-PCNA complex and implications for clamp sliding during DNA replication and repair. *Nat Commun* **6**, 6439, doi:10.1038/ncomms7439 (2015).
- 30 Wang, X. *et al.* PAF-Wnt signaling-induced cell plasticity is required for maintenance of breast cancer cell stemness. *Nat Commun* **7**, 10633, doi:10.1038/ncomms10633 (2016).
- 31 He, G. W. *et al.* Optimized human intestinal organoid model reveals interleukin-22-dependency of paneth cell formation. *Cell Stem Cell* **29**, 1333-1345.e1336, doi:10.1016/j.stem.2022.08.002 (2022).
- 32 Jung, H. Y. *et al.* PAF and EZH2 induce Wnt/ β -catenin signaling hyperactivation. *Mol Cell* **52**, 193-205, doi:10.1016/j.molcel.2013.08.028 (2013).
- 33 Jun, S. *et al.* PAF-mediated MAPK signaling hyperactivation via LAMTOR3 induces pancreatic tumorigenesis. *Cell Rep* **5**, 314-322, doi:10.1016/j.celrep.2013.09.026 (2013).
- 34 Chanda, D. *et al.* Mesenchymal stromal cell aging impairs the self-organizing capacity of lung alveolar epithelial stem cells. *Elife* **10**, doi:10.7554/eLife.68049 (2021).
- 35 Tamai, K. *et al.* iPSC-derived mesenchymal cells that support alveolar organoid development. *Cell Rep Methods* **2**, 100314, doi:10.1016/j.crmeth.2022.100314 (2022).
- 36 Leeman, K. T., Pessina, P., Lee, J. H. & Kim, C. F. Mesenchymal Stem Cells Increase Alveolar Differentiation in Lung Progenitor Organoid Cultures. *Sci Rep* **9**, 6479, doi:10.1038/s41598-019-42819-1 (2019).
- 37 Lee, J. H. *et al.* Anatomically and Functionally Distinct Lung Mesenchymal Populations Marked by Lgr5 and Lgr6. *Cell* **170**, 1149-1163.e1112, doi:10.1016/j.cell.2017.07.028 (2017).
- 38 Lee, J. H. *et al.* Lung stem cell differentiation in mice directed by endothelial cells via a BMP4-NFATc1-thrombospondin-1 axis. *Cell* **156**, 440-455, doi:10.1016/j.cell.2013.12.039 (2014).
- 39 Sánchez Alvarado, A. & Yamanaka, S. Rethinking differentiation: stem cells, regeneration, and plasticity. *Cell* **157**, 110-119, doi:10.1016/j.cell.2014.02.041 (2014).
- 40 Varga, J. & Greten, F. R. Cell plasticity in epithelial homeostasis and tumorigenesis. *Nat Cell Biol* **19**, 1133-1141, doi:10.1038/ncb3611 (2017).
- 41 Borok, Z. *et al.* Cell plasticity in lung injury and repair: report from an NHLBI workshop, April 19-20, 2010. *Proc Am Thorac Soc* **8**, 215-222, doi:10.1513/pats.201012-067CB (2011).
- 42 Merrell, A. J. & Stanger, B. Z. Adult cell plasticity in vivo: de-differentiation and transdifferentiation are back in style. *Nat Rev Mol Cell Biol* **17**, 413-425, doi:10.1038/nrm.2016.24 (2016).
- 43 Tata, A., Chow, R. D. & Tata, P. R. Epithelial cell plasticity: breaking boundaries and changing landscapes. *EMBO Rep* **22**, e51921, doi:10.15252/embr.202051921 (2021).
- 44 Basil, M. C. *et al.* The Cellular and Physiological Basis for Lung Repair and Regeneration: Past, Present, and Future. *Cell Stem Cell* **26**, 482-502, doi:10.1016/j.stem.2020.03.009 (2020).
- 45 Kotton, D. N. & Morrissey, E. E. Lung regeneration: mechanisms, applications and emerging stem cell populations. *Nat Med* **20**, 822-832, doi:10.1038/nm.3642 (2014).

REVIEWER COMMENTS

Reviewer #2 (Remarks to the Author):

The authors have done an excellent job responding to the critiques. The lineage tracing particularly strengthens the manuscript. Only minor issues remain:

1. Fig. 3G. The method of quantifying ATI cells is still confusing. Is the first “RAGE+” set quantified as the % of GFP+ cells that are RAGE+? In other words, the number of RAGE+ cells / the number of GFP+ cells? It is impossible to determine the number of RAGE+ cells by fluorescence microscopy. However, given all the other strong data supporting the role of Pclaf in ATI cell regeneration, particularly the lung organoid data, the conclusion is likely to be correct.
2. Please check the reference to Fig. 2K, L in Line 151 to be sure it is correct.
3. In Fig. 5L, M, please check the grey circles vs. squares.
4. Line 293 – please check that the paper referenced actually studied the role of Notch.
5. Line 332 should be revised, as it appears to overstate the role of TGFb. Fig. 5L shows that TGFb has no effect in WT cells.

Reviewer #3 (Remarks to the Author):

I commend the authors effort to address my prior comments. However, some data in the revised manuscript is confusing and does not clarify my prior comments. See below for specific comments.

1. In line175-176, the authors state that “...KRT8+ (intermediate cells) were significantly reduced...” in PCLAF KO mouse lungs. However, their data in Fig.3F contradicts this statement. The quantification data shows an increase KRT8+ cells in KO lungs compared to controls. Moreover, KRT8+ cells quantification data in Fig.3F doesn’t match KRT8+ cell numbers in Fig.3G. Similarly, the authors made similar contrasting statements in the rebuttal letter (see “C2” response).

It is not clear why there is a discrepancy between quantification data in Fig-3F and 3G?

2. In response to my prior comment, the authors responded (C4) that “Unfortunately, there were no specific markers for AT2med/AT1med cells, and double staining of RAGE and SPC was not possible to distinguish these cells due to unclear clear boundaries of RAGE staining and non-epithelial cell types”. However, the authors quantified RAGE and SPC double stained cells in Fig3G. How come the authors were able to quantify cells in one context but not in other?

To quantify cells, authors could also HOPX, a nuclear localized AT1 marker.

3. In rebuttal letter, authors state (in C4) that “Unfortunately, there were no specific markers for AT2med/AT1med cells.....”. Why did the authors label these cells as AT2med/AT1med cells if there are no markers that can identify or distinguish these cells from others? Further, heatmap in Supplementary Fig-1A, the cells designated as AT2med/AT1med appear to show two subsets. The gene expression in one subset seem to resemble AT2 cells and another one matches AT1 cells. Therefore, the authors need to reconsider their cell clustering and cell annotation and revise the figures.

4. The authors claim that loss of Pclaf leads to a decrease in At2s differentiation into AT1. However, it is not clear to me why Pclaf KO lungs show AT1 differentiation defects despite its high expression in proliferating cells (PAPCs)? The authors need to discuss the potential mechanism for such unexpected phenotype.

Responses to Comments

Reviewer 2

“The authors have done an excellent job responding to the critiques. The lineage tracing particularly strengthens the manuscript. Only minor issues remain:”

1. “Fig. 3G. The method of quantifying ATI cells is still confusing. Is the first “RAGE+” set quantified as the % of GFP+ cells that are RAGE+? In other words, the number of RAGE+ cells / the number of GFP+ cells? It is impossible to determine the number of RAGE+ cells by fluorescence microscopy. However, given all the other strong data supporting the role of *Pclaf* in ATI cell regeneration, particularly the lung organoid data, the conclusion is likely to be correct.”

Our apologies for the mislabeling in Figure 3G. As well pointed out, RAGE+ cells originated from AT2 cells were calculated by the % of the number of RAGE+ and GFP+ double-positive cells per all GFP+ cells (i.e., cells genetically labeled by *Sftpc-CreERT2*; *Sun1-EGFP*). Briefly, we performed double or triple immunostaining to detect each cell type: RAGE+ (double staining for RAGE and GFP), SPC+ (SPC+ and GFP+), RAGE+/SPC+ (triple staining for RAGE, SPC, and GFP), and KRT8+ (KRT8+ and GFP double staining) (Fig. R1, Supplementary Fig. 10). Of note, Sun1-GFP staining was quite distinct by nuclear membrane pattern (due to Sun1 nuclear membrane protein fusion) (Fig. R1, Supplementary Fig. 10), which made it easier to detect and quantify.

We revised Figure 3G and Figure legend, accordingly.

Figure 3G. Previous vs. Revised

Figure R1. Quantification of AT1 (RAGE+), AT2 (SPC+), and AT2^{med}/AT1^{med} (RAGE+/SPC+) cells from *Pclaf* WT, KO (germline), and conditional KO mice combined with AT2 cell lineage-tracing strains.

Quantification of lineage-traced cells from *Sftpc-CreERT2; Sun1-GFP* (GFP+) cells in Figure 3G and Supplementary Figure 10. Random spots ($n \geq 10$) from each group were quantified. Cells surrounding RAGE signals within 2 μm were measured as RAGE+ cells.

2. “Please check the reference to Fig. 2K, L in Line 151 to be sure it is correct.”

We revised those. Thank you.

3. “In Fig. 5L, M, please check the grey circles vs. squares.”

As suggested, we added *P* values (NS; not significant) to Figure 5L. There was no significant difference in AT1 cell numbers of *Pclaf* WT LOs between vehicle control and TGF- β treatment (Fig. 5L, lane 1-3). However, TGF- β decreased the AT1 cell number of *Pclaf* WT LOs (Fig. 5M, lane 1-3).

Figure 5L,M. Previous vs. Revised

4. “Line 293 – please check that the paper referenced actually studied the role of Notch.”

Our apologies. We revised it.

“Several developmental signaling pathways (BMP⁵⁰, NOTCH⁵¹, and YAP/TAZ⁵²) contribute to cell transition from AT2 to AT1 during regeneration.

51. Finn, J. et al. Dlk1-Mediated Temporal Regulation of Notch Signaling Is Required for Differentiation of Alveolar Type II to Type I Cells during Repair. *Cell Rep* 26, 2942-2954.e2945, doi:10.1016/j.celrep.2019.02.046 (2019).”

5. “Line 332 should be revised, as it appears to overstate the role of TGFb. Fig. 5L shows that TGFb has no effect in WT cells.”

As suggested, we revised the text (highlighted) as follows.

“Our results showed that TGF- β signaling activation is sufficient to complement PCLAF loss for subsequent AT1 cell regeneration.”

Reviewer 3

"I commend the authors effort to address my prior comments. However, some data in the revised manuscript is confusing and does not clarify my prior comments. See below for specific comments."

1. "In line 175-176, the authors state that "...KRT8+ (intermediate cells) were significantly reduced..." in PCLAF KO mouse lungs. However, their data in Fig.3F contradicts this statement. The quantification data shows an increase KRT8+ cells in KO lungs compared to controls. Moreover, KRT8+ cells quantification data in Fig.3F doesn't match KRT8+ cell numbers in Fig.3G. Similarly, the authors made similar contrasting statements in the rebuttal letter (see "C2" response). It is not clear why there is a discrepancy between quantification data in Fig-3F and 3G?"

We appreciate your comments. Our apologies for the mislabeling in Figure 3G. Luckily, Reviewer 2 made a similar comment. Briefly, "GFP+" was missing in the X-axis. We quantified RAGE+, SPC+, RAGE+/SPC+, and KRT8+ cells out of all GFP+ cells, which are derived from the AT2 cells (*Sftpc*^{CreERT2} [*Sftpc-CreERT2*]; *Rosa26*^{Sun1^{GFP}} [*Sun1-GFP*] lineage-tracing mice). Briefly, we performed double or triple immunostaining to detect each cell type: RAGE+ (double staining for RAGE and GFP), SPC+ (SPC+ and GFP+), RAGE+/SPC+ (triple staining for RAGE, SPC, and GFP), and KRT8+ (KRT8+ and GFP double staining) (Fig. R1, Supplementary Fig. 10). We revised Figure 3G, texts, and legends accordingly.

Figure 3G. Previous vs. Revised

Figure R1. Quantification of AT1 (RAGE+), AT2 (SPC+), and AT2^{med}/AT1^{med} (RAGE+/SPC+) cells from Pclaf WT, KO (germline), and conditional KO mice combined with AT2 cell lineage-tracing strains.

Quantification of lineage-traced cells from *Sftpc-CreERT2*; *Sun1-GFP* (GFP+) cells in Figure 3G and Supplementary Figure 10. Random spots ($n \geq 10$) from each group were quantified. Cells surrounding RAGE signals within 2 μm were measured as RAGE+ cells.

2. “In response to my prior comment, the authors responded (C4) that “Unfortunately, there were no specific markers for AT2^{med}/AT1^{med} cells, and double staining of RAGE and SPC was not possible to distinguish these cells due to unclear clear boundaries of RAGE staining and non-epithelial cell types”. However, the authors quantified RAGE and SPC double stained cells in Fig3G. How come the authors were able to quantify cells in one context but not in other? To quantify cells, authors could also HOPX, a nuclear localized AT1 marker.”

Reviewer 2 previously recommended repeating all RAGE+ and cell quantification with E-cadherin/CDH1 co-immunostaining to ensure that these cells are indeed epithelial cells. So, we repeated all experiments with CDH1 co-staining (1st revision). Yes, we responded that “double staining for RAGE and SPC was not possible...non-epithelial types”. For clarification, we meant to indicate that triple staining (RAGE+, SPC+, and CDH1+) is not feasible. To follow Reviewer 2’s suggestion, i.e., adding CDH1, we had to quantify RAGE+ and SPC+ double-positive cells by triple staining for RAGE, SPC, and CDH1. However, our triple staining for RAGE+, SPC+, and CDH1+ did not work due to incompatible antigen retrieval buffers between RAGE, SPC, and CDH1 antibodies. Conversely, triple staining for RAGE+, SPC+, and GFP+ worked well (Fig. 3G).

3. *“In rebuttal letter, authors state (in C4) that “Unfortunately, there were no specific markers for AT2med/AT1med cells.....”. Why did the authors label these cells as AT2med/AT1med cells if there are no markers that can identify or distinguish these cells from others? Further, heatmap in Supplementary Fig-1A, the cells designated as AT2med/AT1med appear to show two subsets. The gene expression in one subset seem to resemble AT2 cells and another one matches AT1 cells. Therefore, the authors need to reconsider their cell clustering and cell annotation and revise the figures.”*

“Why did the authors label these cells as AT2med/AT1med cells if there are no markers that can identify or distinguish these cells from others?”

Our apologies for misleading. We meant “no specific markers” biomarkers for practical experimental detection or cell sorting. We annotated that AT2med/AT1med cell cluster by distinct ‘transcriptional signature’ displaying both AT1 and AT2 gene expression patterns. Cell annotation from scRNA-seq is highly subjective and varies in the literature. Thus, we tried to annotate cells with minimum bias. Both mouse and human public scRNA-seq datasets (Fig. 1, Supplementary Fig. 1) showed this distinct cell cluster expressing both AT1 and AT2 marker genes after “**unsupervised**” clustering, consistent with previous studies ^{1,2}.

“Further, heatmap in Supplementary Fig-1A, the cells designated as AT2med/AT1med appear to show two subsets. The gene expression in one subset seem to resemble AT2 cells and another one matches AT1 cells. Therefore, the authors need to reconsider their cell clustering and cell annotation and revise the figures.”

We respectfully disagree with this comment. For clarification, cell clustering was done by “unsupervised analysis”. From the same perspective of AT2med/AT1med sub-dividing, Krt8+ cell clusters and PAPC clusters could have been divided into two additional sub-clusters (Supplementary Fig. 1A). Therefore, dividing AT2med/AT1med (in a supervised manner) may impair the overall integrity of scRNA-seq data analysis.

4. *“The authors claim that loss of Pclaf leads to a decrease in At2s differentiation into AT1. However, it is not clear to me why Pclaf KO lungs show AT1 differentiation defects despite its high expression in proliferating cells (PAPCs)? The authors need to discuss the potential mechanism for such unexpected phenotype.”*

We appreciate your comment. Here is what we think.

1. Accumulation of precursor or intermediate cells upon inhibition of differentiation

Several studies have shown that disruption of cell differentiation induces the accumulation of intermediate cells ³⁻⁵. During lung regeneration, IL-1 β prevents the terminal differentiation of AT1 cells from AT2 cells along with the accumulating of intermediate cells, Krt8-expressing the damage-associated transient progenitors ⁴. Additionally, *Ndufs2* conditional knockout using *Sftpc-Cre* induces the accumulation of the transitioning cells, which share the transcriptional features of AT1 and AT2 cells, and inhibits terminal differentiation into AT1 cells ³. Therefore, it is plausible that increased PAPCs by *Pclaf* KO might be the consequence of inhibition of AT1 differentiation.

2. Cell differentiation vs. cell proliferation

During development or tissue regeneration, cell proliferation and cell differentiation are inversely correlated and often counteract each other⁶. Thus, it is also possible that increased cell proliferation of PAPCs by *Pclaf* KO might inhibit AT1 cell differentiation.

We added this to the Discussion of the revised manuscript.

“Unexpectedly, *Pclaf* KO lungs exhibited increased MKI67+ PAPCs during regeneration compared to *Pclaf* WT (Supplementary Fig. 3B, C). Consistently, *Pclaf* KO LOs showed elevated PAPCs (Supplementary Fig. 5F, G). These results may be the consequences of inhibition of AT1 differentiation. Recent studies have shown the accumulation of intermediate cells along with the disrupted AT1 differentiation during lung regeneration^{33,59}. On the other hand, during development or tissue regeneration, cell proliferation and cell differentiation are inversely correlated and often counteract each other⁶⁰. Thus, it is also possible that increased cell proliferation of PAPCs by *Pclaf* KO might inhibit AT1 cell differentiation. The further detailed mechanism of PAPCs in lung regeneration remains to be determined in future study.”

References

- 1 Duong, T. E. *et al.* A single-cell regulatory map of postnatal lung alveologenesis in humans and mice. *Cell Genom* **2**, doi:10.1016/j.xgen.2022.100108 (2022).
- 2 Guo, M. *et al.* Single cell RNA analysis identifies cellular heterogeneity and adaptive responses of the lung at birth. *Nat Commun* **10**, 37, doi:10.1038/s41467-018-07770-1 (2019).
- 3 Han, S. *et al.* Mitochondrial integrated stress response controls lung epithelial cell fate. *Nature* **620**, 890-897, doi:10.1038/s41586-023-06423-8 (2023).
- 4 Choi, J. *et al.* Inflammatory Signals Induce AT2 Cell-Derived Damage-Associated Transient Progenitors that Mediate Alveolar Regeneration. *Cell Stem Cell* **27**, 366-382.e367, doi:10.1016/j.stem.2020.06.020 (2020).
- 5 Rosado-Olivieri, E. A., Anderson, K., Kenty, J. H. & Melton, D. A. YAP inhibition enhances the differentiation of functional stem cell-derived insulin-producing β cells. *Nat Commun* **10**, 1464, doi:10.1038/s41467-019-09404-6 (2019).
- 6 Ruijtenberg, S. & van den Heuvel, S. Coordinating cell proliferation and differentiation: Antagonism between cell cycle regulators and cell type-specific gene expression. *Cell Cycle* **15**, 196-212, doi:10.1080/15384101.2015.1120925 (2016).

REVIEWER COMMENTS

Reviewer #2 (Remarks to the Author):

Major Comments

1. POINT 1 FROM PRIOR REVIEW: Quantitation in Fig. 3G is still problematic, a concern also raised by Reviewer 3. The authors make the excellent point that Sun1+ cells are easy to count because Sun1 is nuclear. By the same reasoning, how do they count RAGE+ cells? In the upper left panel of Figure R1, it is impossible to tell whether a Sun1+ cell is also RAGE+ because RAGE (plasma membrane) and GFP (nuclear) are in different compartments and RAGE+ cells cannot be counted.

Minor Comments

1. The manuscript is difficult to read in general, which I think will limit its reception by the Nat Commun audience and the field at large. For example, the abstract should be understandable on its own – however, most readers will not know what PCLAF or DREAM is. After reading only the abstract, I do not have a sense of what the manuscript is about – when the authors say PCLAF remodels the DREAM complex for cell cycle re-entry, do they mean that PCLAF-dependent DREAM complex remodeling is necessary for cell cycle re-entry in general? In that case, the next sentence is not really an interesting finding, but one to be expected – that PCLAF is enriched in proliferating cells. More importantly, based on reading the abstract alone, one might conclude that in PCLAF KO, there are less AT1 cells because there are less AT2 cells to differentiate into AT1 cells. This would be a major limitation of this work. The authors need to explain to the reader what PCLAF and DREAM are, not merely defining the abbreviations but briefly explaining what they are. One might guess that DREAM is a transcription factor because “target genes” and “transactivates” are mentioned, but the reader should not have to guess. Another example is the statement that TGFb signaling “regulates the balance between AT1 and AT2 cells”. What direction does this regulation go in? Does it increase or decrease AT1 cells? Similarly, the statement that “a drug candidate....was identified and validated in organoids and mice” – validated to show what? Words like “validated” and “regulates” do not let the reader know the results. The same could be said about the last sentence – “controlling alveolar cell plasticity” – how does it control it? What exactly is the effect of PCLAF-DREAM on the cells? Admittedly, abstracts are difficult to write because of the word limit, but it could be much more clear. More importantly, the rest of the manuscript is also difficult to read, which limits the reader’s ability to understand what was done, how outcomes were measured, and what the data really shows.

2. POINT 3 FROM PRIOR REVIEW: The grey circles and squares in Fig. 5L have not been corrected. The figure presented in the rebuttal letter does not match the figure in the manuscript, which still shows grey circles on the legend but not in the figure.

3. POINT 5 FROM PRIOR REVIEW: The statement no longer overstates the role of TGFb, but it is unclear what the term “complement” means. It appears that they may mean that TGFb “rescues”

the PCLAF KO phenotype. It's also unclear what "subsequent" means – subsequent to what?
Another example of the difficulty understanding what the manuscript is actually showing.

Reviewer #3 (Remarks to the Author):

I have no further comments.

Responses to Comments

Reviewer #2

Major Comments

1. "POINT 1 FROM PRIOR REVIEW: Quantitation in Fig. 3G is still problematic, a concern also raised by Reviewer 3. The authors make the excellent point that Sun1+ cells are easy to count because Sun1 is nuclear. By the same reasoning, how do they count RAGE+ cells? In the upper left panel of Figure R1, it is impossible to tell whether a Sun1+ cell is also RAGE+ because RAGE (plasma membrane) and GFP (nuclear) are in different compartments and RAGE+ cells cannot be counted."

We appreciate your comment. To better quantify AT1+ cells, we alternatively used a nuclear marker, which goes well with Sun1-GFP localized in the inner nuclear membrane. The HOPX is a nuclear marker for AT1 cells^{1,2}. Consistent with our previous results, AT1 cells derived from AT2 cells (EGFP and HOPX double-positive) were significantly decreased in both *Pclaf* KO and *Pclaf* cKO mice during lung regeneration (Fig. R1). We added these results to the revised manuscript (Fig. 3G, H).

Figure R1. G, H, Quantification of cells derived from AT2 cells using AT2 cell lineage-tracing animal model. The lung tissues of *Sftpc-CreERT2; Sun1-GFP* lineage-tracing mice (14 dpi with tamoxifen for 7 days) were analyzed by immunostaining (G) and quantification (H). Each cell type was detected by co-immunostaining with an anti-GFP antibody (Supplementary Figure 10) and calculated by their cell numbers per GFP+ cells. Sun1-GFP is localized in the inner nuclear membrane; Student's *t*-test.

Minor Comments

1. “The manuscript is difficult to read in general, which I think will limit its reception by the Nat Commun audience and the field at large. For example, the abstract should be understandable on its own – however, most readers will not know what PCLAF or DREAM is. After reading only the abstract, I do not have a sense of what the manuscript is about – when the authors say PCLAF remodels the DREAM complex for cell cycle re-entry, do they mean that PCLAF-dependent DREAM complex remodeling is necessary for cell cycle re-entry in general? In that case, the next sentence is not really an interesting finding, but one to be expected – that PCLAF is enriched in proliferating cells. More importantly, based on reading the abstract alone, one might conclude that in PCLAF KO, there are less AT1 cells because there are less AT2 cells to differentiate into AT1 cells. This would be a major limitation of this work. The authors need to explain to the reader what PCLAF and DREAM are, not merely defining the abbreviations but briefly explaining what they are. One might guess that DREAM is a transcription factor because “target genes” and “transactivates” are mentioned, but the reader should not have to guess. Another example is the statement that TGF β signaling “regulates the balance between AT1 and AT2 cells”. What direction does this regulation go in? Does it increase or decrease AT1 cells? Similarly, the statement that “a drug candidate...was identified and validated in organoids and mice” – validated to show what? Words like “validated” and “regulates” do not let the reader know the results. The same could be said about the last sentence – “controlling alveolar cell plasticity” – how does it control it? What exactly is the effect of PCLAF-DREAM on the cells? Admittedly, abstracts are difficult to write because of the word limit, but it could be much more clear. More importantly, the rest of the manuscript is also difficult to read, which limits the reader’s ability to understand what was done, how outcomes were measured, and what the data really shows.”

We appreciate your comments. As suggested, we revised the abstract.

“Cell plasticity, changes in cell fate, is essential for tissue regeneration. In the lung, failure of regeneration leads to diseases, including fibrosis. However, the mechanisms governing alveolar cell plasticity during lung repair remain elusive. The dimerization partner, RB-like, E2F, and multi-vulval class B (DREAM) complex regulates cell quiescence and proliferation depending on its binding partners. We previously showed that PCNA clamp-associated factor (PCLAF) remodels the DREAM complex, shifting the balance from cell quiescence towards cell proliferation. Here, we found that PCLAF expression is specific to proliferating lung progenitor cells, along with the DREAM target genes by lung injury. Genetic ablation of *Pclaf* impaired alveolar type I (AT1) cell regeneration from alveolar type II (AT2) cells, leading to lung fibrosis. Mechanistically, the PCLAF-DREAM complex transactivates *CLIC4*, triggering TGF- β signaling, which promotes AT1 cell generation from AT2 cells. Furthermore, a drug candidate that mimics the PCLAF-DREAM transcriptional signature increases AT2 cell plasticity, preventing lung fibrosis in organoids and mice. Our study reveals the unexpected role of the PCLAF-DREAM axis in promoting alveolar cell plasticity, beyond cell proliferation control, proposing a potential therapeutic avenue for lung fibrosis prevention.”

2. POINT 3 FROM PRIOR REVIEW: The grey circles and squares in Fig. 5L have not been corrected. The figure presented in the rebuttal letter does not match the figure in the manuscript, which still shows grey circles on the legend but not in the figure.

The Figure in the rebuttal letter does match the Figure in the manuscript. This is the Figure in the 2nd revised manuscript (Fig. R2). Figure R3 is from the rebuttal letter (Fig. R3). Your specific comment would be appreciated.

Figure R2. Figure in the 2nd revised manuscript.

Figure R3. Figure in the rebuttal letter.

3. "POINT 5 FROM PRIOR REVIEW: The statement no longer overstates the role of TGFb, but it is unclear what the term "complement" means. It appears that they may mean that TGFb "rescues" the PCLAF KO phenotype. It's also unclear what "subsequent" means – subsequent to what? Another example of the difficulty understanding what the manuscript is actually showing."

We revised the sentence.

"Our results showed that TGF-β signaling activation is sufficient to rescue the *Pclaf* KO-suppressed AT1 cell regeneration in the context of PAPCs."

References

- 1 Barkauskas, C. E. *et al.* Type 2 alveolar cells are stem cells in adult lung. *J Clin Invest* **123**, 3025-3036 (2013). <https://doi.org/10.1172/jci68782>
- 2 Jain, R. *et al.* Plasticity of Hopx(+) type I alveolar cells to regenerate type II cells in the lung. *Nat Commun* **6**, 6727 (2015). <https://doi.org/10.1038/ncomms7727>